# iWorld-Bench: A Benchmark for Interactive World Models with a Unified Action Generation Framework

**Jianjie Fang** [* 1]  **Yingshan Lei** [* 1]  **Qin Wan** [* 1]  **Ziyou Wang** [2]  **Yuchao Huang** [2]  **Yongyan Xu** [3]  **Baining Zhao** [1]
**Weichen Zhang** [1]  **Chen Gao** [1]  **Xinlei Chen** [1]  **Yong Li** [1]

🌐 *Project Page*        🗄 *Dataset*        ⌂ *Code*        🏆 *Leaderboard*

## Abstract

Achieving Artificial General Intelligence (AGI) requires agents that learn and interact adaptively, with interactive world models providing scalable environments for perception, reasoning, and action. Yet current research still lacks large-scale datasets and unified benchmarks to evaluate their physical interaction capabilities. To address this, we propose **iWorld-Bench**, a comprehensive benchmark for training and testing world models on interaction-related abilities such as distance perception and memory. We construct a diverse dataset with 330k video clips and select 2.1k high-quality samples covering varied perspectives, weather, and scenes. As existing world models differ in interaction modalities, we introduce an **Action Generation Framework** to unify evaluation and design six task types, generating 4.9k test samples. These tasks jointly assess model performance across **visual generation, trajectory following, and memory**. Evaluating 14 representative world models, we identify key limitations and provide insights for future research. The iWorld-Bench model leaderboard is publicly available at iWorld-Bench.com.

## 1. Introduction

Recently, world models (Ha & Schmidhuber, 2018; Hafner et al., 2023; Ball et al., 2025; Zhao et al., 2025b) have garnered significant attention for their ability to understand the world through interaction and predict future environmental changes. Unlike general video generation models (Liu et al., 2024; Wan et al., 2025a) and embodied world mdoles (Guo et al., 2025; Shang et al., 2026), interactive world models can generate causally consistent environmental responses based on external action sequences (e.g., camera movements and keyboard inputs), enabling bidirectional communication between agents and their environments (Guo et al., 2023; Ding et al., 2025). This capability has demonstrated strong potential across various fields, including game engines (Hafner et al., 2023; Valevski et al., 2024), autonomous driving (Guan et al., 2024), and embodied intelligence (Zhao et al., 2026; Zhang et al., 2025a). Recent studies on urban video understanding (Zhao et al., 2025a) and open-space spatial reasoning (Zhang et al., 2025b) further show that embodied agents require persistent 3D spatial understanding beyond frame-level visual generation.

Interactivity is the core characteristic of current world models, as it directly determines whether a model can realistically simulate dynamic changes in the world and whether its generated content is suitable for agent training (Ding et al., 2025; Bar et al., 2025). However, as shown in Table 1, existing evaluation benchmarks have the following limitations: 1) **Limited diversity in scenes and perspectives.** Existing benchmarks are often derived from single datasets, with scenes and perspectives restricted to pedestrian views (Upadhyay et al., 2026; Ling et al., 2025; Duan et al., 2025). Furthermore, existing datasets with high-quality intrinsic and extrinsic camera parameters are difficult to directly adapt for world model training due to inconsistencies in coordinate systems and parameter formats (Zhou et al., 2018; Sun et al., 2020; Wang et al., 2020). 2) **Lack of a unified definition of action inputs.** Interactive world models adopt heterogeneous action representations, such as text commands (Wu et al., 2025; Wan et al., 2025a), keyboard inputs (He et al., 2025b), or continuous trajectories (Wang et al., 2024; AIGC-Apps & Team, 2024). These representations are not directly aligned with one another, making it difficult to establish fair and consistent comparisons across models. For example, a textual action such as "move forward" may correspond to multiple low-level keyboard or control commands, and directly comparing single-step outputs across different action

---

[1]Tsinghua University, China [2]Northeastern University, China [3]South China University of Technology, China. Correspondence to: Xinlei Chen <chen.xinlei@sz.tsinghua.edu.cn>, Chen Gao <chgao96@gmail.com>.

*Proceedings of the $43^{rd}$ International Conference on Machine Learning*, Seoul, South Korea. PMLR 306, 2026. Copyright 2026 by the author(s).

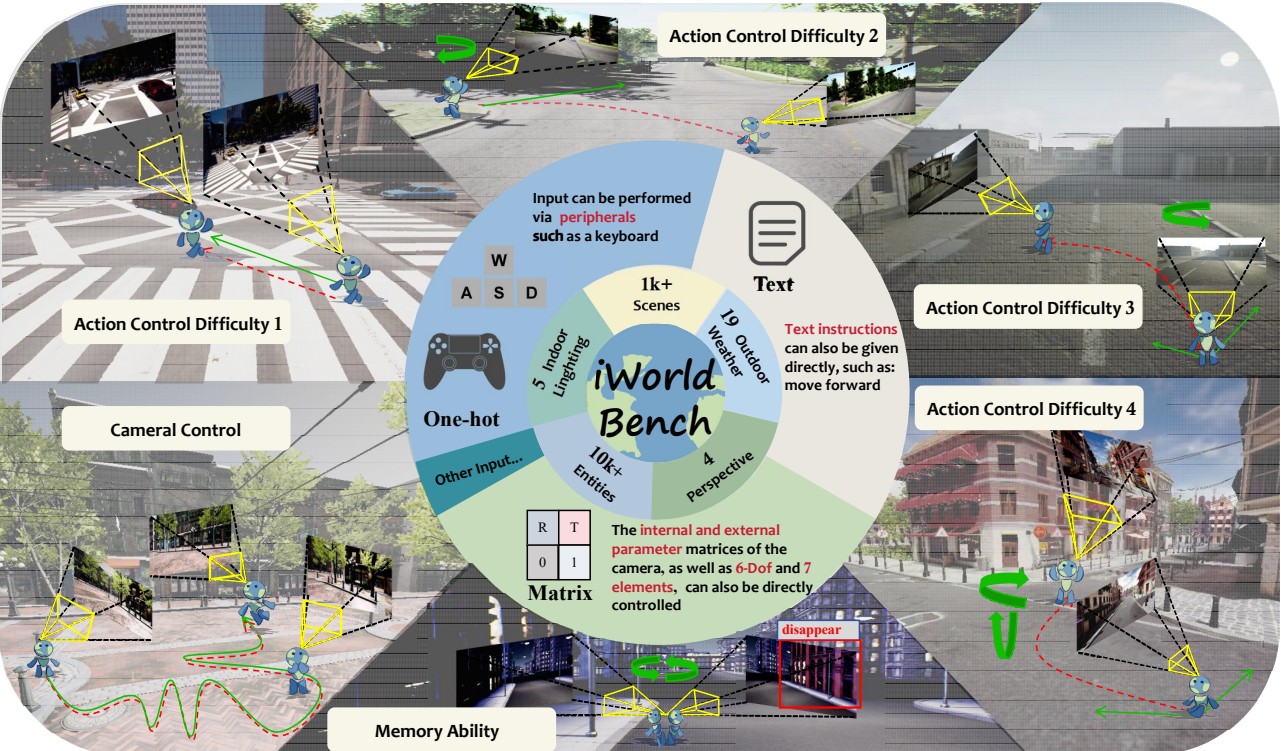

Figure 1. **Overview of iWorld-Bench.** iWorld-Bench encompasses four distinct perspectives: Unmanned Ground Vehicles (UGVs), Unmanned Aerial Vehicles (UAVs), humans, and robotics. It incorporates nine types of outdoor weather conditions, five different indoor lighting conditions, thousand of diverse scenes, and thousands of entities, providing a comprehensive and diverse evaluation environment. The benchmark leverages an **Action Generation Framework** to systematically and uniformly assess the interaction capabilities of interactive world models across various input modalities. It is composed of six tasks, each involving a varying number of trajectories, designed to evaluate the adaptability and performance of models in dynamic and complex scenarios.Visualization of camera trajectory and view control in iWorldBench: → denotes linear control commands for directional movement, --→ represents actual trajectories generated by world models, and ⌒ indicates curved view rotation in the specified direction.

modalities can lead to unfair evaluations. 3) **Insufficient task design for evaluating interactivity.** Current benchmarks are mostly designed for general-purpose world models (Duan et al., 2025; Li et al., 2025a) or embodied world models (Yue et al., 2025; Li et al., 2025c), neglecting the evaluation of interactive world models' responsiveness to external action sequences and interactivity. They also lack tasks of varying difficulty and memory tasks to test the memory capabilities of models (Zhao et al., 2025c). Therefore, there is an urgent need for a dedicated framework to evaluate the interaction capabilities of interactive world models, enabling comprehensive assessment of their performance across diverse scenes and modalities.

To address the aforementioned issues, as shown in Figure 1, we propose **iWorld-Bench**, which has the following three significant advantages: 1) **Diverse World Representations**: We established a comprehensive data processing protocol to clean and standardize 12 high-quality datasets, unifying the original coordinate systems and intrinsic/extrinsic parameter formats to generate high-quality video data suitable

for world models. Additionally, we designed an automated collection and filtering pipeline to gather 100k 1080P video clips from 18 high-quality environments across 4 simulators. Combined with vision-language models (VLMs), all video clips were uniformly annotated, as illustrated in Figure 2. Ultimately, we constructed a high-quality dataset that includes 330k video clips, 4 types of world observation perspectives, 9 types of outdoor weather variations, 5 types of indoor lighting differences, day-night transitions, and thousands of diverse scenes. 2) **Action Generation Framework**: We defined a complete action space dictionary and built a unified action generation framework with modality-agnostic encoding, enabling the unique representation of 81 fundamental motion actions across different modalities. This framework is highly extensible, supporting additional modality encodings and enabling the generation of diverse downstream tasks. 3) **Diverse Interactive Task Design**: Based on the action generation framework, we designed 6 types of task trajectories with varying difficulty levels, including memory capability tests and diverse interactive action tasks, to comprehensively evaluate the interaction

capabilities of world models.

To comprehensively evaluate the interaction capabilities in diverse worlds, we selected a subset of 2,100 videos from a carefully processed dataset of 330,000 high-quality video clips. Tasks were assigned difficulty levels with the assistance of high-quality human annotators, resulting in the design of 4,900 evaluation tasks, including logically consistent memory tasks. We constructed a comprehensive evaluation system, defining 9 evaluation metrics across three dimensions: visual quality, action following, and memory ability, to thoroughly assess the performance of interactive world models. Using this system, we evaluated 14 existing interactive world models, including 5 text-controlled world generation models (based on effective video generation architectures), 2 one-hot encoding-based world generation models, and 7 interactive world models controlled by intrinsic and extrinsic camera parameters. In summary, our contributions can be outlined as follows:

- We constructed a diverse world dataset containing 330,000 high-quality video clips, covering multiple scenes and perspectives, which can be used for training world models. From this dataset, we carefully selected 2,100 videos to build a high-quality subset for evaluating world models.

- We proposed the first benchmark framework specifically designed for interactive world models—**iWorld-Bench**. A general **Action Generation Framework** was defined to unify the evaluation of interaction capabilities across different modalities of world models. Based on this framework, we designed 6 types of tasks, resulting in 4,900 evaluation tasks.

- We introduced 9 evaluation metrics to comprehensively assess 14 existing interactive world models, providing an in-depth analysis of their strengths and limitations, and offering valuable guidance for future research and development of interactive world models.

## 2. Related Work

**Datasets with Camera Parameters.** The development of interactive world models depends on high-quality first-person datasets (Zha et al., 2025) with precise intrinsic and extrinsic camera parameters. These datasets can be grouped into three categories: (1) autonomous driving, robotics, and drone datasets (Zha et al., 2026) (e.g., KITTI (Geiger et al., 2012), NCLT (Carlevaris-Bianco et al., 2016b), TartanAir (Wang et al., 2020)) offering diverse dynamic scenes with accurate annotations; (2) 3D reconstruction datasets (e.g., DL3DV-10K (Ling et al., 2024), Realestate-10K (Zhou et al., 2018),Princeton365 (Kayan et al., 2025b)) featuring varied indoor and outdoor environments with high-quality

camera parameters; and (3) large-scale datasets like SpatialVid (Wang et al., 2025a), tailored for world model training. As shown in Table 2, our benchmark data is curated from these sources to ensure diverse scenes, perspectives, and all-weather conditions.

**World Model Benchmark.** Existing benchmarks for interactive world models primarily focus on evaluating text-to-video generation, with most evaluations based on text control (Upadhyay et al., 2026; Chu et al., 2025; Ling et al., 2025). These benchmarks lack assessments for action sequence generation and do not include specifically designed benchmarks for memory-related tasks. Some benchmarks, such as EWMbench (Yue et al., 2025) and Worldeval (Li et al., 2025c), focus on embodied world generation and are primarily centered on embodied tasks, failing to adequately evaluate the interaction capabilities of interactive world models. While WorldScore (Duan et al., 2025) considers camera control for world models, it is designed for general-purpose world models and lacks the design of interactive tasks.

**Interactive World Model.** Based on interaction methods, existing world models can be categorized into text-controlled interaction, one-hot encoding interaction, and interaction through intrinsic and extrinsic camera parameters. *Text-controlled interaction* (Mao et al., 2025; Alhaija et al., 2025) is primarily based on traditional video generation models, where interaction is achieved through text-based control. While these models can generate worlds corresponding to the given text, they essentially remain video generation models with limited degrees of freedom (Yang et al., 2024; Wu et al., 2025; Wan et al., 2025b). Models such as HY-World 1.5 (Sun et al., 2025) and Matrix-game 2.0 (He et al., 2025b) use key inputs and other encoding methods for one-hot encoding, which expands the degrees of freedom for camera control. However, these models still fail to learn physical laws and cannot perform more flexible actions. In contrast, *interaction through intrinsic and extrinsic camera parameters* (Zheng et al., 2024; Bahmani et al., 2025; Zhu et al., 2025; Li et al., 2025b) offers significantly higher degrees of freedom, enabling the model to follow various complex camera controls (Wang et al., 2024; AIGC-Apps & Team, 2024; He et al., 2025a). **iWorld-Bench** provides a Unified Action Generation Framework, which facilitates the generation of action tasks for different interactive world models, enabling standardized evaluation.

## 3. iWorld-Bench

In this subsection, we provide a comprehensive introduction to our benchmark, **iWorld-Bench**. The design of this benchmark aims to establish a unified evaluation framework that enables a thorough and fair assessment of the interaction capabilities of world models across different modali-

*Table 1.* Comparison of benchmarks for world models. Our benchmark focuses on interactive world modeling with comprehensive capabilities, including multiple inputs, action control, camera control, memory ability, multi-scene, multi-perspective, and all-weather adaptability.

| Benchmark | Field | Multiple Inputs | Interactive Task | Camera Control | Memory Ability | Multi-Scene | Multi-Perspective | All-Weather | Example Num. |
|---|---|---|---|---|---|---|---|---|---|
| EWMBench (Yue et al., 2025) | Manipulation Policies | × | × | × | × | × | × | × | 2,100 |
| Movebench (Chu et al., 2025) | Motion-Controllable Video Generation | × | × | × | × | ✓ | ✓ | × | 1,018 |
| WorldEval (Li et al., 2025c) | Manipulation Policies | × | × | × | × | × | × | × | 1,400 |
| VMbench (Ling et al., 2025) | Motion-Controllable Video Generation | × | × | × | × | ✓ | × | × | 1,050 |
| WorldModelBench (Li et al., 2025a) | General World Model | × | × | × | × | × | × | × | 67,000 |
| WorldBench (Upadhyay et al., 2026) | Motion-Controllable Video Generation | × | × | × | × | × | × | × | 425 |
| WorldScore (Duan et al., 2025) | General World Model | ✓ | × | ✓ | × | ✓ | × | × | 3,000 |
| **iWorld-Bench (Ours)** | Interactive World Model | ✓ | ✓ | ✓ | ✓ | ✓ | ✓ | ✓ | 4,900 |

ties. Specifically, **iWorld-Bench** carefully selected 2,100 high-quality evaluation videos from a dataset of 330,000 high-quality world model video clips and designed 4,900 interactive tasks. The dataset itself covers 5 types of indoor lighting conditions, 9 types of outdoor weather, thousands of environmental scenes, and tens of millions of unique entities. The data processing pipeline is detailed in Section 3.1. Based on task difficulty and the characteristics of world models, we designed six fundamental types of interactive tasks, which are elaborated in Section 3.2. Additionally, we introduced 9 evaluation metrics, as described in Section 3.3.

### 3.1. Dataset Pipline

In this subsection, to construct a multi-scene, multi-perspective, high-quality world model dataset, we introduce a novel data processing pipeline designed to automate the generation of datasets suitable for world models, enabling subsequent benchmark data selection. Specifically, our data processing pipeline consists of three main components: 1) **Video Generate Inherit Past**: We searched for high-quality datasets with intrinsic and extrinsic camera parameters from previous video dataset works. Ultimately, we retrieved 12 high-quality datasets covering various perspectives and diverse scenes, all containing intrinsic and extrinsic parameters. These datasets were standardized in terms of video format, coordinate systems, and data structure. 2) **Video Generate Create Future**: To expand the dataset, we automated the collection of a large amount of high-quality world model data from 18 high-quality scenes across 4 simulators. The collected data was filtered and processed into a unified format. 3) **High-quality Labeling**: All collected data was uniformly labeled to ensure that the final benchmark dataset maximally covers the features of the entire collected dataset. From this, we selected 2,100 high-quality videos for model evaluation. Additionally, with the assistance of high-quality human annotators, we designed memory tasks and assigned task difficulty levels, resulting in 4,900 tasks.

#### 3.1.1. VIDEO GENERATE INHERIT PAST

We conducted a thorough investigation of existing high-quality datasets that inherently include intrinsic and ex-

trinsic camera parameters. Ultimately, we processed 12 datasets, which primarily include: traditional autonomous driving datasets such as KITTI-360 (Geiger et al., 2012), Waymo (Sun et al., 2020), and nuScenes (Caesar et al., 2020); datasets specifically designed with precise intrinsic and extrinsic camera parameters such as RealEstate-10K (Zhou et al., 2018) and Princeton365 (Kayan et al., 2025b); 3D reconstruction datasets such as 7-Scenes (Shotton et al., 2013), DL3DV-10K (Ling et al., 2024), and TUM-RGB-D (Sturm et al., 2012b); robotics datasets such as TartanGround (Patel et al., 2025) and NCLT (Carlevaris-Bianco et al., 2016b); drone datasets such as TartanAir (Wang et al., 2020), CrossLoc (Yan et al., 2022), UAVScenes (Wang et al., 2025b), NTU VIRAL (Nguyen et al., 2022), and MUN-FRL (Thalagala et al., 2024); and, more recently, world model datasets such as SpatialVid (Wang et al., 2025a). As shown in Table 2, we provide detailed information about these datasets.

To harmonize these heterogeneous sources, we re-localized original coordinate systems and unified modality representations into a standardized storage format. Further details regarding the data processing protocols are provided in Appendix A.1.

#### 3.1.2. VIDEO GENERATE CREATE FUTURE

Existing high-quality world model training data is predominantly sourced from indoor scenes, while outdoor datasets are limited in both quantity and mobility rates. To effectively expand the dataset, we selected 18 high-quality environments from four outdoor urban simulators: Aerial_VLN (Liu et al., 2023), UAV_ON (Xiao et al., 2025), OpenFly (Gao et al., 2025), and Embodied_City (Gao et al., 2024). As shown in Figure 2, we manually identified 450 high-quality points within these 18 scenes. Using the 89 action-space tasks defined in Section 3.2, we designed an automated data collection program. Based on the collected data, we developed a filtering pipeline, ultimately generating a high-quality outdoor dataset containing 100,000 videos. Data quality is ensured through a multi-stage, modular filtering pipeline. First, Single-Frame Filtering identifies point anomalies—such as brightness spikes or color muta-

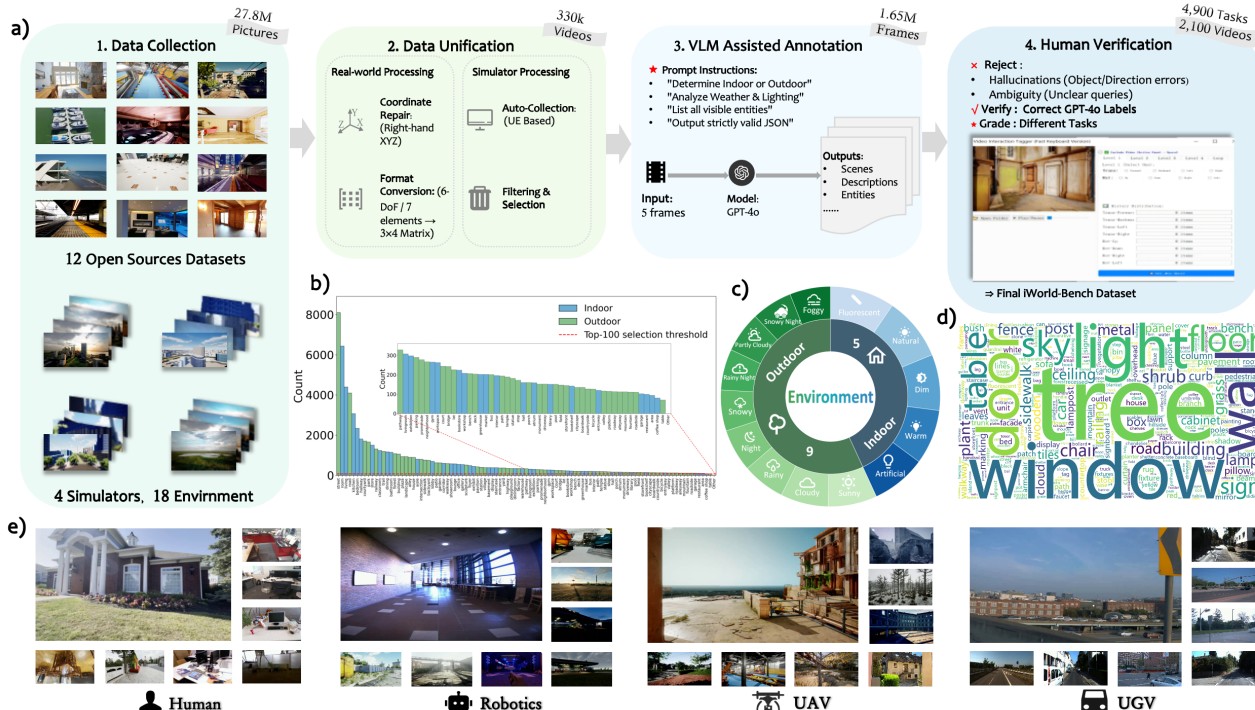

*Figure 2.* **Data Processing Pipeline and Overview.** As shown in Figure 2 a), our data processing pipeline consists of the following four steps: 1) Data Collection: Collecting 27.8M multi-image data from 12 open-source datasets and 18 high-quality simulators. 2) Data Unification: Standardizing the coordinate systems and formats of the data, followed by filtering, resulting in 330K video data. 3) VLM-Assisted Annotation: Using vision-language models (VLMs) to automatically annotate the 330K videos. 4) Human Verification: Selecting 2,100 videos from the dataset and generating 4,900 high-quality tasks through human annotation. Figure 2 b) shows the distribution of the top 100 scenes in the dataset, reflecting the diversity of the scenes. Figure 2 c) illustrates the 9 types of outdoor environments (including foggy, snowy night, partly cloudy, rainy night, snowy, night, rainy, cloudy, and sunny) and 5 types of indoor lighting conditions (fluorescent, natural, dim, warm, and artificial). Figure 2 d) uses a word cloud to demonstrate the complexity of entities in the dataset, covering a wide variety of objects and scene elements. Figure 2 e) presents examples of world observations under four perspectives (drone, autonomous vehicle, pedestrian, and robot), visually showcasing the diversity and high quality of the dataset.

tions—via per-frame visual metrics. Subsequently, Sample Filtering employs statistical density analysis to prune low-quality temporal windows, effectively extracting coherent and high-fidelity sequences. For more details about the datasets and processing steps, please refer to Appendix A.2. Additionally, the detailed filtering design and process can be found in Appendix A.3.

### 3.1.3. HIGH-QUALITY LABELING

To ensure high-precision data annotation, we utilized a vision-language model (VLM) as the primary engine to process a large-scale corpus. A unified annotation scheme was adopted to label each video with semantic attributes such as *environment type*, *scene description*, *weather or lighting*, and *entities*. The specific prompts used are detailed in Appendix A.4. To reduce hallucinated tags and single-model bias, all annotations were further checked by a multi-model verification and human refinement pipeline, as detailed in Appendix A.5. As shown in Figure 2, the dataset encompasses diverse visual and physical characteristics. For subsequent benchmark evaluations, we carefully selected

a representative set of 2,100 high-quality videos from the processed corpus. This selection ensures comprehensive and sufficient coverage across all scene categories, various weather conditions, and most entity types.

### 3.2. iWorld-Bench Design

Our goal is to systematically evaluate the interaction capabilities of different world models. The generation process of a world model can be decoupled into the following representation: $V_{t+1} = W(I_t, C_t)$, where $V_{t+1}$ represents the output of the world model, $W$ denotes the specific world model, $I_t$ represents the current scene frame, and $C_t = [D, T, R, V]$ is a quadruple that controls the current frame $I_t$. Here, $D$ represents the current action difficulty, $T$ represents the translational ID, $R$ represents the rotational ID, and $V$ represents the validity. Detail explanation and a complete description of all motion spaces are provided in Appendix B.

*Table 2.* Summary of datasets with intrinsic and extrinsic camera parameters across various domains. Selected Number $[N_1, N_2]$, where $N_1$ represents the number of video clips selected from the current dataset for action-related tasks, totaling 700, and $N_2$ represents the number of video clips selected from the current dataset for memory-related tasks, totaling 200.

| Dataset | Year | Domain | Pose Representation | Viewpoint | Clip Count | Selected Number |
|---|---|---|---|---|---|---|
| KITTI | 2012 | Autonomous driving | External parameter matrix | UGV | 281 | [20,5] |
| NuScenes | 2019 | Autonomous driving | Seven-element | UGV | 1,000+ | [15,5] |
| Waymo | 2019 | Autonomous driving | External parameter matrix | UGV | 453 | [15,5] |
| TUM-RGB-D | 2011 | 3D reconstruction | Seven-element | Human UGV | 405 | [15,5] |
| 7-Scenes | 2013 | 3D reconstruction | External parameter matrix | Human | 516 | [30,10] |
| RealEstate-10K | 2018 | 3D reconstruction | External parameter matrix | Human UGV | 20,000+ | [200,47] |
| Princeton365 | 2019 | 3D reconstruction | External parameter matrix | Human | 365 | [29,2] |
| DL3DV-10K | 2023 | 3D reconstruction | External parameter matrix | Human | 10,000+ | [60,10] |
| NCLT Dataset | 2016 | Robotics inspection | 6-DoF | UGV | 10,000+ | [60,21] |
| TartanGround | 2024 | Robotics inspection | Seven-element | UGV | 3,000+ | [56,10] |
| TartanAir-V2 | 2024 | Drone inspection | Seven-element | UAV | 2,000+ | [100,40] |
| SpatialVid | 2025 | World model | 6-DoF | Human UGV UAV | 180,000 | [100,40] |
| Total | - | - | - | - | 230,000+ | [700,200] |

### 3.2.1. ACTION GENERATION FRAMEWORK

This is a unified and comprehensive framework. The design and definition of the Action Generation Framework support inputs from any modality of world models, enabling the design of action tasks and guiding the generation of world models. Specifically, it consists of two main components: Interactive Action Encoding and Unified Encoding Mapping.

**Interactive Action Encoding**: We systematically defined the action control space of existing world models. First-person perspective motion can be divided into two modalities: translational motion and rotational motion. Translational motion includes stationary, forward, backward, left, right, upward, and downward movements, totaling 27 actions, denoted as $T_{ID} = [0, 1, 2, \ldots, 26]$. Similarly, rotational motion also includes 27 actions, denoted as $R_{ID} = [0, 1, 2, \ldots, 26]$. The combination of the translational space $T$ and the rotational space $R$ forms the complete motion space, comprising a total of 729 actions. To distinguish the complexity of actions, we defined an action difficulty $D = [1, 2, 3, 4, 5, 6]$, where the difficulty of stationary motion is set to 1. Additionally, based on the training dataset, actions are classified by their validity using

the indicator $V = [0, 1]$, where $V = 1$ represents common actions, and $V = 0$ represents complex and rare actions.

**Unified Encoding Mapping**: To address the differences among existing world models, we performed a unified mapping of the action space. Since some world models do not support upward or downward translational motion in the translational space $T$, or clockwise or counterclockwise camera rotation in the rotational space $R$, the currently available translational and rotational spaces each consist of 9 actions, forming a total of 81 combined actions. During action encoding, we prioritized the design of these 81 actions and constructed a unified mapping dictionary. This dictionary assigns a unique encoding to each action and maps it uniformly to intrinsic and extrinsic camera parameters, one-hot encodings, and text control signals, ensuring compatibility with world models of different modalities. Through this mapping, all actions can achieve consistent control and evaluation across different types of world models.

This approach endows the Action Generation Framework with high robustness, enabling it to encompass more complex actions and support the complete motion space. Additionally, the framework is compatible with world models of various encoding modalities, offering unified definitions and flexible extensibility. Any interactive action encoding modality can be uniquely defined within this framework, and combinations of different actions can generate diverse control signals, facilitating the design of rich downstream tasks.

### 3.2.2. iWORLD-BENCH DESIGN

Based on the definition of the **Interactive World Framework**, as well as a curated selection of 2,100 videos, 4,200 meticulously annotated tasks, and 700 camera parameter files (partially showcased in Appendix C), we designed the following six types of tasks to comprehensively evaluate the interaction capabilities of world models:

- **Action Control Difficulty 1**: Tests the model's action-following ability on basic tasks (difficulty $D = 1$), including 9 basic actions such as stationary. A total of 1,000 tasks were designed.

- **Action Control Difficulty 2**: Tests the model's action-following ability on two-degree-of-freedom tasks (difficulty $D = 2$), covering 24 actions. A total of 1,000 tasks were designed.

- **Action Control Difficulty 3**: Tests the model's action-following ability on three-degree-of-freedom tasks (difficulty $D = 3$), covering 32 actions. A total of 1,000 tasks were designed.

- **Action Control Difficulty 4**: Tests the model's action-following ability on four-degree-of-freedom complex

tasks (difficulty $D = 4$), covering 16 actions. A total of 1,000 tasks were designed.

- **Memory Ability**: Tests the model's memory capability by constructing cyclic paths, requiring the model to visit the same location during a single inference. Consistency of generation and trajectory alignments are evaluated. A total of 200 tasks were designed.

- **Camera Following**: Specifically designed for world models controlled by intrinsic and extrinsic camera parameters, this task uses 700 camera parameter files to test the model's ability to follow trajectories.

### 3.3. Evaluation Metrics

#### 3.3.1. GENERATION QUALITY

**Image Quality.** We evaluate low-level visual distortions by calculating the normalized average MUSIQ (Ke et al., 2021) score across all frames to reflect fundamental rendering fidelity.

**Brightness Consistency.** It is designed to quantify the temporal consistency of brightness distribution between each frame and the initial frame in video sequences, addressing the issue of insufficient visual stability caused by inter-frame brightness shifts and abrupt changes.

**Color Temperature Constraint.** By penalizing hue drifting in the HSV space through weighted temporal similarity, this metric ensures global color temperature remains logically unified for consistent scene coherence.

**Sharpness Retention.** By coupling a vectorized Tenengrad method with a BRISQUE-triggered "circuit breaker" logic (Mittal et al., 2012), we distinguish genuine texture stability from high-frequency artifacts. This mechanism effectively rewards edge clarity while penalizing systemic visual collapses and noise-induced distortions.

#### 3.3.2. TRAJECTORY FOLLOWING

**Motion Smoothness.** We utilize motion priors from frame interpolation models to evaluate the reconstruction quality of sampled frames via LPIPS, SSIM, and MSE, effectively identifying unnatural jitters or physical inconsistencies.

**Trajectory Accuracy.** By evaluating directional mapping accuracy in the motion tangent space via ViPE-extracted trajectories (Huang et al., 2025), we quantify how precisely the model adheres to camera commands. This coordinate-aligned measurement provides a fine-grained scale for assessing the instruction-level controllability of world models.

**Trajectory Tolerance.** To eliminate estimator-induced variance, we calibrate the model's trajectory fidelity by cross-referencing generated sequences with ground-truth videos under the same ViPE framework. This comparative paradigm cancels out estimation uncertainties, providing a refined measurement of physical execution fidelity under ideal constraint conditions.conditions.

#### 3.3.3. MEMORY ABILITY

**Memory Symmetry.** To quantify logical loop-closure in reciprocal tasks, we evaluate the pixel-wise consistency of symmetric frame pairs relative to the temporal midpoint. This metric effectively captures memory decay or structural logic failure during extended temporal reasoning.

**Trajectory alignment.** By calculating the mirror similarity of instantaneous displacement vectors, we assess the spatial topological consistency of camera movements in reciprocal tasks. This dimension evaluates the model's 3D space persistence, ensuring geometric logic remains intact during "return-trip" scenarios.

## 4. Experiments

### 4.1. Setup

We selected 14 representative interactive world models for evaluation, including five text-conditioned camera control models: NVIDIA Cosmos-predict2.5 (Alhaija et al., 2025) , HunyuanVideo-1.5 (Wu et al., 2025), WAN 2.2 (Wan et al., 2025b), CogVideoX-5B-I2V (Yang et al., 2024), and YUME 1.5 (Mao et al., 2025); two one-hot–conditioned camera control models: Matrix-Game 2.0 (He et al., 2025b) and HY-World 1.5 (Sun et al., 2025); and seven models with camera control via explicit intrinsics and extrinsics: CameraCtrl (He et al., 2025a), MotionCtrl (Wang et al., 2024), CamI2V (Zheng et al., 2024), RealCam-I2V (Li et al., 2025b), VideoX-Fun-WAN (AIGC-Apps & Team, 2024), AC3D (Bahmani et al., 2025), and ASTRA (Zhu et al., 2025). All model inferences were conducted on NVIDIA A800 GPUs.

### 4.2. Results

#### 4.2.1. HUMAN PREFERENCE VALIDATION

We validate iWorld-Bench's metrics via a human preference study and results confirm they align well with human subjective perceptions of world model performance and are robust to variations in video resolution and aspect ratio.

#### 4.2.2. ACTION CONTROL AND MEMORY ABILITY

Table 3 summarizes the evaluation results on Action Control and Memory Ability tasks across three control paradigms. Overall, one-hot encoding models demonstrate superior interaction capabilities but lack the flexibility offered by camera-parameter control. Text-controlled models excel in generation quality but lag in trajectory following.

*Table 3.* Comparison of metric scores for different models and performance on Action Control and Memory Ability tasks.

| Method | Rank | Avg.↑ | Generation Quality | | | | Trajectory Following | | Memory Ability | |
| | | | Image Quality ↑ | Brightness Consistency ↑ | Color Temperature Constraint ↑ | Sharpness Retention ↑ | Motion Smoothness ↑ | Trajectory Accuracy ↑ | Memory Symmetry ↑ | Trajectory Alignment ↑ |
|---|---|---|---|---|---|---|---|---|---|---|
| *Camera Control via Text* | | | | | | | | | | |
| NVIDIA Cosmos | 7 | 0.6275 | 0.6778 | 0.6952 | 0.7170 | 0.4363 | 0.9907 | 0.4955 | 0.3738 | 0.6419 |
| HunyuanVideo-1.5 | 3 | 0.7188 | **0.7128** | 0.7027 | 0.7477 | 0.5545 | 0.9908 | 0.6844 | 0.6336 | 0.6449 |
| WAN 2.2 | 12 | 0.5731 | 0.5545 | 0.3886 | 0.3411 | 0.3428 | 0.9557 | 0.6514 | 0.4480 | 0.5703 |
| CogVideoX-I2V | 5 | 0.6963 | 0.6521 | **0.8988** | **0.8129** | **0.7951** | **0.9938** | 0.5950 | 0.6010 | 0.4084 |
| YUME 1.5 | 8 | 0.6209 | 0.6232 | 0.3810 | 0.4165 | 0.4023 | 0.9765 | 0.7113 | 0.5276 | 0.5988 |
| *Camera Control via One-hot Encoding* | | | | | | | | | | |
| Matrix-game 2.0 | 13 | 0.5663 | 0.4851 | 0.2963 | 0.2937 | 0.4149 | 0.9848 | 0.7008 | 0.3311 | 0.6362 |
| HY-World 1.5 | 1 | **0.7873** | 0.6675 | 0.8051 | 0.7819 | 0.6634 | 0.9921 | **0.7472** | 0.8481 | 0.6776 |
| *Camera Control via Intrinsics and Extrinsics* | | | | | | | | | | |
| CameraCtrl | 11 | 0.5762 | 0.4473 | 0.3717 | 0.2511 | 0.4545 | 0.9796 | 0.6778 | 0.4279 | 0.6097 |
| MotionCtrl | 14 | 0.5486 | 0.4562 | 0.3980 | 0.2012 | 0.4294 | 0.9735 | 0.6730 | 0.3098 | 0.5932 |
| CamI2V | 10 | 0.5765 | 0.5284 | 0.4343 | 0.3568 | 0.4297 | 0.9861 | 0.6314 | 0.3631 | 0.6038 |
| RealCam-I2V | 6 | 0.6865 | 0.6227 | 0.4130 | 0.5547 | 0.6269 | 0.9860 | 0.5630 | 0.7948 | 0.6668 |
| videox-fun-Wan | 2 | 0.7474 | 0.6410 | 0.5972 | 0.5473 | 0.5998 | 0.9858 | 0.7172 | 0.9009 | **0.6876** |
| AC3D | 4 | 0.7149 | 0.4573 | 0.7307 | 0.6524 | 0.5332 | 0.9919 | 0.5785 | **0.9068** | 0.6250 |
| ASTRA | 9 | 0.5980 | 0.5335 | 0.5091 | 0.4338 | 0.5488 | 0.9799 | 0.6115 | 0.4323 | 0.5518 |

*Table 4.* Comparison of metric scores for different models and performance on Camera Following tasks.

| Method | Generation Quality | | | | Trajectory Following | |
| | Image Quality ↑ | Brightness Consistency ↑ | Color Temperature Constraint ↑ | Sharpness Retention ↑ | Motion Smoothness ↑ | Trajectory Tolerance ↑ |
|---|---|---|---|---|---|---|
| *Camera Control via Intrinsics and Extrinsics* | | | | | | |
| CameraCtrl | 0.3980 | 0.3497 | 0.2008 | 0.4211 | 0.9659 | 0.7099 |
| MotionCtrl | 0.4270 | 0.3924 | 0.1810 | 0.4256 | 0.9622 | 0.7120 |
| CamI2V | 0.4046 | 0.3674 | 0.2605 | 0.3830 | 0.9766 | 0.7143 |
| RealCam-I2V | **0.5889** | 0.4777 | 0.5521 | **0.6838** | 0.9783 | 0.7480 |
| videox-fun-Wan | 0.5701 | 0.5584 | 0.3659 | 0.4925 | 0.9604 | 0.7381 |
| AC3D | 0.5208 | **0.8927** | **0.7404** | 0.6472 | **0.9919** | **0.9091** |
| ASTRA | 0.4743 | 0.3972 | 0.2819 | 0.4171 | 0.9615 | 0.4286 |

Among all models, HY-World 1.5 ranks first with an average score of 0.7873, excelling in both memory ability and trajectory following. Compared to text-controlled models such as CogVideoX-I2V, which achieves a trajectory accuracy of only 0.5950, the substantially higher accuracy of HY-World 1.5 (0.7472) demonstrates the advantage of discrete action signals over continuous text descriptions. In the camera-parameter category, videox-fun-Wan and AC3D show strong overall performance, particularly in memory tasks, whereas early methods like MotionCtrl and CameraCtrl exhibit limited effectiveness across all metrics. Text-controlled models such as CogVideoX-I2V achieve the highest visual consistency (Brightness Consistency: 0.8988) but sacrifice trajectory accuracy, revealing a fundamental trade-off between visual quality and action controllability. As visualized in Figure 3(a), HY-World 1.5 and videox-fun-Wan exhibit the most balanced performance profiles across all metrics, while text-controlled models show strong generation quality but weaker trajectory following.

An interesting observation emerges when comparing camera-controlled models with their base models. For instance, AC3D is fine-tuned from CogVideoX-I2V, and HY-World 1.5 is derived from HunyuanVideo-1.5. While these specialized models achieve significantly improved trajectory following capabilities, they exhibit slight degradation in generation quality metrics compared to their base models. This indicates that fine-tuning on camera-control-specific datasets enhances controllability at the cost of visual fidelity. This trade-off warrants careful consideration in future training strategies.

### 4.2.3. CAMERA FOLLOWING

Table 4 reports fine-grained trajectory following results for camera-parameter-controlled models using 700 annotated camera files. AC3D achieves the best overall performance, with the highest Trajectory Tolerance (0.9091), Brightness Consistency (0.8927), and Motion Smoothness (0.9919), demonstrating that its architecture effectively leverages explicit camera parameters for precise control.

A significant performance gap exists in trajectory tolerance, ranging from 0.4286 (ASTRA) to 0.9091 (AC3D), indicating that current approaches vary considerably in translating camera parameters into coherent visual sequences. Notably, RealCam-I2V achieves the highest Image Quality (0.5889)

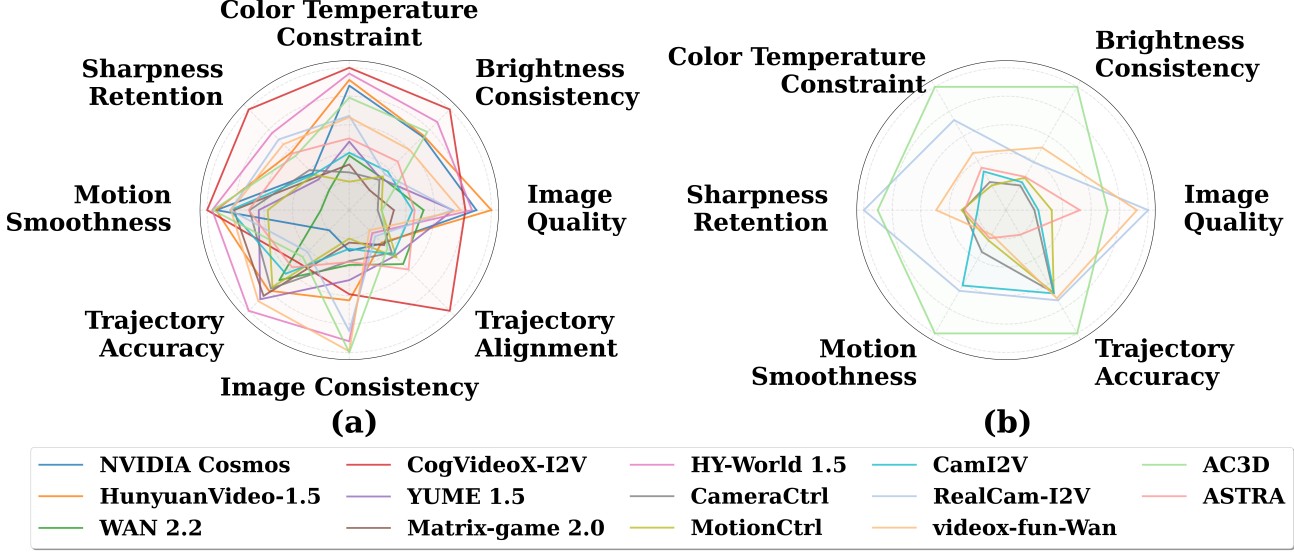

Figure 3. **Radar charts of model performance across evaluation metrics.** (a) Performance comparison of all 14 models on Action Control and Memory Ability tasks across 8 metrics. (b) Performance of 7 camera-parameter-controlled models on Camera Following tasks.

yet falls behind in Trajectory Tolerance (0.7480), reinforcing that visual fidelity and action controllability represent orthogonal evaluation dimensions. Early camera control methods (CameraCtrl, MotionCtrl) show limited effectiveness compared to recent approaches, highlighting substantial architectural advances in the field. As shown in Figure 3(b), AC3D exhibits coverage across the radar chart, while early methods cluster near the center with limited reach.

## 5. Conclusion and Future Work

iWorld-Bench is a unified benchmark specifically designed for interactive world models, integrating a multi-dimensional evaluation framework. It systematically evaluates model performance across diverse worlds through six types of interactive tasks and reveals the limitations of current models in terms of interaction capability. We conducted extensive evaluations on 14 interactive world models and their base models, analyzing the differences and shortcomings of various paradigms in interaction capability and highlighting key trade-offs between generation quality and controllability. In addition, we proposed a comprehensive data processing pipeline and constructed a high-quality, multi-scene, multi-perspective dataset containing 330,000 videos, providing a solid data foundation for world model research. We hope that iWorld-Bench can serve as a standard benchmark for evaluating interactive world models and further advance the development of this field. Given that iWorld-Bench covers UAV perspectives and trajectory-level interaction tasks, future extensions can further examine whether interactive world models support downstream aerial decision-making scenarios (Chen et al., 2024; Wang

et al., 2026), such as multi-UAV wildfire suppression and emergency network scheduling (Seraj et al., 2022; Xu et al., 2024), where agents must reason over long horizons while reacting to dynamic environments. In the future, we plan to further improve the benchmark with evaluations of real-time performance and long-horizon consistency, with the goal of establishing a more influential and comprehensive benchmark for the community.

## Acknowledgments

This paper was supported by the Natural Science Foundation of China under Grant 62371269, Shenzhen Science and Technology Program ZDCY202517012 and Shenzhen Low-Altitude Airspace Strategic Program Portfolio (Grant No. Z25306110). This work is also supported in part by Tsinghua University-Toyota Research Center.

## Impact Statement

This paper presents work whose goal is to advance the field of machine learning. There are many potential societal consequences of our work, none of which we feel must be specifically highlighted here.

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

# A. Dataset details

## A.1. Open Sources Datasets Details

**Realestate-10K.** RealEstate10K is a large dataset of camera poses corresponding to 10 million frames derived from about 80,000 video clips, gathered from about 10,000 YouTube videos. For each clip, the poses form a trajectory where each pose specifies the camera position and orientation along the trajectory. These poses are derived by running SLAM and bundle adjustment algorithms on a large set of videos.

**KITTI.** KITTI (Geiger et al., 2013) is captured by driving around the mid-size city of Karlsruhe, in rural areas and on highways. Up to 15 cars and 30 pedestrians are visible per image. This dataset could be used in stereo, optical flow, visual odometry, 3D object detection and 3D tracking.

**TUM-RGB-D.** TUM-RGB-D (Sturm et al., 2012a) contains the color and depth images of a Microsoft Kinect sensor along the ground-truth trajectory of the sensor. The data was recorded at full frame rate (30 Hz) and sensor resolution (640x480). The ground-truth trajectory was obtained from a high-accuracy motion-capture system with eight high-speed tracking cameras (100 Hz). This dataset can be used on the evaluation of visual odometry and visual SLAM systems.

**TartanGround.** TartanGround (Patel et al., 2025) is a large-scale, multi-modal dataset to advance the perception and autonomy of ground robots operating in diverse environments. This dataset, collected in various photorealistic simulation environments includes multiple RGB stereo cameras for 360-degree coverage, along with depth, optical flow, stereo disparity, LiDAR point clouds, ground truth poses, semantic segmented images, and occupancy maps with semantic labels. Data is collected using an integrated automatic pipeline, which generates trajectories mimicking the motion patterns of various ground robot platforms, including wheeled and legged robots. Dataset collects 878 trajectories across 63 environments, resulting in 1.44 million samples.

**DL3DV-10K.** DL3DV-10K (Ling et al., 2024)is a large-scale real-world dataset containing over 10,000 high-quality videos. Each video is manually annotated with key scene points and complexity, and also provides camera pose, NeRF (Mildenhall et al., 2021) depth estimation, point clouds, and 3D meshes. The dataset can be used for general NeRF research, scene consistency tracking, visual language models, and other computer vision studies.

**Waymo.** Waymo (Sun et al., 2020) is a specialized dataset designed for motion prediction tasks in autonomous driving. It integrates extensive multi-modal motion data from real-world driving scenarios, along with detailed maps and sensor information. By offering rich annotations and diverse data scenarios, this dataset provides a solid data foundation for motion prediction, behavior modeling, and related tasks in the field of autonomous driving.

**Princeton365.** Princeton365 is a large-scale diverse dataset of 365 videos with accurate camera pose. The dataset collects indoor, outdoor, and object scanning videos with synchronized monocular and stereo RGB video outputs as well as IMU. It also proposes a new scene scale-aware evaluation metric for SLAM based on the the optical flow induced by the camera pose estimation error.

**7-sences.** 7-sences (Kayan et al., 2025a) is a collection of tracked RGB-D camera frames. This dataset can be used to evaluate methods across various applications, such as dense tracking and mapping, as well as relocalization techniques. All scenes were recorded with a handheld Kinect RGB-D camera at a resolution of 640×480. The dataset creators utilized an implementation of the KinectFusion system to obtain ground-truth camera trajectories and dense 3D models. Each scene comprises several sequences captured by different users and is divided into separate sets of training and testing sequences.

**TartanAir-V2.** TartanAir-V2 is a large-scale synthetic dataset for visual SLAM. With a data volume exceeding 3TB, it contains over 1 million frames. The dataset offers more than 30 types of environmental scenes, categorized into two motion difficulty levels: Easy and Hard. It encompasses six types of data, including stereo RGB images, depth maps, semantic segmentation maps, optical flow fields, camera poses, and LiDAR point clouds. The dataset supports evaluation for both monocular and stereo vision and includes complete evaluation tools as well as ground truth trajectories.

**NCLT.** NCLT (Carlevaris-Bianco et al., 2016a) dataset records comprehensive sensor data from the campus of the University of Michigan North Campus and its surrounding areas. The dataset exhibits significant temporal variation, encompassing diverse weather conditions and illumination changes from early morning to dusk. The environment features a variety of dynamic elements, including pedestrian activity, vehicular traffic (campus buses, cars, bicycles), construction zones, and constantly changing vehicle arrangements in parking lots. This vision and LIDAR dataset could be used in 3D Reconstruction field.

**SpatailVid.** SpatialVID (Wang et al., 2025a) is a large-scale video dataset incorporating spatial annotations, designed for tasks such as text-to-video, text-to-3D, image-to-3D, image-to-video, and others. With a size of approximately 10TB, the dataset is organized into 545 groups for ease of management. Each group contains roughly 14 GB of video data and 1.5 GB of annotation data. The dataset includes metadata for each video clip, which can be filtered and analyzed using pandas. The annotations encompass captions, dynamic masks, camera intrinsics, camera poses, and motion instructions.

**NuSence.** The NuScenes dataset is a large-scale autonomous driving dataset featuring comprehensive 3D object annotations. It was collected from two cities, Boston and Singapore, covering both left-hand and right-hand traffic scenarios. The dataset consists of 1,000 continuous driving segments, each lasting 20 seconds. With a total data volume exceeding 1.4 TB, it includes 1.4 million camera images and 390,000 LiDAR scans. The dataset provides a complete sensor suite along with detailed high-definition map information. Its core annotations comprise 1.4 million manually labeled 3D bounding boxes and 11 billion labeled LiDAR point clouds, while also recording object attributes such as visibility, activity status, and pose.

## A.2. Simulator Details

In this study, we examined four simulators and their corresponding environments: `aerial_VLN`, `UAV_ON`, `Openfly`, and `Embodied_City`. Each simulator contains multiple environments, and we selected high-quality environments based on their representativeness and quality for further research.

- **aerial_VLN** (Liu et al., 2023): This simulator includes 22 environments. We carefully selected 8 high-quality environments as the core dataset for our experiments.

- **UAV_ON** (Xiao et al., 2025): This simulator has 15 environments. We selected 5 high-quality and representative environments for detailed analysis.

- **Openfly** (Gao et al., 2025): This simulator contains 4 environments. Due to the small number of environments, we chose all of them to ensure comprehensive data coverage.

- **Embodied_City** (Gao et al., 2024): This simulator only includes 1 environment, and we selected it for inclusion in the experiment.

Through careful selection of these simulators and environments, we ensured that the environments involved in the experiment are representative and of high quality, providing a reliable foundation for subsequent analysis and experimental results.

*Table 5.* Scene maps and sampling coordinates at three altitude levels. Points are ordered by sampling index: the first third are low-altitude, the middle third are mid-altitude, and the final third are high-altitude.

| Scene Name | Scene Map | Low-altitude Points | Mid-altitude Points | High-altitude Points |
|---|---|---|---|---|
| **UAV_ON_ENV3** |  | (6.128, 3.793, -1.000) (15.801, 38.299, -1.000) (-18.850, 28.706, -1.000) (-25.089, 5.967, -1.000) | (-26.272, 15.784, -4.300) (-2.570, 5.718, -4.300) (15.545, 13.003, -4.300) (2.839, 5.687, -4.300) | (-7.654, 6.988, -10.000) (21.699, 36.950, -10.000) (-20.670, 43.517, -20.000) (15.447, 11.867, -20.000) |
| **UAV_ON_ENV4** |  | (8.716, 15.849, -2.600) (37.748, 88.478, -2.600) (-23.843, 91.871, -2.600) (-64.359, 19.372, -2.600) | (-84.403, 7.045, -8.400) (-16.554, -40.680, -8.400) (71.827, -57.095, -8.400) (46.966, -25.692, -8.400) | (51.230, 88.299, -15.000) (-28.491, 61.143, -15.000) (109.670, -71.236, -25.000) (-76.350, 27.255, -25.000) |
| **UAV_ON_ENV6** |  | (-22.442, -72.798, 4.000) (-193.283, -71.143, 4.000) (-136.822, -69.932, 4.000) (-63.675, -147.873, 4.000) | (-63.089, -74.590, -2.300) (-192.890, -68.654, -2.300) (-153.451, -200.782, -2.300) (-62.568, -198.329, -2.300) | (-116.327, -70.731, -6.000) (-165.262, -203.039, -6.000) (-6.542, -205.697, -14.000) (-9.939, -72.644, -14.000) |

| Scene Name | Scene Map | Low-altitude Points | Mid-altitude Points | High-altitude Points |
|---|---|---|---|---|
| UAV_ON_ENV7 |  | (37.979, 13.984, -1.800)
(1.736, 12.045, -1.800)
(-30.320, -17.451, -1.800)
(-28.697, -81.708, -1.800) | (-35.108, -79.722, -5.000)
(48.905, -82.563, -5.000)
(77.235, -78.398, -5.000)
(-30.230, -26.098, -5.000) | (-11.101, -153.493, -8.500)
(-57.210, -78.937, -8.500)
(32.256, -82.409, -14.000)
(-47.865, -79.469, -14.000) |
| UAV_ON_ENV8 |  | (18.264, -5.685, -4.000)
(0.192, -9.546, -3.000)
(14.710, -0.772, -4.000)
(18.054, -19.357, -4.179) | (15.451, -5.606, -6.258)
(40.793, -0.366, -4.683)
(-11.779, -10.805, -3.456)
(-36.910, -14.313, -2.013) | (-29.561, -5.932, -6.013)
(-22.331, -54.815, -5.180)
(-38.579, -17.058, -9.104)
(-36.509, -8.328, -7.633) |
| aerial_VLN_ENV9 |  | (-91.626, 35.289, 18.790)
(-190.175, 36.170, 19.000)
(-190.538, 139.300, 19.000)
(-155.324, 38.899, 19.000) | (-78.413, 39.033, 14.000)
(-140.166, 37.936, 14.000)
(-189.142, 0.570, 14.000)
(-192.911, 124.054, 14.000) | (-156.697, 101.423, 11.000)
(-102.769, -20.182, 11.000)
(-110.043, 65.631, 1.000)
(-203.003, 119.275, 1.000) |
| aerial_VLN_ENV10 |  | (11.793, 2.102, -1.300)
(54.984, 2.333, -1.300)
(55.469, 30.166, -1.300)
(48.182, 61.720, -1.300) | (31.364, 59.734, -6.000)
(-67.950, 39.215, -6.000)
(-44.708, 30.488, -6.000)
(-34.287, 95.669, -6.000) | (41.643, 80.889, -13.000)
(60.604, -0.578, -13.000)
(57.875, -35.892, -20.000)
(56.287, 56.520, -20.000) |
| aerial_VLN_ENV11 |  | (45.235, 55.911, -1.300)
(27.488, 55.291, -1.300)
(-0.310, 70.299, -1.300)
(-56.052, 76.477, -1.300)
(-97.131, 76.047, -1.300)
(-90.822, 25.716, -1.300)
(-94.564, -27.929, -1.300)
(-92.500, -84.088, -1.300)
(-59.766, -81.448, -1.300)
(-55.119, -119.142, -1.300)
(2.520, -110.351, -1.300)
(6.552, -153.028, -1.300) | (42.391, -152.923, -8.000)
(48.397, -151.875, -8.000)
(47.686, -194.987, -8.000)
(-28.585, -190.693, -8.000)
(-33.865, -234.361, -8.000)
(-29.857, -290.922, -8.000)
(67.817, -295.840, -8.000)
(74.385, -229.124, -8.000)
(104.008, -151.863, -8.000)
(32.370, -195.518, -8.000)
(-43.617, -193.723, -8.000)
(-102.995, -183.336, -8.000) | (6.049, -200.551, -20.000)
(46.628, -192.362, -20.000)
(42.671, -156.086, -20.000)
(69.831, -143.099, -20.000)
(3.741, -54.421, -20.000)
(43.468, 13.521, -20.000)
(-54.745, -0.133, -35.500)
(-99.412, -68.246, -35.500)
(48.517, -43.437, -35.500)
(68.403, -138.615, -35.500)
(71.259, -220.091, -35.500)
(42.917, -214.620, -35.500) |
| aerial_VLN_ENV12 |  | (-37.441, 90.557, -2.700)
(-94.581, 92.097, -2.700)
(-100.108, 127.987, -2.700)
(-66.235, 135.011, -2.700) | (-25.425, 135.177, -5.400)
(1.064, 152.387, -5.400)
(35.971, 89.486, -5.400)
(-10.684, 49.802, -5.400) | (-10.105, 90.295, -7.600)
(-89.618, 58.294, -7.600)
(-98.266, 138.670, -12.500)
(-2.286, 88.559, -12.500) |
| aerial_VLN_ENV13 |  | (6.576, -30.470, -1.400)
(30.722, -28.208, -1.400)
(27.859, -79.975, -1.400)
(28.383, -54.011, -1.400) | (17.174, -29.063, -6.000)
(-13.253, -29.194, -6.000)
(-14.979, -78.660, -6.000)
(29.385, -81.904, -6.000) | (-11.739, -55.504, -9.600)
(-3.800, -80.401, -9.600)
(36.010, -28.062, -14.700)
(-11.056, -39.497, -14.700) |
| aerial_VLN_ENV14 |  | (30.565, 10.610, -2.300)
(-6.211, 8.254, -2.300)
(-19.429, 21.130, -2.300)
(-18.832, 69.564, -2.300) | (-12.025, 9.556, -6.500)
(26.898, 8.891, -6.500)
(3.135, 27.596, -6.500)
(-14.117, 67.712, -6.500) | (-7.684, 11.805, -18.000)
(39.650, 64.258, -18.000)
(-13.431, 74.214, -28.000)
(-31.034, 26.319, -28.000) |

| Scene Name | Scene Map | Low-altitude Points | Mid-altitude Points | High-altitude Points |
|---|---|---|---|---|
| **Openfly**_ENV16 | | (1.608, 5.258, -1.800)
(29.834, 133.353, -1.800)
(82.088, 128.970, -1.800)
(159.274, 125.609, -1.800)
(224.114, 123.309, -1.800)
(-52.923, 132.015, -1.800)
(-93.901, 275.997, -1.800)
(-30.730, 479.773, -1.800)
(96.107, 695.252, -1.800)
(-33.149, 507.116, -1.800)
(-141.161, 483.803, -1.800)
(-827.664, 493.460, -1.800) | (-1038.693, 497.148, -9.555)
(-1113.814, 327.928, -9.500)
(-1089.126, 307.862, -9.500)
(-1047.304, 305.670, -9.500)
(-1046.536, 212.556, -9.500)
(-930.675, 206.340, -9.500)
(-907.500, 59.028, -9.500)
(-1006.916, 244.430, -9.500)
(-27.092, 378.632, -9.500)
(-5.033, 771.811, -9.500)
(91.237, 728.410, -9.500)
(259.049, 478.681, -9.500) | (15.028, 287.368, -25.000)
(-29.599, 529.443, -25.000)
(172.125, 786.667, -25.000)
(82.636, 174.686, -25.000)
(-18.017, 494.930, -25.000)
(6.929, 120.294, -25.000)
(-97.236, 370.495, -45.000)
(36.778, 469.593, -45.000)
(254.723, 611.107, -45.000)
(-22.458, 369.423, -45.000)
(-971.804, 488.350, -45.000)
(-1094.522, 258.790, -45.000) |
| **Openfly**_ENV18 | | (227.153, -27.180, 45.400)
(412.628, -23.343, 45.400)
(668.505, -9.502, 45.400)
(665.249, -287.738, 45.000)
(640.051, -330.825, 44.093)
(657.060, -486.326, 44.205)
(657.524, -597.022, 44.314)
(927.707, -576.267, 42.719)
(1013.134, -592.300, 42.719)
(1357.697, -695.550, 42.754)
(1371.944, -481.993, 41.714)
(1350.897, -226.589, 41.715) | (1488.110, -241.337, 18.428)
(1384.244, -222.607, 13.541)
(1352.811, 28.125, 16.332)
(1366.548, -107.876, 16.333)
(1294.857, -246.484, 16.334)
(1080.612, -212.575, 16.335)
(1015.504, -10.065, 20.385)
(926.253, 123.051, 22.591)
(490.769, -30.297, 30.236)
(562.973, -22.301, 19.857)
(665.584, -84.334, 17.400)
(700.391, -230.819, 14.626) | (959.067, -230.513, -63.235)
(1379.624, -189.139, -65.727)
(1164.886, -284.344, -65.726)
(1110.427, -331.984, -65.725)
(1379.458, 156.838, -67.228)
(1357.443, -580.837, -79.361)
(1386.540, 113.418, -158.576)
(1461.480, -86.156, -161.780)
(1140.107, -270.806, -156.295)
(1145.839, -556.763, -155.810)
(1409.625, -585.073, -146.769)
(1704.326, -624.882, -146.769) |
| **aerial_VLN**_ENV21 | | (0.000, 10.207, -0.000)
(-2.343, -17.940, 0.156)
(-0.453, -79.805, -0.357)
(28.721, -70.401, -0.357) | (13.532, -31.747, -2.649)
(-2.362, -31.263, -3.295)
(-4.686, 11.679, -4.292)
(0.406, 72.552, -3.601) | (-1.461, 38.794, -8.569)
(-32.978, -48.198, -10.767)
(-6.377, -67.191, -17.526)
(-33.241, 86.204, -15.721) |
| **Openfly**_ENV23 | | (-0.695, -1.600, -0.000)
(14.574, 24.399, -0.000)
(30.871, 6.555, -0.000)
(-31.542, -27.665, 0.114) | (8.642, -27.146, -7.929)
(12.624, -64.019, -6.601)
(-9.235, -110.649, -5.191)
(-9.241, 58.938, -10.211) | (5.555, 58.516, -24.798)
(-2.040, -98.520, -24.105)
(-81.832, -8.559, -49.823)
(72.293, -1.800, -50.216) |
| **aerial_VLN**_ENV24 | | (178.929, -6.939, -1.509)
(213.362, -10.063, -1.337)
(194.355, -72.791, -1.861)
(192.122, -7.992, -1.591) | (159.579, -6.974, -6.383)
(192.390, 5.251, -7.346)
(216.476, -7.963, -7.313)
(192.129, -45.342, -7.801) | (164.211, -7.587, -13.512)
(190.906, -57.764, -13.031)
(161.063, -9.265, -21.202)
(192.004, 21.248, -20.456) |
| **Openfly**_ENV26 | | (56.792, 1612.142, 319.724)
(172.345, 1608.101, 319.671)
(210.733, 1563.752, 317.088)
(218.712, 1472.690, 321.085)
(382.039, 1474.189, 318.487)
(457.813, 1411.601, 320.918)
(474.287, 1459.438, 320.918)
(650.094, 1410.407, 320.372)
(712.532, 1446.226, 319.157)
(715.294, 1525.223, 319.943)
(748.302, 1599.735, 319.943)
(739.684, 1667.046, 319.943) | (814.153, 1279.290, 313.388)
(652.131, 1279.303, 310.028)
(574.232, 1266.361, 307.777)
(609.408, 1600.208, 299.993)
(608.156, 1672.409, 299.993)
(474.105, 1744.272, 299.993)
(281.956, 1740.018, 299.994)
(206.855, 1682.662, 299.995)
(28.950, 1682.682, 299.995)
(-103.565, 1672.622, 299.995)
(-146.394, 1605.316, 299.995)
(-618.620, 1608.822, 306.535) | (-612.109, 1773.636, 273.971)
(-450.941, 2059.763, 263.197)
(-153.782, 2087.542, 280.616)
(-31.106, 2078.945, 280.616)
(124.921, 2291.537, 270.210)
(369.216, 2529.219, 270.212)
(1075.481, 2872.042, 179.045)
(1195.161, 2406.784, 171.651)
(635.556, 2026.965, 188.813)
(225.034, 1879.450, 237.215)
(-465.376, 2016.724, 229.713)
(-399.756, 2902.279, 229.275) |

| Scene Name | Scene Map | Low-altitude Points | Mid-altitude Points | High-altitude Points |
|---|---|---|---|---|
| | | (5948.222, -4033.881, -3.000) | (6241.531, -4705.508, -17.000) | (5890.578, -4866.875, -35.000) |
| | | (5993.686, -4033.881, -3.000) | (6388.013, -4645.165, -17.000) | (6031.218, -4806.867, -35.000) |
| | | (5995.087, -3983.510, -3.000) | (6480.295, -4623.860, -17.000) | (6127.604, -4748.621, -35.000) |
| | | (5998.127, -3944.917, -3.000) | (6592.987, -4588.834, -17.000) | (6279.022, -4671.134, -35.000) |
| | | (6000.699, -3875.759, -3.000) | (6681.311, -4568.443, -17.000) | (6432.394, -4633.987, -35.000) |
| | | (6001.806, -3812.312, -3.000) | (6777.038, -4539.845, -17.000) | (6476.112, -4529.924, -35.000) |
| | | (5998.611, -3700.258, -3.000) | (6898.770, -4519.922, -17.000) | (6475.305, -4376.582, -35.000) |
| | | (5999.305, -3630.596, -3.000) | (7039.883, -4512.527, -17.000) | (6464.704, -4226.787, -35.000) |
| | | (5998.527, -3579.056, -3.000) | (7179.951, -4505.186, -17.000) | (6469.441, -3981.623, -35.000) |
| | | (6027.343, -3632.901, -3.000) | (7159.016, -4882.421, -17.000) | (6606.227, -3916.281, -35.000) |
| | | (6109.682, -3634.518, -3.000) | (7140.139, -5088.776, -17.000) | (6772.756, -3913.637, -35.000) |
| | | (6197.296, -3634.390, -3.000) | (6962.964, -5248.603, -17.000) | (6894.070, -3910.247, -35.000) |
| | | (6207.231, -3714.186, -3.000) | (6726.297, -5248.143, -17.000) | (6995.506, -3912.017, -35.000) |
| | | (6209.154, -3879.545, -3.000) | (6440.733, -5312.550, -17.000) | (7149.131, -3875.853, -35.000) |
| | | (6207.116, -3919.984, -3.000) | (6399.547, -4999.819, -17.000) | (7216.097, -4135.895, -35.000) |
| | | (6314.389, -3915.768, -3.000) | (6589.197, -4589.403, -17.000) | (7210.454, -4353.343, -35.000) |
| | | (6459.531, -3915.768, -3.000) | (6475.974, -4585.449, -17.000) | (7100.654, -4521.011, -35.000) |
| | | (6476.874, -4019.463, -3.000) | (6474.979, -4396.807, -17.000) | (6943.265, -4523.094, -35.000) |
| | | (6474.580, -4154.050, -3.000) | (6407.366, -4199.765, -17.000) | (6748.140, -4551.673, -35.000) |
| | | (6471.100, -4253.688, -3.000) | (6472.378, -4067.782, -17.000) | (6513.162, -4611.606, -35.000) |
| | | (6469.600, -4400.345, -3.000) | (6475.749, -3971.271, -17.000) | (6314.093, -4671.002, -35.000) |
| | | (6474.832, -4574.854, -3.000) | (6475.675, -3792.963, -17.000) | (6138.969, -4762.144, -35.000) |
| Embodied_City | | (6470.219, -4640.397, -3.000) | (6350.728, -3636.378, -17.000) | (5969.305, -4829.059, -35.000) |
| | | (6471.215, -4774.531, -3.000) | (6202.980, -3630.616, -17.000) | (5870.481, -4863.039, -35.000) |
| | | (6469.685, -4915.708, -3.000) | (6089.829, -3630.165, -17.000) | (5775.307, -4325.917, -35.000) |
| | | (6477.274, -5339.204, -3.000) | (6011.199, -3628.792, -17.000) | (6003.525, -4181.135, -70.000) |
| | | (6762.667, -5058.132, -3.000) | (5846.073, -3631.596, -17.000) | (6176.218, -4186.541, -70.000) |
| | | (6974.092, -4962.887, -3.000) | (5692.889, -3628.923, -17.000) | (6371.880, -4189.956, -70.000) |
| | | (7148.401, -5077.070, -3.000) | (5620.646, -3908.698, -17.000) | (6602.166, -4195.726, -70.000) |
| | | (7247.312, -5081.909, -3.000) | (5742.287, -4175.687, -17.000) | (6734.788, -4180.065, -70.000) |
| | | (7257.460, -5339.683, -3.000) | (5998.052, -4180.561, -17.000) | (6725.812, -4310.885, -70.000) |
| | | (7175.379, -5062.077, -3.000) | (6127.672, -4177.669, -17.000) | (6755.426, -4522.965, -70.000) |
| | | (7036.920, -4939.656, -3.000) | (6264.615, -4187.245, -17.000) | (6719.977, -4846.263, -70.000) |
| | | (6841.633, -4805.606, -3.000) | (6383.114, -4189.384, -17.000) | (6640.424, -5028.033, -70.000) |
| | | (6732.402, -4705.335, -3.000) | (6476.690, -4183.088, -17.000) | (6433.556, -4887.969, -70.000) |
| | | (6717.309, -4559.614, -3.000) | (6632.462, -4186.132, -17.000) | (6260.846, -4727.553, -70.000) |
| | | (6571.285, -4598.712, -3.000) | (6762.124, -4186.132, -17.000) | (6208.921, -4574.336, -70.000) |
| | | (6473.776, -4586.892, -3.000) | (6931.914, -4191.464, -17.000) | (6359.913, -4460.806, -70.000) |
| | | (6473.344, -4402.319, -3.000) | (7203.099, -4196.529, -17.000) | (6431.431, -4172.967, -70.000) |
| | | (6475.043, -4305.007, -3.000) | (7223.191, -4444.652, -17.000) | (6475.348, -3967.526, -70.000) |
| | | (6404.955, -4189.553, -3.000) | (7128.851, -4512.014, -17.000) | (6706.256, -3915.028, -70.000) |
| | | (6614.860, -4179.223, -3.000) | (6951.857, -4519.316, -17.000) | (6920.749, -3978.040, -70.000) |
| | | (6719.227, -4264.966, -3.000) | (6813.986, -4539.558, -17.000) | (6971.826, -4139.938, -70.000) |
| | | (6722.694, -4292.494, -3.000) | (6647.727, -4573.792, -17.000) | (6968.629, -4466.940, -70.000) |
| | | (6725.189, -4398.576, -3.000) | (6515.922, -4611.586, -17.000) | (6829.308, -4536.860, -70.000) |
| | | (6725.189, -4398.576, -3.000) | (6476.886, -4739.687, -17.000) | (6575.293, -4587.064, -70.000) |
| | | (6725.189, -4398.576, -3.000) | (6471.735, -4997.638, -17.000) | (6335.976, -4683.194, -70.000) |
| | | (6480.953, -4429.711, -3.000) | (6478.620, -5164.615, -17.000) | (6141.691, -4753.320, -70.000) |
| | | (6278.803, -4484.838, -3.000) | (6006.740, -5130.435, -17.000) | (6019.621, -4674.785, -70.000) |
| | | (6218.582, -4719.335, -3.000) | (5814.432, -4902.671, -17.000) | (6002.608, -4290.909, -70.000) |

## A.3. Fliter

To maintain high data fidelity across diverse and heterogeneous sources, we develop a modular, two-stage refinement pipeline that decouples visual metric extraction from temporal sequence pruning. This architecture enables incremental updates to filtering criteria without re-evaluating the entire dataset, facilitating an efficient "detect-then-prune" workflow. The process begins with frame-level anomaly detection to capture transient visual artifacts, followed by a statistical sample filtering stage that transforms discrete anomalies into coherent, high-fidelity temporal segments, ensuring long-term consistency for downstream training. The first stage identifies per-frame point anomalies through two primary visual metrics: brightness stability and chromatic mutation. For peak exposure detection, we compute the 95th percentile of grayscale intensity $M_t^{light} = P_{95}(\mathrm{gray}(x_t))$. To capture rendering glitches or "wall-clipping" artifacts, we measure the Mean Squared Error (MSE) between consecutive frames, $M_t^{MSE}$. We define the anomaly detection criteria using median residuals and rolling Z-scores as follows:

$$\text{Flag } t \in \mathcal{B} \text{ if: } \begin{cases} |M_t^{light} - \mathrm{median}(M_{t-1:t+1}^{light})| > \max(3\sigma_{res}, 10) \\ Z_t = (M_t^{MSE} - \mu_{win})/\sigma_{win} > 4 \end{cases} \quad (1)$$

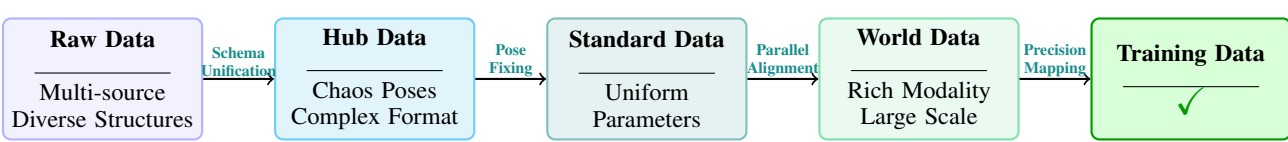

*Figure 4.* The systematic data curation pipeline: transforming raw multi-source inputs into high-fidelity training data via structured standardization and trajectory rectification.

where $\sigma_{res}$ is the local residual deviation, and $(\mu_{win}, \sigma_{win})$ are computed over a rolling window. These detected indices are logged in a non-destructive metadata dictionary $\mathcal{B}$ for downstream density analysis. The second stage, statistical sample filtering, transforms these discrete point anomalies into continuous high-quality segments by evaluating the local error distribution. Given an anomaly indicator signal $I_t \in \{0, 1\}$ (where $I_t = 1$ if $t \in \mathcal{B}$), we compute a local anomaly density $\rho_t$ via a 1D boxcar convolution:

$$\rho_t = \frac{1}{W} \sum_{k=t-W/2}^{t+W/2} I_k \tag{2}$$

Potential "high-fidelity zones" are identified where $\rho_t$ remains below a threshold $\tau = 0.06$. To ensure sequence robustness, we apply a bridge-merging operation to connect proximal clean segments, followed by a duration pruning step that discards segments shorter than $L_{min}$ frames. This rigorous process yields a refined dataset characterized by superior temporal stability, visual coherence, and high-fidelity motion trajectories, providing a clean and reliable distribution for generative modeling.

---

**Algorithm 1** Global Video Refinement and Statistical Pruning

---

**Require:** Video Dataset $\mathcal{D}_{raw}$, Density threshold $\tau$, Min duration $L_{min}$, Merge gap $G_{merge}$
**Ensure:** Refined high-fidelity dataset $\mathcal{S}$

1: **for** each video $\mathcal{V} \in \mathcal{D}_{raw}$ **do**
2:      $\mathcal{B} \leftarrow \emptyset$
3:      **for** $t = 1$ **to** $T$ **do**
4:          Compute $M_t^{light}$ and $M_t^{MSE}$
5:          **if** is_outlier$(M_t^{light}) \vee$ Z-Score$(M_t^{MSE}) > 4$ **then**
6:              $\mathcal{B} \leftarrow \mathcal{B} \cup \{t\}$
7:          **end if**
8:      **end for**
9:      $I \leftarrow$ ConstructSignal$(\mathcal{B}, T)$
10:     $\rho \leftarrow I * $Kernel$(W)$
11:     $\mathcal{G} \leftarrow \{t \mid \rho_t < \tau\}$
12:     $\mathcal{G}_{merged} \leftarrow$ Bridge$(\mathcal{G}, G_{merge})$
13:     $\mathcal{G}_{final} \leftarrow \{g \in \mathcal{G}_{merged} \mid \text{length}(g) \geq L_{min}\}$
14:     **for** each segment $g \in \mathcal{G}_{final}$ **do**
15:        $\mathcal{S} \leftarrow \mathcal{S} \cup \{\text{Frames}(g)\}$
16:     **end for**
17: **end for**
18: **return** $\mathcal{S}$

---

The construction of a large-scale training corpus necessitates a systematic pipeline capable of transforming diverse, heterogeneous raw data into a high-fidelity, standardized format. This process is structured into four interconnected functional stages: multi-source acquisition, structural standardization, spatial rectification, and interactive model synthesis.

The pipeline commences with a multi-source data acquisition and synchronization stage. Depending on the source infrastructure, data is retrieved via direct server-side synchronization or manual cloud-storage fetching. To optimize engineering efficiency, a "probabilistic observation" strategy is implemented during the pre-download phase, where minimal data subsets are extracted to calibrate the specific coordinate system of each dataset and map original axes to a global motion reference. This ensures that the subsequent full-scale download—prioritizing RGB sequences, depth maps, and camera parameters—is executed accurately within pre-allocated storage constraints. Following acquisition, the data enters a structural standardization stage, where disparate raw formats are ingested into CityWorld, a unified hierarchical representation. This stage utilizes a modular framework that supports idempotent execution and checkpoint-resume capabilities, organizing data into a four-layer architecture: a modality layer (separating RGB, depth, and pose), a category layer (classified by agent perspective), an identity layer (unique sequence IDs), and a parameter layer. To maximize downstream compatibility, camera extrinsics are maintained in three redundant, mutually convertible formats: transformation matrices, quaternions, and six-degree-of-freedom (6-DoF) vectors.

Building upon this standardized foundation, the spatial rectification stage addresses inconsistencies in spatial orientation across different sources. By evaluating the calibrated motion mappings from the initial phase, mathematical corrections are applied to align all trajectories with a unified right-handed coordinate system. For instance, matrix-based poses are rectified through a diagonal correction matrix to ensure that translation and rotation components are physically consistent across the entire corpus. Finally, the process culminates in the interactive model synthesis stage, which performs temporal quantization and semantic packaging. Continuous trajectories are partitioned into standardized 81-frame clips, while sequences failing to meet this duration are discarded to maintain batch homogeneity. Each synthesized sample is paired with synchronized intrinsics and row-major 3x4 projection matrices, yielding a refined dataset characterized by superior temporal stability and visual coherence.

## A.4. Data Annotation Prompts

To ensure high-quality annotations, we constructed a structured prompt pipeline using GPT-4o (Hurst et al., 2024). The specific instructions provided to the model, including the expert persona definition and the logical analysis flow, are presented in Listing A.4. We used **GPT-4o 2025-01-01-preview** and processed a total of 330,000 videos. This large-scale annotation process involved 119 million input tokens and 21.86 million output tokens, with an associated financial cost of approximately $518.

---

**Data Annotation Prompt**

**META PROMPT:**
You are an AI World Model Data Specialist.
Your task is to annotate first-person view (FPV) video sequences.
You must analyze the input frames as a continuous sequence to understand the environment structure.

---

```
Output strictly valid JSON.
USER TASK PROMPT:
Please analyze the 5 provided frames following this logical flow:
1. Spatial Analysis (Indoor/Outdoor):
- Determine if the agent is Indoor (enclosed) or Outdoor (open air).
2. Detailed Scene Description:
- Generate a descriptive English phrase for the specific scene visible in the frames.
- Example: 'Abandoned industrial courtyard with rusty pipes and overgrown grass'.
3. Scene Categorization (Dynamic Summary):
- Based on your description above, summarize the scene into a SINGLE Root Noun (Category).
- This should be the most representative word for the place.
- Do NOT use adjectives here, just the noun.
4. Atmospheric Analysis:
- Outdoor: Weather (Sunny, Cloudy, Rainy, Night).
- Indoor: Lighting (Fluorescent, Dim, Natural).
5. Entity Extraction:
- List 15+ distinct objects visible in the scene (structural + dynamic).
OUTPUT JSON FORMAT:
{
  "Environment_Type": "Indoor" or "Outdoor",
  "Scene_Description": "Your detailed descriptive phrase",
  "Scene_Tag": "Single Root Noun (The dynamic summary)",
  "Weather_Lighting": "Weather or Lighting condition",
  "Entities": "List of objects separated by commas"
}
```

## A.5. Multi-Model Verification and Human Refinement

To assess labeling quality and reduce the risk that a single VLM injects hallucinations or systematic tagging preferences into the corpus, we did not directly accept GPT-4o annotations as final labels. Instead, each of the 330,000 video annotations was independently verified by three additional VLMs with different model families: Gemini 3.0 Flash, Qwen-VL-Max, and Kimi-K2.5. Given the sampled video frames and the existing annotation fields, each verifier was asked to make a conservative binary judgment on whether the annotation was accurate overall. The prompt used for this verification stage is shown below.

**Data Verification Prompt**

```
VERIFY PROMPT:
You are an expert evaluator verifying whether an AI-generated video
annotation accurately describes the given video frames.

The existing annotation is:
- Environment Type: {environment_type}
- Scene Tag: {scene_tag}
- Scene Description: {scene_description}
- Weather/Lighting: {weather_lighting}
- Entities present: {entities}

Please carefully analyze all provided video frames and determine whether
this annotation is accurate and correct overall.

Respond in EXACTLY this format (no other text before or after):
Think: [your detailed step-by-step reasoning comparing the annotation
against what you observe in the frames]
Answer: Yes or No
```

**Agreement-based quality assessment.** We use the three verifier outputs as a quality assessment signal rather than as an automatic relabeling mechanism. In particular, unanimous agreement is treated as a high-confidence label, while any cross-model disagreement is treated as uncertainty and routed to human annotators. This design is stricter than majority voting: even a 2/3 "Yes" case is not automatically accepted, because such disagreements often correspond to ambiguous weather, lighting, scene granularity, or missing entity tags. The resulting agreement distribution is summarized in Table 6.

**Human refinement protocol.** Overall, 61,380 non-unanimous clips, corresponding to 18.6% of the full corpus, were flagged for human inspection. Annotators reviewed the sampled frames together with the GPT-4o label and the verifier decisions, then either accepted the original label or corrected the inaccurate fields. Only 6.35% of the flagged clips required an actual edit, corresponding to approximately 3,897 clips, or 1.2% of the full 330,000-clip corpus. For the 268,620 unanimously accepted clips, we additionally spot-checked 10,000 samples and observed a 100% pass rate under the same review criterion. Therefore, 98.8% of the annotations ultimately required no modification after verification.

**Manual versus model-based inspection.** At the corpus level, model-based quality inspection covers 100% of the 330,000 clips, with three independent verification judgments per clip. Human inspection covers all disagreement cases plus the 10,000 unanimous-case spot checks, yielding 71,380 manually inspected clips in total, or 21.6% of the corpus.

*Table 6.* Agreement distribution in the multi-model verification stage. All non-unanimous cases were routed to human review rather than automatically resolved by majority vote.

| Verifier Agreement | Ratio | Approx. Clips | Handling Policy |
|---|---|---|---|
| 3/3 Yes | 81.4% | 268,620 | Accepted with manual spot check |
| 2/3 Yes | 15.1% | 49,830 | Human review |
| 1/3 Yes | 2.9% | 9,570 | Human review |
| 0/3 Yes | 0.5% | 1,650 | Human review |

The final benchmark subset was then manually curated and graded as described in Appendix A.6. We used the same disagreement-triggered policy across scene, perspective, weather/lighting, and downstream task categories; thus, category-specific variation arises only from the observed model disagreement frequency rather than from different acceptance thresholds. This prevents any single VLM from unilaterally determining labels in ambiguous cases and reduces the chance that the benchmark favors a particular model's annotation style.

**Annotation effort and cost.** The human verification process was carried out by 10 volunteers with approximately 1,200 person-hours of effort. The complete multi-model verification stage remained practically scalable, costing approximately $2.8K in API fees for all 330,000 clips.

## A.6. Human Annotation Procedures

To ensure the high diversity and semantic precision of the benchmark, we implemented a diversity-driven data curation strategy that serves as a prerequisite for manual inspection. Leveraging the comprehensive metadata generated by the Multi-modal Large Language Model during the automated annotation phase, we obtained fine-grained labels covering spatial analysis (identifying Indoor versus Outdoor environments), atmospheric conditions (ranging from sunny and cloudy to complex lighting scenarios like dim or night settings), and detailed entity extraction. Instead of employing simple random sampling which might lead to data redundancy, we utilized these semantic attributes to conduct a stratified sampling of the candidate videos. This approach allowed us to deliberately balance the distribution of scenes—selecting from a wide array of environments such as plazas, streets, and villages—while simultaneously ensuring a rich variety of dynamic entities and weather conditions. By systematically filtering the dataset based on these VLM-generated tags, we constructed a candidate pool that maximizes the coverage of real-world physical scenarios, providing a robust foundation for the subsequent human verification process.

To facilitate the rigorous verification and grading of these candidates, we developed a dedicated in-house annotation tool tailored for this benchmark. This specialized software interface integrates video playback with a dynamic grading panel, allowing annotators to efficiently review the VLM-generated metadata and define the final generation tasks. The annotation workflow within the tool follows a strict "Verify-Reject-Grade" protocol based on the **first frame** of each video: annotators first validate the visual content to reject hallucinations or ambiguous samples, and then manually correct any discrepancies in the scene or entity tags. Crucially, the tool provides specific controls for task formulation, where annotators identify special categories such as the Memory task—which tests the model's ability to retain initial visual context for operations like video inversion—and assign one of four difficulty levels to the sequence. These difficulty ratings are determined by visual complexities observed in the first frame, such as object occlusion and background clutter. Upon finalizing the review, the tool supports a one-click export function that automatically pairs the video's first frame with its verified task instructions and difficulty labels, ensuring a standardized and high-quality output for the final dataset.

# B. Framework Details

The design of a precise motion encoding scheme is foundational for analyzing and generating interactions. We propose a quadruple descriptor to formulate both translation and rotation within a unified representation. This systematic framework facilitates the construction of an action space that is both quantifiable and extensible. The following subsections detail this encoding specification.

## B.1. Action Mapping

This subsection details the motion encoding employed in this paper. Specifically, we design a quadruple to describe translation and rotation, comprising Difficulty, ID, Direction and Keys as presented in Table 7. We utilize three-dimensional coordinates to describe position and rotations about the three coordinate axes to describe the camera's orientation. Based on this representation, the motion difficulty for a single parameter is defined as 1. If multiple parameters change simultaneously, their difficulty values are summed. Notably, the state of complete stillness is also assigned a difficulty of 1 and is included in our complete action space.

For both translation and rotation, there are 27 distinct actions, each of which is assigned a unique identifier. Moreover, the entries not shaded in gray in the table correspond to actions that have predefined key controls. We have also comprehensively incorporated actions that currently lack assigned keys. Should corresponding keys be defined in the future, the applicability of our framework remains fully preserved.

| Translation Motion | | | | Rotation Motion | | | |
|---|---|---|---|---|---|---|---|
| **Difficulty** | **ID** | **Direction** | **Keys** | **Difficulty** | **ID** | **Direction** | **Keys** |
| 1 | 0 | Stationary | - | 1 | 0 | Stationary | - |
| 1 | 1 | Forward | W | 1 | 1 | Camera Up | ↑ |
| 1 | 2 | Backward | S | 1 | 2 | Camera Down | ↓ |
| 1 | 3 | Left | A | 1 | 3 | Camera Right | → |
| 1 | 4 | Right | D | 1 | 4 | Camera Left | ← |
| 2 | 5 | Forward+Left | W+A | 2 | 5 | Camera Up+Right | ↑+→ |
| 2 | 6 | Forward+Right | W+D | 2 | 6 | Camera Up+Left | ↑+← |
| 2 | 7 | Backward+Left | S+A | 2 | 7 | Camera Down+Right | ↓+→ |
| 2 | 8 | Backward+Right | S+D | 2 | 8 | Camera Down+Left | ↓+← |
| 1 | 9 | Upward | - | 1 | 9 | Clockwise | - |
| 1 | 10 | Downward | - | 1 | 10 | Counterclockwise | - |
| 2 | 11 | Forward+Upward | - | 2 | 11 | Camera Up+Clockwise | - |
| 2 | 12 | Forward+Downward | - | 2 | 12 | Camera Up+Counterclockwise | - |
| 2 | 13 | Backward+Upward | - | 2 | 13 | Camera Down+Clockwise | - |
| 2 | 14 | Backward+Downward | - | 2 | 14 | Camera Down+Counterclockwise | - |
| 2 | 15 | Left+Upward | - | 2 | 15 | Camera Left+Clockwise | - |
| 2 | 16 | Left+Downward | - | 2 | 16 | Camera Left+Counterclockwise | - |
| 2 | 17 | Right+Upward | - | 2 | 17 | Camera Right+Clockwise | - |
| 2 | 18 | Right+Downward | - | 2 | 18 | Camera Right+Counterclockwise | - |
| 3 | 19 | Forward+Left+Upward | - | 3 | 19 | Camera Up+Right+Clockwise | - |
| 3 | 20 | Forward+Right+Upward | - | 3 | 20 | Camera Up+Right+Counterclockwise | - |
| 3 | 21 | Forward+Left+Downward | - | 3 | 21 | Camera Up+Left+Clockwise | - |
| 3 | 22 | Forward+Right+Downward | - | 3 | 22 | Camera Up+Left+Counterclockwise | - |
| 3 | 23 | Backward+Left+Upward | - | 3 | 23 | Camera Down+Right+Clockwise | - |
| 3 | 24 | Backward+Right+Upward | - | 3 | 24 | Camera Down+Left+Counterclockwise | - |
| 3 | 25 | Backward+Left+Downward | - | 3 | 25 | Camera Down+Right+Clockwise | - |
| 3 | 26 | Backward+Right+Downward | - | 3 | 26 | Camera Down+Left+Counterclockwise | - |

*Table 7.* Action mapping table. This table enumerates motion primitives with corresponding difficulty levels and unique IDs for both translation and rotation movements. Keyboard controls are provided where applicable.

## B.2. Entire Action Space

This subsection presents the mapping table for the Entire Action Space, labeled as 8.The table contains the complete set of 729 motions. These are the permutations of the 27 distinct translation actions and the 27 distinct rotation actions defined in Table 7.The difficulty range spans from 1 to 6. Specifically, all stationary is defined with a difficulty of 1.

**Note: D**: Difficulty; **T**: Translation ID; **R**: Rotation ID; **V**: Validity

Where validity is defined based on the frequency of corresponding actions in our collected dataset. Actions considered exceptional cases are determined as invalid.

| Col 1 | | | | Col 2 | | | | Col 3 | | | | Col 4 | | | | Col 5 | | | | Col 6 | | | | Col 7 | | | | Col 8 | | | | Col 9 | | | | Col 10 | | | | Col 11 | | | |
|---|---|---|---|---|---|---|---|---|---|---|---|---|---|---|---|---|---|---|---|---|---|---|---|---|---|---|---|---|---|---|---|---|---|---|---|---|---|---|---|---|---|---|---|
| D | T | R | V | D | T | R | V | D | T | R | V | D | T | R | V | D | T | R | V | D | T | R | V | D | T | R | V | D | T | R | V | D | T | R | V | D | T | R | V | D | T | R | V |
| 1 | 0 | 0 | 0 | 2 | 13 | 0 | 0 | 3 | 7 | 2 | 1 | 3 | 16 | 3 | 1 | 4 | 5 | 8 | 1 | 4 | 11 | 13 | 1 | 4 | 17 | 6 | 1 | 4 | 26 | 3 | 1 | 5 | 14 | 26 | 1 | 5 | 21 | 17 | 1 | 6 | 19 | 24 | 1 |
| 1 | 0 | 1 | 0 | 2 | 14 | 0 | 0 | 3 | 7 | 3 | 1 | 3 | 16 | 4 | 1 | 4 | 5 | 11 | 1 | 4 | 11 | 14 | 1 | 4 | 17 | 7 | 1 | 4 | 26 | 4 | 1 | 5 | 15 | 19 | 1 | 5 | 21 | 18 | 1 | 6 | 19 | 25 | 1 |
| 1 | 0 | 2 | 0 | 2 | 15 | 0 | 0 | 3 | 7 | 4 | 1 | 3 | 16 | 9 | 1 | 4 | 5 | 12 | 1 | 4 | 11 | 15 | 1 | 4 | 17 | 8 | 1 | 4 | 26 | 9 | 1 | 5 | 15 | 20 | 1 | 5 | 22 | 5 | 1 | 6 | 19 | 26 | 1 |
| 1 | 0 | 3 | 0 | 2 | 16 | 0 | 0 | 3 | 7 | 9 | 1 | 3 | 16 | 10 | 1 | 4 | 5 | 13 | 1 | 4 | 11 | 16 | 1 | 4 | 17 | 11 | 1 | 4 | 26 | 10 | 1 | 5 | 15 | 21 | 1 | 5 | 22 | 6 | 1 | 6 | 20 | 19 | 1 |
| 1 | 0 | 4 | 0 | 2 | 17 | 0 | 0 | 3 | 7 | 10 | 1 | 3 | 17 | 1 | 1 | 4 | 5 | 14 | 1 | 4 | 11 | 17 | 1 | 4 | 17 | 12 | 1 | 5 | 5 | 19 | 1 | 5 | 15 | 22 | 1 | 5 | 22 | 7 | 1 | 6 | 20 | 20 | 1 |
| 1 | 0 | 9 | 0 | 2 | 18 | 0 | 0 | 3 | 8 | 1 | 1 | 3 | 17 | 2 | 1 | 4 | 5 | 15 | 1 | 4 | 11 | 18 | 1 | 4 | 17 | 13 | 1 | 5 | 5 | 20 | 1 | 5 | 15 | 23 | 1 | 5 | 22 | 8 | 1 | 6 | 20 | 21 | 1 |
| 1 | 0 | 10 | 0 | 3 | 1 | 5 | 0 | 3 | 8 | 2 | 1 | 3 | 17 | 3 | 1 | 4 | 5 | 16 | 1 | 4 | 12 | 5 | 1 | 4 | 17 | 14 | 1 | 5 | 5 | 21 | 1 | 5 | 15 | 24 | 1 | 5 | 22 | 11 | 1 | 6 | 20 | 22 | 1 |
| 1 | 1 | 0 | 0 | 3 | 1 | 6 | 0 | 3 | 8 | 3 | 1 | 3 | 17 | 4 | 1 | 4 | 5 | 17 | 1 | 4 | 12 | 6 | 1 | 4 | 17 | 15 | 1 | 5 | 5 | 22 | 1 | 5 | 15 | 25 | 1 | 5 | 22 | 12 | 1 | 6 | 20 | 23 | 1 |
| 1 | 2 | 0 | 0 | 3 | 1 | 7 | 1 | 3 | 8 | 4 | 1 | 3 | 17 | 9 | 1 | 4 | 5 | 18 | 1 | 4 | 12 | 7 | 1 | 4 | 17 | 16 | 1 | 5 | 5 | 23 | 1 | 5 | 15 | 26 | 1 | 5 | 22 | 13 | 1 | 6 | 20 | 24 | 1 |
| 1 | 3 | 0 | 0 | 3 | 1 | 8 | 1 | 3 | 8 | 9 | 1 | 3 | 17 | 10 | 1 | 4 | 6 | 5 | 1 | 4 | 12 | 8 | 1 | 4 | 17 | 17 | 1 | 5 | 5 | 24 | 1 | 5 | 16 | 19 | 1 | 5 | 22 | 14 | 1 | 6 | 20 | 25 | 1 |
| 1 | 4 | 0 | 0 | 3 | 1 | 11 | 1 | 3 | 8 | 10 | 1 | 3 | 18 | 1 | 1 | 4 | 6 | 6 | 1 | 4 | 12 | 11 | 1 | 4 | 17 | 18 | 1 | 5 | 5 | 25 | 1 | 5 | 16 | 20 | 1 | 5 | 22 | 15 | 1 | 6 | 20 | 26 | 1 |
| 1 | 9 | 0 | 0 | 3 | 1 | 12 | 1 | 3 | 9 | 5 | 1 | 3 | 18 | 2 | 1 | 4 | 6 | 7 | 1 | 4 | 12 | 12 | 1 | 4 | 18 | 5 | 1 | 5 | 5 | 26 | 1 | 5 | 16 | 21 | 1 | 5 | 22 | 16 | 1 | 6 | 21 | 19 | 1 |
| 1 | 10 | 0 | 0 | 3 | 1 | 13 | 1 | 3 | 9 | 6 | 1 | 3 | 18 | 3 | 1 | 4 | 6 | 8 | 1 | 4 | 12 | 13 | 1 | 4 | 18 | 6 | 1 | 5 | 6 | 19 | 1 | 5 | 16 | 22 | 1 | 5 | 22 | 17 | 1 | 6 | 21 | 20 | 1 |
| 2 | 1 | 1 | 0 | 3 | 1 | 14 | 1 | 3 | 9 | 7 | 1 | 3 | 18 | 4 | 1 | 4 | 6 | 11 | 1 | 4 | 12 | 14 | 1 | 4 | 18 | 7 | 1 | 5 | 6 | 20 | 1 | 5 | 16 | 23 | 1 | 5 | 22 | 18 | 1 | 6 | 21 | 21 | 1 |
| 2 | 1 | 2 | 0 | 3 | 1 | 15 | 1 | 3 | 9 | 8 | 1 | 3 | 18 | 9 | 1 | 4 | 6 | 12 | 1 | 4 | 12 | 15 | 1 | 4 | 18 | 8 | 1 | 5 | 6 | 21 | 1 | 5 | 16 | 24 | 1 | 5 | 23 | 5 | 1 | 6 | 21 | 22 | 1 |
| 2 | 1 | 3 | 0 | 3 | 1 | 16 | 1 | 3 | 9 | 11 | 1 | 3 | 18 | 10 | 1 | 4 | 6 | 13 | 1 | 4 | 12 | 16 | 1 | 4 | 18 | 11 | 1 | 5 | 6 | 22 | 1 | 5 | 16 | 25 | 1 | 5 | 23 | 6 | 1 | 6 | 21 | 23 | 1 |
| 2 | 1 | 4 | 0 | 3 | 1 | 17 | 1 | 3 | 9 | 12 | 1 | 3 | 0 | 19 | 0 | 4 | 6 | 14 | 1 | 4 | 12 | 17 | 1 | 4 | 18 | 12 | 1 | 5 | 6 | 23 | 1 | 5 | 16 | 26 | 1 | 5 | 23 | 7 | 1 | 6 | 21 | 24 | 1 |
| 2 | 1 | 9 | 0 | 3 | 1 | 18 | 1 | 3 | 9 | 13 | 1 | 3 | 0 | 20 | 0 | 4 | 6 | 15 | 1 | 4 | 12 | 18 | 1 | 4 | 18 | 13 | 1 | 5 | 6 | 24 | 1 | 5 | 17 | 19 | 1 | 5 | 23 | 8 | 1 | 6 | 21 | 25 | 1 |
| 2 | 1 | 10 | 0 | 3 | 2 | 5 | 1 | 3 | 9 | 14 | 1 | 3 | 0 | 21 | 0 | 4 | 6 | 16 | 1 | 4 | 13 | 5 | 1 | 4 | 18 | 14 | 1 | 5 | 6 | 25 | 1 | 5 | 17 | 20 | 1 | 5 | 23 | 11 | 1 | 6 | 21 | 26 | 1 |
| 2 | 2 | 1 | 0 | 3 | 2 | 6 | 1 | 3 | 9 | 15 | 1 | 3 | 0 | 22 | 0 | 4 | 6 | 17 | 1 | 4 | 13 | 6 | 1 | 4 | 18 | 15 | 1 | 5 | 6 | 26 | 1 | 5 | 17 | 21 | 1 | 5 | 23 | 12 | 1 | 6 | 22 | 19 | 1 |
| 2 | 2 | 2 | 0 | 3 | 2 | 7 | 1 | 3 | 9 | 16 | 1 | 3 | 0 | 23 | 0 | 4 | 6 | 18 | 1 | 4 | 13 | 7 | 1 | 4 | 18 | 16 | 1 | 5 | 7 | 19 | 1 | 5 | 17 | 22 | 1 | 5 | 23 | 13 | 1 | 6 | 22 | 20 | 1 |
| 2 | 2 | 3 | 0 | 3 | 2 | 8 | 0 | 3 | 9 | 17 | 1 | 3 | 0 | 24 | 0 | 4 | 7 | 5 | 1 | 4 | 13 | 8 | 1 | 4 | 18 | 17 | 1 | 5 | 7 | 20 | 1 | 5 | 17 | 23 | 1 | 5 | 23 | 14 | 1 | 6 | 22 | 21 | 1 |
| 2 | 2 | 4 | 0 | 3 | 2 | 11 | 1 | 3 | 9 | 18 | 1 | 3 | 0 | 25 | 0 | 4 | 7 | 6 | 1 | 4 | 13 | 11 | 1 | 4 | 18 | 18 | 1 | 5 | 7 | 21 | 1 | 5 | 17 | 24 | 1 | 5 | 23 | 15 | 1 | 6 | 22 | 22 | 1 |
| 2 | 2 | 9 | 1 | 3 | 2 | 12 | 1 | 3 | 10 | 5 | 1 | 3 | 0 | 26 | 0 | 4 | 7 | 7 | 1 | 4 | 13 | 12 | 1 | 4 | 19 | 1 | 1 | 5 | 7 | 22 | 1 | 5 | 17 | 25 | 1 | 5 | 23 | 16 | 1 | 6 | 22 | 23 | 1 |
| 2 | 2 | 10 | 1 | 3 | 2 | 13 | 1 | 3 | 10 | 6 | 1 | 3 | 19 | 0 | 0 | 4 | 7 | 8 | 1 | 4 | 13 | 13 | 1 | 4 | 19 | 2 | 1 | 5 | 7 | 23 | 1 | 5 | 17 | 26 | 1 | 5 | 23 | 17 | 1 | 6 | 22 | 24 | 1 |
| 2 | 3 | 1 | 1 | 3 | 2 | 14 | 1 | 3 | 10 | 7 | 1 | 3 | 20 | 0 | 0 | 4 | 7 | 11 | 1 | 4 | 13 | 14 | 1 | 4 | 19 | 3 | 1 | 5 | 7 | 24 | 1 | 5 | 18 | 19 | 1 | 5 | 23 | 18 | 1 | 6 | 22 | 25 | 1 |
| 2 | 3 | 2 | 1 | 3 | 2 | 15 | 1 | 3 | 10 | 8 | 1 | 3 | 21 | 0 | 0 | 4 | 7 | 12 | 1 | 4 | 13 | 15 | 1 | 4 | 19 | 4 | 1 | 5 | 7 | 25 | 1 | 5 | 18 | 20 | 1 | 5 | 24 | 5 | 1 | 6 | 22 | 26 | 1 |
| 2 | 3 | 3 | 0 | 3 | 2 | 16 | 1 | 3 | 10 | 11 | 1 | 3 | 22 | 0 | 0 | 4 | 7 | 13 | 1 | 4 | 13 | 16 | 1 | 4 | 19 | 9 | 1 | 5 | 7 | 26 | 1 | 5 | 18 | 21 | 1 | 5 | 24 | 6 | 1 | 6 | 23 | 19 | 1 |
| 2 | 3 | 4 | 0 | 3 | 2 | 17 | 1 | 3 | 10 | 12 | 1 | 3 | 23 | 0 | 0 | 4 | 7 | 14 | 1 | 4 | 13 | 17 | 1 | 4 | 19 | 10 | 1 | 5 | 8 | 19 | 1 | 5 | 18 | 22 | 1 | 5 | 24 | 7 | 1 | 6 | 23 | 20 | 1 |
| 2 | 3 | 9 | 1 | 3 | 2 | 18 | 1 | 3 | 10 | 13 | 1 | 3 | 24 | 0 | 0 | 4 | 7 | 15 | 1 | 4 | 13 | 18 | 1 | 4 | 20 | 1 | 1 | 5 | 8 | 20 | 1 | 5 | 18 | 23 | 1 | 5 | 24 | 8 | 1 | 6 | 23 | 21 | 1 |
| 2 | 3 | 10 | 1 | 3 | 3 | 5 | 1 | 3 | 10 | 14 | 1 | 3 | 25 | 0 | 0 | 4 | 7 | 16 | 1 | 4 | 14 | 5 | 1 | 4 | 20 | 2 | 1 | 5 | 8 | 21 | 1 | 5 | 18 | 24 | 1 | 5 | 24 | 11 | 1 | 6 | 23 | 22 | 1 |
| 2 | 4 | 1 | 1 | 3 | 3 | 6 | 1 | 3 | 10 | 15 | 1 | 3 | 26 | 0 | 0 | 4 | 7 | 17 | 1 | 4 | 14 | 6 | 1 | 4 | 20 | 3 | 1 | 5 | 8 | 22 | 1 | 5 | 18 | 25 | 1 | 5 | 24 | 12 | 1 | 6 | 23 | 23 | 1 |
| 2 | 4 | 2 | 1 | 3 | 3 | 7 | 1 | 3 | 10 | 16 | 1 | 4 | 1 | 19 | 1 | 4 | 7 | 18 | 1 | 4 | 14 | 7 | 1 | 4 | 20 | 4 | 1 | 5 | 8 | 23 | 1 | 5 | 18 | 26 | 1 | 5 | 24 | 13 | 1 | 6 | 23 | 24 | 1 |
| 2 | 4 | 3 | 0 | 3 | 3 | 8 | 1 | 3 | 10 | 17 | 1 | 4 | 1 | 20 | 1 | 4 | 8 | 5 | 1 | 4 | 14 | 8 | 1 | 4 | 20 | 9 | 1 | 5 | 8 | 24 | 1 | 5 | 19 | 5 | 1 | 5 | 24 | 14 | 1 | 6 | 23 | 25 | 1 |
| 2 | 4 | 4 | 0 | 3 | 3 | 11 | 1 | 3 | 10 | 18 | 1 | 4 | 1 | 21 | 1 | 4 | 8 | 6 | 1 | 4 | 14 | 11 | 1 | 4 | 20 | 10 | 1 | 5 | 8 | 25 | 1 | 5 | 19 | 6 | 1 | 5 | 24 | 15 | 1 | 6 | 23 | 26 | 1 |
| 2 | 4 | 9 | 1 | 3 | 3 | 12 | 1 | 3 | 11 | 1 | 0 | 4 | 1 | 22 | 1 | 4 | 8 | 7 | 1 | 4 | 14 | 12 | 1 | 4 | 21 | 1 | 1 | 5 | 8 | 26 | 1 | 5 | 19 | 7 | 1 | 5 | 24 | 16 | 1 | 6 | 24 | 19 | 1 |
| 2 | 4 | 10 | 1 | 3 | 3 | 13 | 1 | 3 | 11 | 2 | 1 | 4 | 1 | 23 | 1 | 4 | 8 | 8 | 1 | 4 | 14 | 13 | 1 | 4 | 21 | 2 | 1 | 5 | 11 | 19 | 1 | 5 | 19 | 8 | 1 | 5 | 24 | 17 | 1 | 6 | 24 | 20 | 1 |
| 2 | 9 | 1 | 1 | 3 | 3 | 14 | 1 | 3 | 11 | 3 | 1 | 4 | 1 | 24 | 1 | 4 | 8 | 11 | 1 | 4 | 14 | 14 | 1 | 4 | 21 | 3 | 1 | 5 | 11 | 20 | 1 | 5 | 19 | 11 | 1 | 5 | 24 | 18 | 1 | 6 | 24 | 21 | 1 |
| 2 | 9 | 2 | 1 | 3 | 3 | 15 | 1 | 3 | 11 | 4 | 1 | 4 | 1 | 25 | 1 | 4 | 8 | 12 | 1 | 4 | 14 | 15 | 1 | 4 | 21 | 4 | 1 | 5 | 11 | 21 | 1 | 5 | 19 | 12 | 1 | 5 | 25 | 5 | 1 | 6 | 24 | 22 | 1 |
| 2 | 9 | 3 | 0 | 3 | 3 | 16 | 1 | 3 | 11 | 9 | 1 | 4 | 1 | 26 | 1 | 4 | 8 | 13 | 1 | 4 | 14 | 16 | 1 | 4 | 21 | 9 | 1 | 5 | 11 | 22 | 1 | 5 | 19 | 13 | 1 | 5 | 25 | 6 | 1 | 6 | 24 | 23 | 1 |
| 2 | 9 | 4 | 0 | 3 | 3 | 17 | 1 | 3 | 11 | 10 | 1 | 4 | 2 | 19 | 1 | 4 | 8 | 14 | 1 | 4 | 14 | 17 | 1 | 4 | 21 | 10 | 1 | 5 | 11 | 23 | 1 | 5 | 19 | 14 | 1 | 5 | 25 | 7 | 1 | 6 | 24 | 24 | 1 |
| 2 | 9 | 9 | 1 | 3 | 3 | 18 | 1 | 3 | 12 | 1 | 1 | 4 | 2 | 20 | 1 | 4 | 8 | 15 | 1 | 4 | 14 | 18 | 1 | 4 | 22 | 1 | 1 | 5 | 11 | 24 | 1 | 5 | 19 | 15 | 1 | 5 | 25 | 8 | 1 | 6 | 24 | 25 | 1 |
| 2 | 9 | 10 | 1 | 3 | 4 | 5 | 1 | 3 | 12 | 2 | 0 | 4 | 2 | 21 | 1 | 4 | 8 | 16 | 1 | 4 | 15 | 5 | 1 | 4 | 22 | 2 | 1 | 5 | 11 | 25 | 1 | 5 | 19 | 16 | 1 | 5 | 25 | 11 | 1 | 6 | 24 | 26 | 1 |
| 2 | 10 | 1 | 1 | 3 | 4 | 6 | 1 | 3 | 12 | 3 | 1 | 4 | 2 | 22 | 1 | 4 | 8 | 17 | 1 | 4 | 15 | 6 | 1 | 4 | 22 | 3 | 1 | 5 | 11 | 26 | 1 | 5 | 19 | 17 | 1 | 5 | 25 | 12 | 1 | 6 | 25 | 19 | 1 |
| 2 | 10 | 2 | 1 | 3 | 4 | 7 | 1 | 3 | 12 | 4 | 1 | 4 | 2 | 23 | 1 | 4 | 8 | 18 | 1 | 4 | 15 | 7 | 1 | 4 | 22 | 4 | 1 | 5 | 12 | 19 | 1 | 5 | 19 | 18 | 1 | 5 | 25 | 13 | 1 | 6 | 25 | 20 | 1 |
| 2 | 10 | 3 | 0 | 3 | 4 | 8 | 1 | 3 | 12 | 9 | 1 | 4 | 2 | 24 | 1 | 4 | 9 | 19 | 1 | 4 | 15 | 8 | 1 | 4 | 22 | 9 | 1 | 5 | 12 | 20 | 1 | 5 | 20 | 5 | 1 | 5 | 25 | 14 | 1 | 6 | 25 | 21 | 1 |
| 2 | 10 | 4 | 0 | 3 | 4 | 11 | 1 | 3 | 12 | 10 | 1 | 4 | 2 | 25 | 1 | 4 | 9 | 20 | 1 | 4 | 15 | 11 | 1 | 4 | 22 | 10 | 1 | 5 | 12 | 21 | 1 | 5 | 20 | 6 | 1 | 5 | 25 | 15 | 1 | 6 | 25 | 22 | 1 |
| 2 | 10 | 9 | 1 | 3 | 4 | 12 | 1 | 3 | 13 | 1 | 1 | 4 | 2 | 26 | 1 | 4 | 9 | 21 | 1 | 4 | 15 | 12 | 1 | 4 | 23 | 1 | 1 | 5 | 12 | 22 | 1 | 5 | 20 | 7 | 1 | 5 | 25 | 16 | 1 | 6 | 25 | 23 | 1 |
| 2 | 10 | 10 | 1 | 3 | 4 | 13 | 1 | 3 | 13 | 2 | 1 | 4 | 3 | 19 | 1 | 4 | 9 | 22 | 1 | 4 | 15 | 13 | 1 | 4 | 23 | 2 | 1 | 5 | 12 | 23 | 1 | 5 | 20 | 8 | 1 | 5 | 25 | 17 | 1 | 6 | 25 | 24 | 1 |
| 2 | 0 | 5 | 0 | 3 | 4 | 14 | 1 | 3 | 13 | 3 | 1 | 4 | 3 | 20 | 1 | 4 | 9 | 23 | 1 | 4 | 15 | 14 | 1 | 4 | 23 | 3 | 1 | 5 | 12 | 24 | 1 | 5 | 20 | 11 | 1 | 5 | 25 | 18 | 1 | 6 | 25 | 25 | 1 |
| 2 | 0 | 6 | 0 | 3 | 4 | 15 | 1 | 3 | 13 | 4 | 1 | 4 | 3 | 21 | 1 | 4 | 9 | 24 | 1 | 4 | 15 | 15 | 1 | 4 | 23 | 4 | 1 | 5 | 12 | 25 | 1 | 5 | 20 | 12 | 1 | 5 | 26 | 5 | 1 | 6 | 25 | 26 | 1 |
| 2 | 0 | 7 | 0 | 3 | 4 | 16 | 1 | 3 | 13 | 9 | 1 | 4 | 3 | 22 | 1 | 4 | 9 | 25 | 1 | 4 | 15 | 16 | 1 | 4 | 23 | 9 | 1 | 5 | 12 | 26 | 1 | 5 | 20 | 13 | 1 | 5 | 26 | 6 | 1 | 6 | 26 | 19 | 1 |
| 2 | 0 | 8 | 0 | 3 | 4 | 17 | 1 | 3 | 13 | 10 | 1 | 4 | 3 | 23 | 1 | 4 | 9 | 26 | 1 | 4 | 15 | 17 | 1 | 4 | 23 | 10 | 1 | 5 | 13 | 19 | 1 | 5 | 20 | 14 | 1 | 5 | 26 | 7 | 1 | 6 | 26 | 20 | 1 |
| 2 | 0 | 11 | 0 | 3 | 4 | 18 | 1 | 3 | 14 | 1 | 1 | 4 | 3 | 24 | 1 | 4 | 10 | 19 | 1 | 4 | 15 | 18 | 1 | 4 | 24 | 1 | 1 | 5 | 13 | 20 | 1 | 5 | 20 | 15 | 1 | 5 | 26 | 8 | 1 | 6 | 26 | 21 | 1 |
| 2 | 0 | 12 | 0 | 3 | 5 | 1 | 1 | 3 | 14 | 2 | 1 | 4 | 3 | 25 | 1 | 4 | 10 | 20 | 1 | 4 | 16 | 5 | 1 | 4 | 24 | 2 | 1 | 5 | 13 | 21 | 1 | 5 | 20 | 16 | 1 | 5 | 26 | 11 | 1 | 6 | 26 | 22 | 1 |
| 2 | 0 | 13 | 0 | 3 | 5 | 2 | 1 | 3 | 14 | 3 | 1 | 4 | 3 | 26 | 1 | 4 | 10 | 21 | 1 | 4 | 16 | 6 | 1 | 4 | 24 | 3 | 1 | 5 | 13 | 22 | 1 | 5 | 20 | 17 | 1 | 5 | 26 | 12 | 1 | 6 | 26 | 23 | 1 |
| 2 | 0 | 14 | 0 | 3 | 5 | 3 | 1 | 3 | 14 | 4 | 1 | 4 | 4 | 19 | 1 | 4 | 10 | 22 | 1 | 4 | 16 | 7 | 1 | 4 | 24 | 4 | 1 | 5 | 13 | 23 | 1 | 5 | 20 | 18 | 1 | 5 | 26 | 13 | 1 | 6 | 26 | 24 | 1 |
| 2 | 0 | 15 | 0 | 3 | 5 | 4 | 1 | 3 | 14 | 9 | 1 | 4 | 4 | 20 | 1 | 4 | 10 | 23 | 1 | 4 | 16 | 8 | 1 | 4 | 24 | 9 | 1 | 5 | 13 | 24 | 1 | 5 | 21 | 5 | 1 | 5 | 26 | 14 | 1 | 6 | 26 | 25 | 1 |
| 2 | 0 | 16 | 0 | 3 | 5 | 9 | 1 | 3 | 14 | 10 | 1 | 4 | 4 | 21 | 1 | 4 | 10 | 24 | 1 | 4 | 16 | 11 | 1 | 4 | 24 | 10 | 1 | 5 | 13 | 25 | 1 | 5 | 21 | 6 | 1 | 5 | 26 | 15 | 1 | 6 | 26 | 26 | 1 |
| 2 | 0 | 17 | 0 | 3 | 5 | 10 | 1 | 3 | 15 | 1 | 1 | 4 | 4 | 22 | 1 | 4 | 10 | 25 | 1 | 4 | 16 | 12 | 1 | 4 | 25 | 1 | 1 | 5 | 13 | 26 | 1 | 5 | 21 | 7 | 1 | 5 | 26 | 16 | 1 | | | | |
| 2 | 0 | 18 | 0 | 3 | 6 | 1 | 1 | 3 | 15 | 2 | 1 | 4 | 4 | 23 | 1 | 4 | 10 | 26 | 1 | 4 | 16 | 13 | 1 | 4 | 25 | 2 | 1 | 5 | 14 | 19 | 1 | 5 | 21 | 8 | 1 | 5 | 26 | 17 | 1 | | | | |
| 2 | 5 | 0 | 0 | 3 | 6 | 2 | 1 | 3 | 15 | 3 | 1 | 4 | 4 | 24 | 1 | 4 | 11 | 5 | 1 | 4 | 16 | 14 | 1 | 4 | 25 | 3 | 1 | 5 | 14 | 20 | 1 | 5 | 21 | 11 | 1 | 5 | 26 | 18 | 1 | | | | |
| 2 | 6 | 0 | 0 | 3 | 6 | 3 | 1 | 3 | 15 | 4 | 1 | 4 | 4 | 25 | 1 | 4 | 11 | 6 | 1 | 4 | 16 | 15 | 1 | 4 | 25 | 4 | 1 | 5 | 14 | 21 | 1 | 5 | 21 | 12 | 1 | 6 | 19 | 19 | 1 | | | | |
| 2 | 7 | 0 | 0 | 3 | 6 | 4 | 1 | 3 | 15 | 9 | 1 | 4 | 4 | 26 | 1 | 4 | 11 | 7 | 1 | 4 | 16 | 16 | 1 | 4 | 25 | 9 | 1 | 5 | 14 | 22 | 1 | 5 | 21 | 13 | 1 | 6 | 19 | 20 | 1 | | | | |
| 2 | 8 | 0 | 0 | 3 | 6 | 9 | 1 | 3 | 15 | 10 | 1 | 4 | 5 | 5 | 1 | 4 | 11 | 8 | 1 | 4 | 16 | 17 | 1 | 4 | 25 | 10 | 1 | 5 | 14 | 23 | 1 | 5 | 21 | 14 | 1 | 6 | 19 | 21 | 1 | | | | |
| 2 | 11 | 0 | 0 | 3 | 6 | 10 | 1 | 3 | 16 | 1 | 1 | 4 | 5 | 6 | 1 | 4 | 11 | 11 | 1 | 4 | 16 | 18 | 1 | 4 | 26 | 1 | 1 | 5 | 14 | 24 | 1 | 5 | 21 | 15 | 1 | 6 | 19 | 22 | 1 | | | | |
| 2 | 12 | 0 | 0 | 3 | 7 | 1 | 1 | 3 | 16 | 2 | 1 | 4 | 5 | 7 | 1 | 4 | 11 | 12 | 1 | 4 | 17 | 5 | 1 | 4 | 26 | 2 | 1 | 5 | 14 | 25 | 1 | 5 | 21 | 16 | 1 | 6 | 19 | 23 | 1 | | | | |

*Table 8.* Complete enumeration of all $27 \times 27$ motion primitive combinations. The table systematically maps translation IDs to rotation IDs with corresponding difficulty levels. Validity of each combination is indicated by checkmarks for valid pairs and crosses for invalid ones. The layout presents 67 combinations per column with 11 columns per page.

## B.3. Existing Keyboard and Mouse Support Space

This subsection presents the complete table for the Existing Keyboard and Mouse Support Space.Based on the motions with defined keys in Table 7, Table 9 contains a total of 81 motions. The difficulty range spans from 1 to 4. Similarly, the state of complete stationary motion is specifically defined with a difficulty of 1.

| Diff. | T-ID | Translation | R-ID | Rotation | Keys | Valid | T | R | Description |
|---|---|---|---|---|---|---|---|---|---|
| 1 | 0 | Stationary | 0 | Stationary | · | 1 | [0, 0, 0, 0] | [0.0, 0.0] | The camera's movement direction remains stationary (·). At the same time, the rotation direction of the camera remains stationary (·). |
| 1 | 0 | Stationary | 1 | Camera Up | ↑ | 1 | [0, 0, 0, 0] | [1.0, 0.0] | The camera tilts up (↑). |
| 1 | 0 | Stationary | 2 | Camera Down | ↓ | 1 | [0, 0, 0, 0] | [-1.0, 0.0] | The camera tilts down (↓). |
| 1 | 0 | Stationary | 3 | Camera Right | → | 1 | [0, 0, 0, 0] | [0.0, 1.0] | The camera pans to the right (→). |
| 1 | 0 | Stationary | 4 | Camera Left | ← | 1 | [0, 0, 0, 0] | [0.0, -1.0] | The camera pans to the left (←). |
| 1 | 1 | Forward | 0 | Stationary | W | 1 | [1, 0, 0, 0] | [0.0, 0.0] | The camera pushes forward (W). |
| 1 | 2 | Backward | 0 | Stationary | S | 1 | [0, 1, 0, 0] | [0.0, 0.0] | The camera pulls back (S). |
| 1 | 3 | Left | 0 | Stationary | A | 1 | [0, 0, 1, 0] | [0.0, 0.0] | The camera moves to the left (A). |
| 1 | 4 | Right | 0 | Stationary | D | 1 | [0, 0, 0, 1] | [0.0, 0.0] | The camera moves to the right (D). |
| 2 | 0 | Stationary | 5 | Camera Up+Right | ↑+→ | 1 | [0, 0, 0, 0] | [1.0, 1.0] | The camera tilts up and pans to the right (↑→). |
| 2 | 0 | Stationary | 6 | Camera Up+Left | ↑+← | 1 | [0, 0, 0, 0] | [1.0, -1.0] | The camera tilts up and pans to the left (↑←). |
| 2 | 0 | Stationary | 7 | Camera Down+Right | ↓+→ | 1 | [0, 0, 0, 0] | [-1.0, 1.0] | The camera tilts down and pans to the right (↓→). |
| 2 | 0 | Stationary | 8 | Camera Down+Left | ↓+← | 1 | [0, 0, 0, 0] | [-1.0, -1.0] | The camera tilts down and pans to the left (↓←). |
| 2 | 5 | Forward+Left | 0 | Stationary | W+A | 1 | [1, 0, 1, 0] | [0.0, 0.0] | The camera pushes forward and moves to the left (W+A). |
| 2 | 6 | Forward+Right | 0 | Stationary | W+D | 1 | [1, 0, 0, 1] | [0.0, 0.0] | The camera pushes forward and moves to the right (W+D). |
| 2 | 7 | Backward+Left | 0 | Stationary | S+A | 1 | [0, 1, 1, 0] | [0.0, 0.0] | The camera pulls back and moves to the left (S+A). |
| 2 | 8 | Backward+Right | 0 | Stationary | S+D | 1 | [0, 1, 0, 1] | [0.0, 0.0] | The camera pulls back and moves to the right (S+D). |
| 2 | 1 | Forward | 1 | Camera Up | W+↑ | 1 | [1, 0, 0, 0] | [1.0, 0.0] | The camera pushes forward (W). At the same time, the camera tilts up (↑). |
| 2 | 1 | Forward | 2 | Camera Down | W+↓ | 0 | [1, 0, 0, 0] | [-1.0, 0.0] | The camera pushes forward (W). At the same time, the camera tilts down (↓). |
| 2 | 1 | Forward | 3 | Camera Right | W+→ | 1 | [1, 0, 0, 0] | [0.0, 1.0] | The camera pushes forward (W). At the same time, the camera pans to the right (→). |
| 2 | 1 | Forward | 4 | Camera Left | W+← | 1 | [1, 0, 0, 0] | [0.0, -1.0] | The camera pushes forward (W). At the same time, the camera pans to the left (←). |
| 2 | 2 | Backward | 1 | Camera Up | S+↑ | 0 | [0, 1, 0, 0] | [1.0, 0.0] | The camera pulls back (S). At the same time, the camera tilts up (↑). |
| 2 | 2 | Backward | 2 | Camera Down | S+↓ | 0 | [0, 1, 0, 0] | [-1.0, 0.0] | The camera pulls back (S). At the same time, the camera tilts down (↓). |
| 2 | 2 | Backward | 3 | Camera Right | S+→ | 1 | [0, 1, 0, 0] | [0.0, 1.0] | The camera pulls back (S). At the same time, the camera pans to the right (→). |
| 2 | 2 | Backward | 4 | Camera Left | S+← | 1 | [0, 1, 0, 0] | [0.0, -1.0] | The camera pulls back (S). At the same time, the camera pans to the left (←). |
| 2 | 3 | Left | 1 | Camera Up | A+↑ | 0 | [0, 0, 1, 0] | [1.0, 0.0] | The camera moves to the left (A). At the same time, the camera tilts up (↑). |
| 2 | 3 | Left | 2 | Camera Down | A+↓ | 0 | [0, 0, 1, 0] | [-1.0, 0.0] | The camera moves to the left (A). At the same time, the camera tilts down (↓). |
| 2 | 3 | Left | 3 | Camera Right | A+→ | 1 | [0, 0, 1, 0] | [0.0, 1.0] | The camera moves to the left (A). At the same time, the camera pans to the right (→). |
| 2 | 3 | Left | 4 | Camera Left | A+← | 1 | [0, 0, 1, 0] | [0.0, -1.0] | The camera moves to the left (A). At the same time, the camera pans to the left (←). |
| 2 | 4 | Right | 1 | Camera Up | D+↑ | 0 | [0, 0, 0, 1] | [1.0, 0.0] | The camera moves to the right (D). At the same time, the camera tilts up (↑). |
| 2 | 4 | Right | 2 | Camera Down | D+↓ | 0 | [0, 0, 0, 1] | [-1.0, 0.0] | The camera moves to the right (D). At the same time, the camera tilts down (↓). |
| 2 | 4 | Right | 3 | Camera Right | D+→ | 1 | [0, 0, 0, 1] | [0.0, 1.0] | The camera moves to the right (D). At the same time, the camera pans to the right (→). |
| 2 | 4 | Right | 4 | Camera Left | D+← | 1 | [0, 0, 0, 1] | [0.0, -1.0] | The camera moves to the right (D). At the same time, the camera pans to the left (←). |
| 3 | 1 | Forward | 5 | Camera Up+Right | W+↑+→ | 1 | [1, 0, 0, 0] | [1.0, 1.0] | The camera pushes forward (W). At the same time, the camera tilts up and pans to the right (↑→). |
| 3 | 1 | Forward | 6 | Camera Up+Left | W+↑+← | 1 | [1, 0, 0, 0] | [1.0, -1.0] | The camera pushes forward (W). At the same time, the camera tilts up and pans to the left (↑←). |
| 3 | 1 | Forward | 7 | Camera Down+Right | W+↓+→ | 0 | [1, 0, 0, 0] | [-1.0, 1.0] | The camera pushes forward (W). At the same time, the camera tilts down and pans to the right (↓→). |
| 3 | 1 | Forward | 8 | Camera Down+Left | W+↓+← | 0 | [1, 0, 0, 0] | [-1.0, -1.0] | The camera pushes forward (W). At the same time, the camera tilts down and pans to the left (↓←). |
| 3 | 2 | Backward | 5 | Camera Up+Right | S+↑+→ | 0 | [0, 1, 0, 0] | [1.0, 1.0] | The camera pulls back (S). At the same time, the camera tilts up and pans to the right (↑→). |
| 3 | 2 | Backward | 6 | Camera Up+Left | S+↑+← | 0 | [0, 1, 0, 0] | [1.0, -1.0] | The camera pulls back (S). At the same time, the camera tilts up and pans to the left (↑←). |
| 3 | 2 | Backward | 7 | Camera Down+Right | S+↓+→ | 0 | [0, 1, 0, 0] | [-1.0, 1.0] | The camera pulls back (S). At the same time, the camera tilts down and pans to the right (↓→). |
| 3 | 2 | Backward | 8 | Camera Down+Left | S+↓+← | 0 | [0, 1, 0, 0] | [-1.0, -1.0] | The camera pulls back (S). At the same time, the camera tilts down and pans to the left (↓←). |
| 3 | 3 | Left | 5 | Camera Up+Right | A+↑+→ | 0 | [0, 0, 1, 0] | [1.0, 1.0] | The camera moves to the left (A). At the same time, the camera tilts up and pans to the right (↑→). |
| 3 | 3 | Left | 6 | Camera Up+Left | A+↑+← | 0 | [0, 0, 1, 0] | [1.0, -1.0] | The camera moves to the left (A). At the same time, the camera tilts up and pans to the left (↑←). |
| 3 | 3 | Left | 7 | Camera Down+Right | A+↓+→ | 0 | [0, 0, 1, 0] | [-1.0, 1.0] | The camera moves to the left (A). At the same time, the camera tilts down and pans to the right (↓→). |
| 3 | 3 | Left | 8 | Camera Down+Left | A+↓+← | 0 | [0, 0, 1, 0] | [-1.0, -1.0] | The camera moves to the left (A). At the same time, the camera tilts down and pans to the left (↓←). |
| 3 | 4 | Right | 5 | Camera Up+Right | D+↑+→ | 0 | [0, 0, 0, 1] | [1.0, 1.0] | The camera moves to the right (D). At the same time, the camera tilts up and pans to the right (↑→). |
| 3 | 4 | Right | 6 | Camera Up+Left | D+↑+← | 0 | [0, 0, 0, 1] | [1.0, -1.0] | The camera moves to the right (D). At the same time, the camera tilts up and pans to the left (↑←). |
| 3 | 4 | Right | 7 | Camera Down+Right | D+↓+→ | 0 | [0, 0, 0, 1] | [-1.0, 1.0] | The camera moves to the right (D). At the same time, the camera tilts down and pans to the right (↓→). |
| 3 | 4 | Right | 8 | Camera Down+Left | D+↓+← | 0 | [0, 0, 0, 1] | [-1.0, -1.0] | The camera moves to the right (D). At the same time, the camera tilts down and pans to the left (↓←). |
| 3 | 5 | Forward+Left | 1 | Camera Up | W+A+↑ | 0 | [1, 0, 1, 0] | [1.0, 0.0] | The camera pushes forward and moves to the left (W+A). At the same time, the camera tilts up (↑). |
| 3 | 5 | Forward+Left | 2 | Camera Down | W+A+↓ | 0 | [1, 0, 1, 0] | [-1.0, 0.0] | The camera pushes forward and moves to the left (W+A). At the same time, the camera tilts down (↓). |
| 3 | 5 | Forward+Left | 3 | Camera Right | W+A+→ | 1 | [1, 0, 1, 0] | [0.0, 1.0] | The camera pushes forward and moves to the left (W+A). At the same time, the camera pans to the right (→). |
| 3 | 5 | Forward+Left | 4 | Camera Left | W+A+← | 1 | [1, 0, 1, 0] | [0.0, -1.0] | The camera pushes forward and moves to the left (W+A). At the same time, the camera pans to the left (←). |
| 3 | 6 | Forward+Right | 1 | Camera Up | W+D+↑ | 0 | [1, 0, 0, 1] | [1.0, 0.0] | The camera pushes forward and moves to the right (W+D). At the same time, the camera tilts up (↑). |
| 3 | 6 | Forward+Right | 2 | Camera Down | W+D+↓ | 0 | [1, 0, 0, 1] | [-1.0, 0.0] | The camera pushes forward and moves to the right (W+D). At the same time, the camera tilts down (↓). |
| 3 | 6 | Forward+Right | 3 | Camera Right | W+D+→ | 1 | [1, 0, 0, 1] | [0.0, 1.0] | The camera pushes forward and moves to the right (W+D). At the same time, the camera pans to the right (→). |
| 3 | 6 | Forward+Right | 4 | Camera Left | W+D+← | 1 | [1, 0, 0, 1] | [0.0, -1.0] | The camera pushes forward and moves to the right (W+D). At the same time, the camera pans to the left (←). |
| 3 | 7 | Backward+Left | 1 | Camera Up | S+A+↑ | 0 | [0, 1, 1, 0] | [1.0, 0.0] | The camera pulls back and moves to the left (S+A). At the same time, the camera tilts up (↑). |
| 3 | 7 | Backward+Left | 2 | Camera Down | S+A+↓ | 0 | [0, 1, 1, 0] | [-1.0, 0.0] | The camera pulls back and moves to the left (S+A). At the same time, the camera tilts down (↓). |
| 3 | 7 | Backward+Left | 3 | Camera Right | S+A+→ | 1 | [0, 1, 1, 0] | [0.0, 1.0] | The camera pulls back and moves to the left (S+A). At the same time, the camera pans to the right (→). |
| 3 | 7 | Backward+Left | 4 | Camera Left | S+A+← | 1 | [0, 1, 1, 0] | [0.0, -1.0] | The camera pulls back and moves to the left (S+A). At the same time, the camera pans to the left (←). |
| 3 | 8 | Backward+Right | 1 | Camera Up | S+D+↑ | 0 | [0, 1, 0, 1] | [1.0, 0.0] | The camera pulls back and moves to right (S+D). At the same time, the camera tilts up (↑). |
| 3 | 8 | Backward+Right | 2 | Camera Down | S+D+↓ | 0 | [0, 1, 0, 1] | [-1.0, 0.0] | The camera pulls back and moves to right (S+D). At the same time, the camera tilts down (↓). |
| 3 | 8 | Backward+Right | 3 | Camera Right | S+D+→ | 1 | [0, 1, 0, 1] | [0.0, 1.0] | The camera pulls back and moves to right (S+D). At the same time, the camera pans to the right (→). |
| 3 | 8 | Backward+Right | 4 | Camera Left | S+D+← | 1 | [0, 1, 0, 1] | [0.0, -1.0] | The camera pulls back and moves to right (S+D). At the same time, the camera pans to the left (←). |
| 4 | 5 | Forward+Left | 5 | Camera Up+Right | W+A+↑+→ | 0 | [1, 0, 1, 0] | [1.0, 1.0] | The camera pushes forward and moves to the left (W+A). At the same time, the camera tilts up and pans to the right (↑→). |
| 4 | 5 | Forward+Left | 6 | Camera Up+Left | W+A+↑+← | 0 | [1, 0, 1, 0] | [1.0, -1.0] | The camera pushes forward and moves to the left (W+A). At the same time, the camera tilts up and pans to the left (↑←). |
| 4 | 5 | Forward+Left | 7 | Camera Down+Right | W+A+↓+→ | 0 | [1, 0, 1, 0] | [-1.0, 1.0] | The camera pushes forward and moves to the left (W+A). At the same time, the camera tilts down and pans to the right (↓→). |
| 4 | 5 | Forward+Left | 8 | Camera Down+Left | W+A+↓+← | 0 | [1, 0, 1, 0] | [-1.0, -1.0] | The camera pushes forward and moves to the left (W+A). At the same time, the camera tilts down and pans to the left (↓←). |
| 4 | 6 | Forward+Right | 5 | Camera Up+Right | W+D+↑+→ | 0 | [1, 0, 0, 1] | [1.0, 1.0] | The camera pushes forward and moves to the right (W+D). At the same time, the camera tilts up and pans to the right (↑→). |
| 4 | 6 | Forward+Right | 6 | Camera Up+Left | W+D+↑+← | 0 | [1, 0, 0, 1] | [1.0, -1.0] | The camera pushes forward and moves to the right (W+D). At the same time, the camera tilts up and pans to the left (↑←). |
| 4 | 6 | Forward+Right | 7 | Camera Down+Right | W+D+↓+→ | 0 | [1, 0, 0, 1] | [-1.0, 1.0] | The camera pushes forward and moves to the right (W+D). At the same time, the camera tilts down and pans to the right (↓→). |
| 4 | 6 | Forward+Right | 8 | Camera Down+Left | W+D+↓+← | 0 | [1, 0, 0, 1] | [-1.0, -1.0] | The camera pushes forward and moves to the right (W+D). At the same time, the camera tilts down and pans to the left (↓←). |
| 4 | 7 | Backward+Left | 5 | Camera Up+Right | S+A+↑+→ | 0 | [0, 1, 1, 0] | [1.0, 1.0] | The camera pulls back and moves to the left (S+A). At the same time, the camera tilts up and pans to the right (↑→). |
| 4 | 7 | Backward+Left | 6 | Camera Up+Left | S+A+↑+← | 0 | [0, 1, 1, 0] | [1.0, -1.0] | The camera pulls back and moves to the left (S+A). At the same time, the camera tilts up and pans to the left (↑←). |
| 4 | 7 | Backward+Left | 7 | Camera Down+Right | S+A+↓+→ | 0 | [0, 1, 1, 0] | [-1.0, 1.0] | The camera pulls back and moves to the left (S+A). At the same time, the camera tilts down and pans to the right (↓→). |
| 4 | 7 | Backward+Left | 8 | Camera Down+Left | S+A+↓+← | 0 | [0, 1, 1, 0] | [-1.0, -1.0] | The camera pulls back and moves to the left (S+A). At the same time, the camera tilts down and pans to the left (↓←). |
| 4 | 8 | Backward+Right | 5 | Camera Up+Right | S+D+↑+→ | 0 | [0, 1, 0, 1] | [1.0, 1.0] | The camera pulls back and moves to the right (S+D). At the same time, the camera tilts up and pans to the right (↑→). |
| 4 | 8 | Backward+Right | 6 | Camera Up+Left | S+D+↑+← | 0 | [0, 1, 0, 1] | [1.0, -1.0] | The camera pulls back and moves to the right (S+D). At the same time, the camera tilts up and pans to the left (↑←). |
| 4 | 8 | Backward+Right | 7 | Camera Down+Right | S+D+↓+→ | 0 | [0, 1, 0, 1] | [-1.0, 1.0] | The camera pulls back and moves to the right (S+D). At the same time, the camera tilts down and pans to the right (↓→). |
| 4 | 8 | Backward+Right | 8 | Camera Down+Left | S+D+↓+← | 0 | [0, 1, 0, 1] | [-1.0, -1.0] | The camera pulls back and moves to the right (S+D). At the same time, the camera tilts down and pans to the left (↓←). |

*Table 9.* These combinations in this table are derived from 9 keyboard-operable translation actions and 9 keyboard-operable rotation actions. Each combination pairs translation and rotation IDs with corresponding difficulty levels. Feasibility is indicated by checkmarks for valid pairs and crosses for invalid ones. Keyboard controls are provided where applicable.

This subsection presents the definition Table 10 for the Mapping of the Memory module. We define the Memory Mapping Table by designing inverses for six corresponding motion patterns of translation and rotation. Based on this table, we design a memory detection experiment to verify the memorability of the generated content.The parts shaded in gray represent cases without corresponding keys.

| ID | Action 1 | | | Action 2 | | | Description |
|---|---|---|---|---|---|---|---|
| | **Encoding** | **Direction** | **Keys** | **Encoding** | **Direction** | **Keys** | |
| 1 | **keyboard:** [1,0,0,0] **mouse:** [0.0,0.0] | Forward | W | **keyboard:** [0,1,0,0] **mouse:** [0.0,0.0] | Backward | S | First, The camera pushes forward (W). Second, The camera pulls back (S). Third, ensure the magnitude of motion before and after is the same. |
| 2 | **keyboard:** [0,1,0,0] **mouse:** [0.0,0.0] | Backward | S | **keyboard:** [1,0,0,0] **mouse:** [0.0,0.0] | Forward | W | First, The camera pulls back (S). Second, The camera pushes forward (W). Third, ensure the magnitude of motion before and after is the same. |
| 3 | **keyboard:** [0,0,1,0] **mouse:** [0.0,0.0] | Left | A | **keyboard:** [0,0,0,1] **mouse:** [0.0,0.0] | Right | D | First, The camera moves to the left (A). Second, The camera moves to the right (D). Third, ensure the magnitude of motion before and after is the same. |
| 4 | **keyboard:** [0,0,0,1] **mouse:** [0.0,0.0] | Right | D | **keyboard:** [0,0,1,0] **mouse:** [0.0,0.0] | Left | A | First, The camera moves to the right (D). Second, The camera moves to the left (A). Third, ensure the magnitude of motion before and after is the same. |
| 5 | **keyboard:** [0,0,0,0] **mouse:** [1.0,0.0] | Camera Up | ↑ | **keyboard:** [0,0,0,0] **mouse:** [-1.0,0.0] | Camera Down | ↓ | First, The camera tilts up (↑). Second, The camera tilts down (↓). Third, ensure the magnitude of motion before and after is the same. |
| 6 | **keyboard:** [0,0,0,0] **mouse:** [-1.0,0.0] | Camera Down | ↓ | **keyboard:** [0,0,0,0] **mouse:** [1.0,0.0] | Camera Up | ↑ | First, The camera tilts down (↓). Second, The camera tilts up (↑). Third, ensure the magnitude of motion before and after is the same. |
| 7 | **keyboard:** [0,0,0,0] **mouse:** [0.0,-1.0] | Camera Left | ← | **keyboard:** [0,0,0,0] **mouse:** [0.0,1.0] | Camera Right | → | First, The camera pans to the left (←). Second, The camera pans to the right (→). Third, ensure the magnitude of motion before and after is the same. |
| 8 | **keyboard:** [0,0,0,0] **mouse:** [0.0,1.0] | Camera Right | → | **keyboard:** [0,0,0,0] **mouse:** [0.0,-1.0] | Camera Left | ← | First, The camera pans to the right (→). Second, The camera pans to the left (←). Third, ensure the magnitude of motion before and after is the same. |
| 9 | **keyboard:** [0,0,0,0] **mouse:** [1.0,0.0] | Upward | - | **keyboard:** [0,0,0,0] **mouse:** [-1.0,0.0] | Downward | - | First, The camera tilts up (↑). Second, The camera tilts down (↓). Third, ensure the magnitude of motion before and after is the same. |
| 10 | **keyboard:** [0,0,0,0] **mouse:** [-1.0,0.0] | Downward | - | **keyboard:** [0,0,0,0] **mouse:** [1.0,0.0] | Upward | - | First, The camera tilts down (↓). Second, The camera tilts up (↑). Third, ensure the magnitude of motion before and after is the same. |

*Table 10.* Memory Mapping Table. This table defines all recallable action spaces through the definition of "back-and-forth" actions. It is used to control and guide the generation of trajectories with memory.

# C. Benchmark Details

## C.1. Data presentation of different difficulty

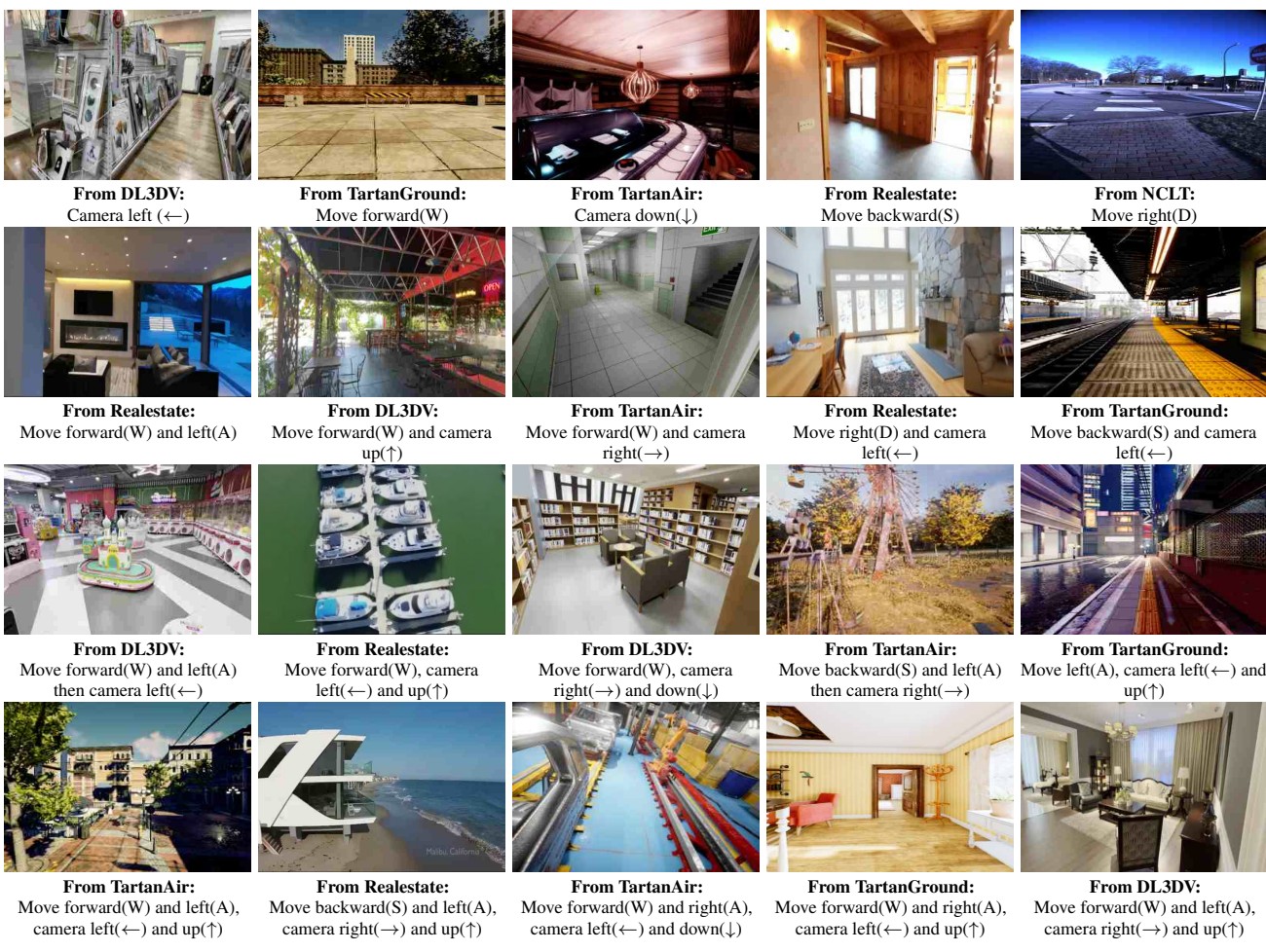

*Figure 5.* A detailed showcase of the Difference Verification task across four difficulty levels (rows 1 to 4), based on camera operation complexity: Row 1 shows basic single-axis movements; Row 2 adds combined translation and rotation; Row 3 features sequential composite trajectories; and Row 4 involves complex multi-axis movements with view changes. These examples demonstrate the model's ability to detect subtle pose differences across various scenarios.

## C.2. Data presentation of memory

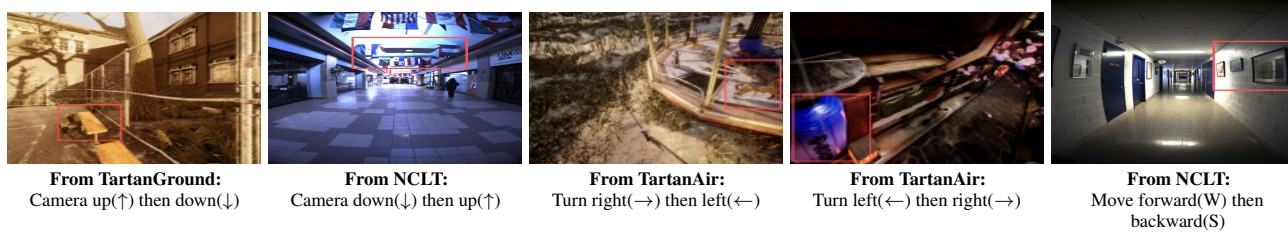

| From TartanGround: | From NCLT: | From TartanAir: | From TartanAir: | From NCLT: |
| Camera up($\uparrow$) then down($\downarrow$) | Camera down($\downarrow$) then up($\uparrow$) | Turn right($\rightarrow$) then left($\leftarrow$) | Turn left($\leftarrow$) then right($\rightarrow$) | Move forward(W) then backward(S) |

*Figure 6.* Detailed showcase of the Memory Verification task focusing on loop closure difficulty. This figure illustrates memory-dependent trajectories where the camera performs reversible actions (e.g., "up then down" or "turn right then left"), requiring the model to recall the initial state to verify the loop closure. The red bounding boxes highlight key visual cues used for temporal reasoning and consistency checks in long-term memory scenarios.

# D. Metrics details

**1. Image Quality** ($S_{\text{Image}}$). This metric assesses low-level visual distortions such as overexposure, noise, or blur in generated video frames. We adopt the MUSIQ quality prediction model, leveraging its ability to perceive diverse resolutions and aspect ratios to score each frame in the video sequence. The final score is obtained by calculating the arithmetic mean of the frame-level scores across the entire sequence and performing linear normalization. This metric objectively reflects the underlying fidelity of the world model in rendering visual details.

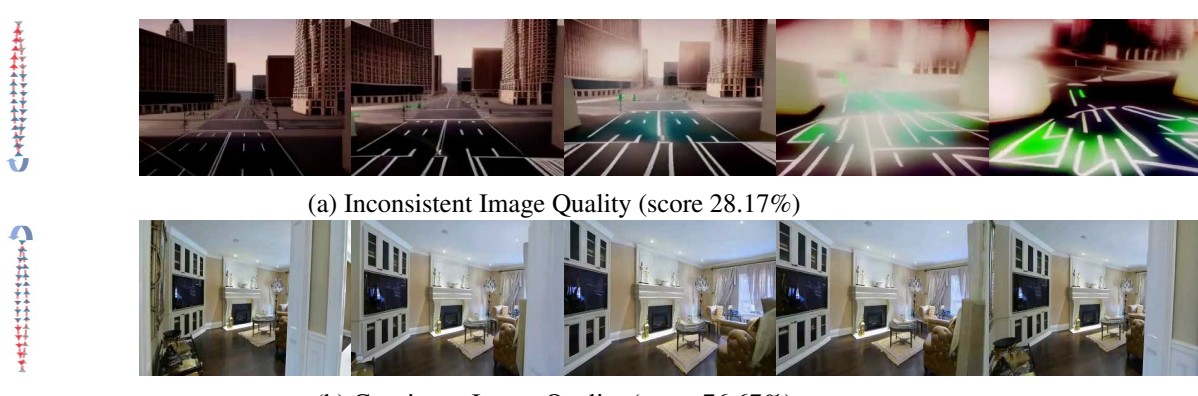

(a) Inconsistent Image Quality (score 28.17%)

(b) Consistent Image Quality (score 76.67%)

*Figure 7.* **Visualization of Subject Consistency of Image Quality.** Image quality measures the memory symmetry. when the model execute memory tasks. (a) has a clear distortion. (b) shows a better color image quality consistency.

**2. Brightness Consistency** ($S_{\text{Brightness}}$). This metric aims to evaluate the stability of the brightness distribution in generated videos. We categorize pixel grayscales into three levels (dark, mid, bright) to construct a 3D brightness distribution vector $\mathbf{v}_t = [p_{\text{dark}}, p_{\text{mid}}, p_{\text{bright}}]^\top$ for each frame. The comprehensive score is derived by calculating the similarity between the $t$-th frame and the initial frame $\mathbf{v}_1$, applying a modified Softmax transform (first use, for single-sample score enhancement) and normalized exponential decay weight $w(d)$. Here, the similarity $\mathcal{S}(\cdot, \cdot)$ is defined as the cosine inner product (first use):

$$\mathcal{S}(\mathbf{a}, \mathbf{b}) = \frac{\mathbf{a} \cdot \mathbf{b}}{\|\mathbf{a}\|\|\mathbf{b}\|} = \frac{\sum_{i=1}^{n} a_i b_i}{\sqrt{\sum_{i=1}^{n} a_i^2}\sqrt{\sum_{i=1}^{n} b_i^2}} \tag{3}$$

where $\mathbf{a}, \mathbf{b}$ are $n$-dimensional vectors, $\mathbf{a} \cdot \mathbf{b}$ denotes their inner product, and $\|\mathbf{a}\|, \|\mathbf{b}\|$ are their Euclidean norms.
The modified Softmax transform (for single-sample score $x \in [0, 1]$, high-score segment enhancement) is:

$$\mathcal{T}(x) = \frac{e^{\lambda x} - 1}{e^{\lambda} - 1} \quad (\lambda > 0) \tag{4}$$

where $\lambda$ controls the distinction degree: the larger $\lambda$ is, the steeper the high-score segment ($x \rightarrow 1$) and the flatter the low-score segment ($x \rightarrow 0$), which can amplify the sensitivity to brightness drift.
The temporal weight adopts a normalized exponential decay algorithm (first use, frame index $i \in [1, T-1]$, frame distance $d = i$):

$$w(d) = e^{-\alpha \cdot d}, \quad \hat{w}(d) = \frac{w(d)}{\sum_{k=1}^{T-1} w(k)} \quad (\alpha > 0) \tag{5}$$

where $\alpha$ is the decay coefficient (the larger $\alpha$ is, the faster the weight decays), $w(d)$ is the original exponential decay weight, and $\hat{w}(d)$ is the normalized weight (sum to 1, avoiding scaling inconsistency caused by different frame numbers). The comprehensive score formula is as follows:

$$S_{\text{Brightness}} = \sum_{d=1}^{T-1} \hat{w}(d) \cdot \mathcal{T}(\mathcal{S}(\mathbf{v}_{d+1}, \mathbf{v}_1)) \tag{6}$$

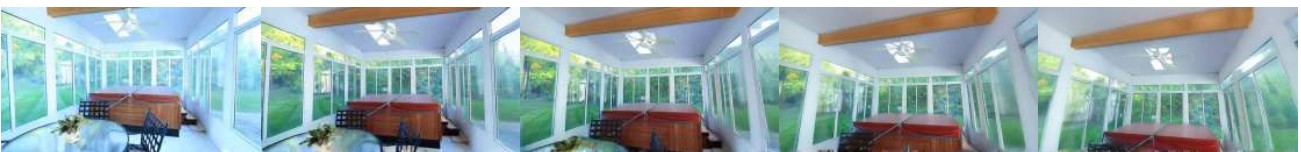

(a) Inconsistent Brightness (score 0.32%)

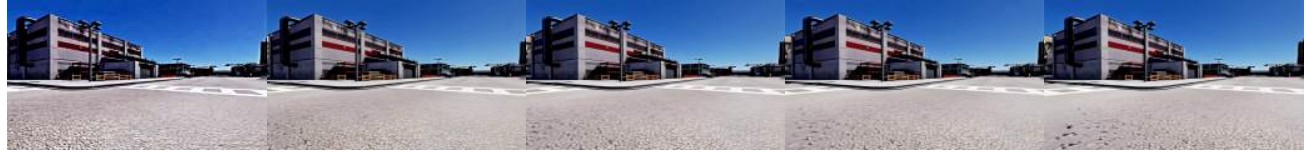

(b) Consistent Brightness (score 99.65%)

*Figure 8.* **Visualization of Subject Consistency of Brightness.** We defined brightness consistency to measure the performance of light comprehensive of models. (a) has a obvious change while (b) hold great consistency throughout.

The three-level classification mechanism reserves space for reasonable brightness changes, while the comparison mechanism with the initial frame effectively monitors style collapse and brightness drift in long-sequence generation.

**3. Color Temperature Constraint** ($S_{\mathbf{Color}}$). To evaluate the consistency of the environmental atmosphere, we analyze the Hue dimension in the HSV color space. The hue spectrum (0-179) is divided into 7 core intervals to construct a hue feature vector $\mathbf{h}_t \in \mathbb{R}^7$. We calculate the weighted perceptual similarity of the entire sequence relative to the initial frame. To balance global constancy and local transition, the similarity for each frame is defined as the average of the current-to-first and current-to-previous frame similarities (the similarity $\mathcal{S}(\cdot, \cdot)$ follows the cosine inner product definition in Eq. 3):

$$\bar{\mathcal{S}}_t = \frac{\mathcal{S}(\mathbf{h}_t, \mathbf{h}_1) + \mathcal{S}(\mathbf{h}_t, \mathbf{h}_{t-1})}{2} \tag{7}$$

The final score is calculated using a normalized exponentially decaying weighted sum, where the weight $\hat{w}'(d)$ (first use, improved version of $\hat{w}(d)$, focusing more on distant frames) and modified Softmax transform (Eq. 4) are adopted, and its algorithm formula is:

$$w'(d) = e^{-\beta \cdot d}, \quad \hat{w}'(d) = \frac{w'(d)}{\sum_{k=1}^{T-1} w'(k)} \quad (\beta > \alpha > 0) \tag{8}$$

Here, $\beta > \alpha$ ensures that $\hat{w}'(d)$ decays faster than $\hat{w}(d)$, so as to severely penalize the "color drift" phenomenon common in long videos, ensuring the consistency of the scene's color temperature logic:

$$S_{\mathbf{Color}} = \sum_{d=1}^{T-1} \hat{w}'(d) \cdot \mathcal{T}(\bar{\mathcal{S}}_{d+1}) \tag{9}$$

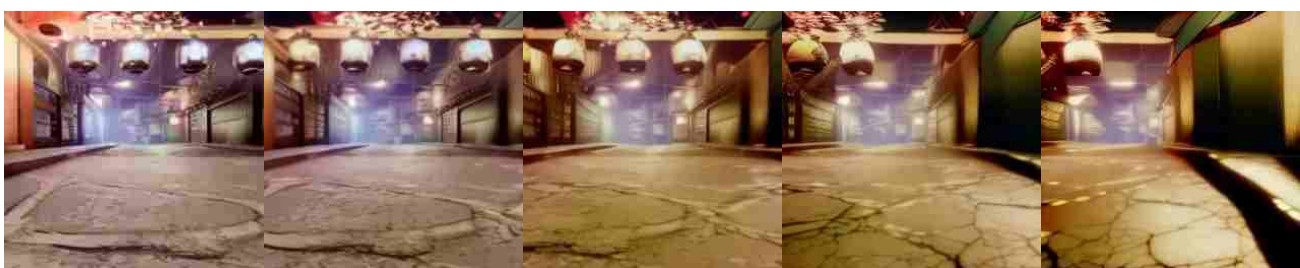

(a) Inconsistent Color Temperature (score 10.1%)

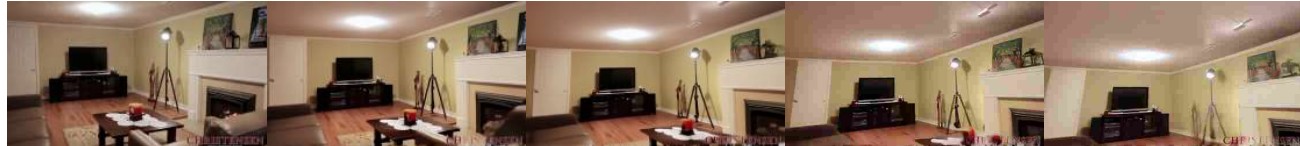

(b) Consistent Color Temperature (score 97.49%)

*Figure 9.* **Visualization of Subject Consistency of Color Temperature.** We use color temperature consistency to measure the color of pictures. (a) changes from a cold tone to a warm tone. (b) has a better color temperature consistency.

**4. Sharpness Retention** ($S_{\mathbf{Sharpness}}$). This metric evaluates the stability of details by monitoring the evolution of edge gradients. We propose a vectorized Tenengrad method: independently calculate the sum of absolute gradients in the horizontal ($G_x$) and vertical ($G_y$) directions, and construct a 2D sharpness vector $\mathbf{g}_t = (\sum |G_x|, \sum |G_y|)^{\top}$ to

capture texture direction features. A noise-aware circuit breaker mechanism is introduced to distinguish true details from high-frequency artifacts: let $n_t$ be the BRISQUE noise score, and $\mathbb{I}_{\text{trig}}$ be the triggered state when high noise ($n > \tau$) occurs for 5 consecutive frames. The final score integrates texture similarity, fusing penalty logic $\mathcal{M}$, combined with an upper-convex logarithmic transform (first use, for detail enhancement of low-middle score segments), and normalized exponential decay weight $\hat{w}(d)$ (Eq. 5) (the similarity $\mathcal{S}(\cdot, \cdot)$ follows the cosine inner product definition in Eq. 3):

The upper-convex logarithmic transform (for $x \in [0, 1]$, avoid meaningless logarithm, normalized) is:

$$\mathcal{L}(x) = \frac{\ln(1 + k \cdot x)}{\ln(1 + k)} \quad (k > 0) \tag{10}$$

where $k$ controls the upper-convex degree, and $k \in [10, 20]$ is commonly used in the visual field. The numerator ensures $y = 0$ when $x = 0$ (monotonically increasing, obvious upper-convex feature), and the denominator realizes normalization to ensure $y = 1$ when $x = 1$.

The fusing penalty logic $\mathcal{M}$ is:

$$\mathcal{M}(x_t) = \begin{cases} \mathcal{S}(\mathbf{g}_t, \mathbf{g}_1), & \mathbb{I}_{\text{trig}} = 0 \text{ and } n_t < \tau \\ \text{clip}(1 - \mathcal{S}(\mathbf{g}_t, \mathbf{g}_1), 0, 0.2), & \text{otherwise} \end{cases} \tag{11}$$

The final score formula is:

$$S_{\text{Sharpness}} = \sum_{d=1}^{T-1} \hat{w}(d) \cdot \mathcal{L}(\mathcal{M}(x_{d+1})) \tag{12}$$

where $\mathcal{S}(\mathbf{g}_t, \mathbf{g}_1)$ is the cosine similarity between the sharpness vector of the $t$-th frame and the initial frame. This mechanism can accurately distinguish true details from high-frequency noise and impose penalties on systematic visual collapse.

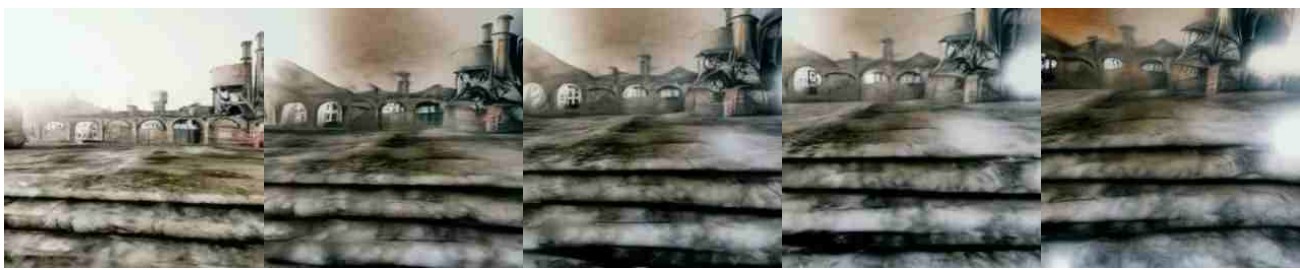

(a) Inconsistent Sharpness Retention (score 2.4%)

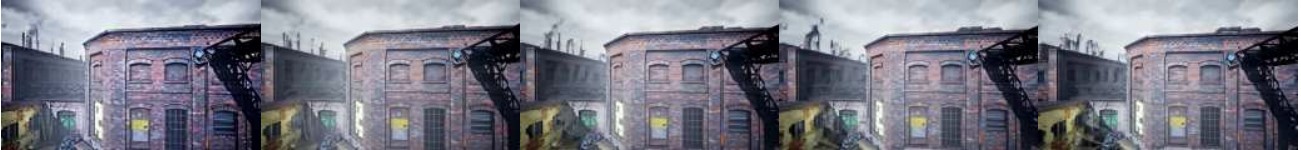

(b) Consistent Sharpness Retention (score 99.19%)

*Figure 10.* **Visualization of Subject Consistency of Sharpness Retention.** We defined sharpness retention consistency to demonstrate the clarity of the pictures. (a) is blurry through time. (b) is clearer as the bricks on the wall can be clearly seen.

**5. Motion Smoothness** ($S_{\textbf{Motion}}$). This metric evaluates the sequence coherence of generated videos using the motion prior of a video interpolation model. We adopt a sampling-reconstruction paradigm: discard the odd frames in the generated video and use the interpolation model to reconstruct them. Subsequently, the smoothness is quantified by calculating the comprehensive deviation between the reconstructed frames and the original discarded frames in terms of perceptual similarity (LPIPS), structural similarity (SSIM), and pixel error (MSE). This method can effectively capture instantaneous jitter or motion incoherence in videos, measuring whether the generation process conforms to basic temporal continuity.

**6. Trajectory Accuracy** ($S_{\textbf{Accuracy}}$). This metric quantifies the accuracy with which the world model follows preset camera control commands. The evaluation includes two stages: trajectory alignment and accuracy calculation. First, ViPE is used to extract the original extrinsic trajectory $\mathbf{E}_{\text{raw}}$; to eliminate coordinate system mismatch, a rotation transformation matrix $\mathbf{R}$ is introduced to align the trajectory to the reference coordinate system combined with the main motion direction of the video, resulting in a synchronized trajectory $\mathbf{E}_{\text{sync}}$. Accuracy is then evaluated by calculating the average absolute cosine similarity (adopting upper-convex logarithmic transform Eq. 10 for score calibration) between the aligned trajectory and the command sequence $\mathbf{E}_{\text{cmd}}$ in the motion tangent space (the similarity $\mathcal{S}(\cdot, \cdot)$ follows the cosine inner product definition in Eq. 3):

$$S_{\text{Accuracy}} = \frac{1}{T-1} \sum_{t=1}^{T-1} \mathcal{L}\left(|\mathcal{S}(\dot{\mathbf{e}}_{\text{sync},t}, \dot{\mathbf{e}}_{\text{cmd},t})|\right) \tag{13}$$

where $\dot{\mathbf{e}}$ represents the tangent derivative of the extrinsic trajectory, and $\mathcal{L}(\cdot)$ is the upper-convex logarithmic transform (Eq. 10) to enhance the detail of low-middle score segments. This metric focuses on the accurate mapping of motion trends and is the core scale for measuring the "instruction-level controllability" of the world model.

**7. Trajectory Tolerance** ($S_{\textbf{Tolerance}}$). This metric aims to evaluate the robustness of the model in trajectory execution under the guidance of accurate Ground-truth. Unlike $S_{\text{Trajectory}}$ which relies on third-party estimators, this metric directly uses the system-built precise extrinsic sequence $\mathbf{E}_{\text{gt}}$ as the benchmark. We adopt the same coordinate alignment and tangential analysis logic, quantifying the score by calculating the average absolute cosine similarity (adopting upper-convex logarithmic transform Eq. 10 for score calibration) of motion trends between the generated trajectory and the Ground-truth trajectory (the similarity $\mathcal{S}(\cdot, \cdot)$ follows the cosine inner product definition in Eq. 3):

$$S_{\text{Tolerance}} = \frac{1}{T-1} \sum_{t=1}^{T-1} \mathcal{L}\left(|\mathcal{S}(\dot{\mathbf{e}}_{\text{sync},t}, \dot{\mathbf{e}}_{\text{gt},t})|\right) \tag{14}$$

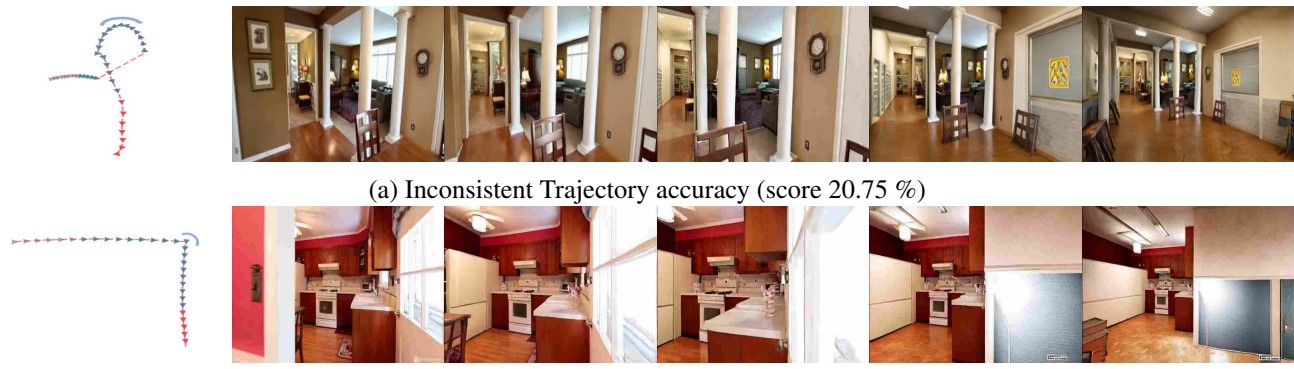

(a) Inconsistent Trajectory accuracy (score 20.75 %)

(b) Consistent Trajectory Accuracy (score 47.44 %)

Figure 11. **Visualization of Subject Consistency of Trajectory Accuracy.** Trajectory Accuracy is a indicator that shows the ability of the model to accurately follow given route. (a) could not follow accurately. (b) shows better better accuracy.

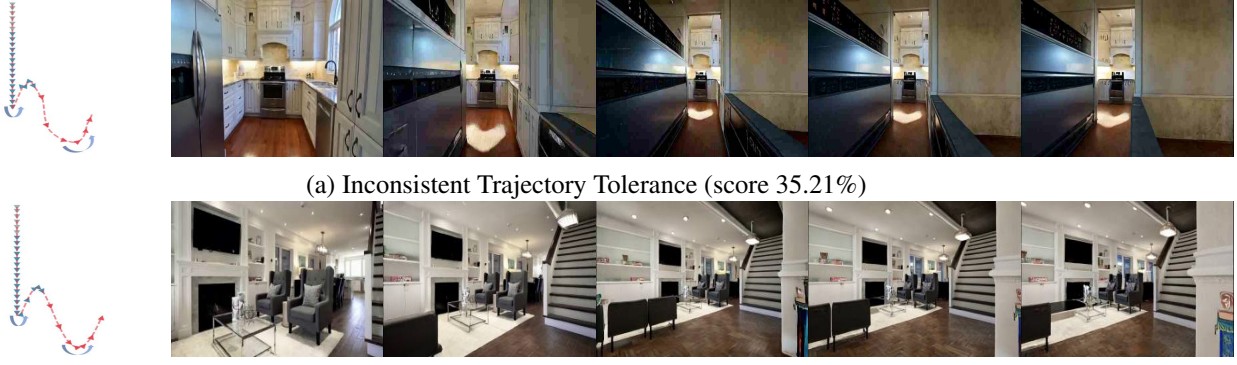

(a) Inconsistent Trajectory Tolerance (score 35.21%)

(b) Consistent Trajectory Tolerance (score 70.01%)

Figure 12. **Visualization of Subject Consistency of Trajectory Tolerance.** Trajectory tolerance measures model's ability to tolerant the route not accurately designed. (a) can not follow the route smoothly while (b) has a trajectory tolerance with a better performance.

where $\dot{\mathbf{e}}$ represents the tangent derivative of the extrinsic trajectory, and $\mathcal{L}(\cdot)$ is the upper-convex logarithmic transform (Eq. 10) to enhance the detail of low-middle score segments. By excluding the uncertainty interference of the estimator, this metric can more purely reflect the physical fidelity of the world model under ideal control conditions, serving as a performance upper bound reference for controllability evaluation.

**8. Memory Symmetry ($S_{\text{Memory}}$).** This metric quantifies the model's logical loop-closure ability by checking the pixel-wise consistency of symmetric frame pairs $(f_t, f_{T-t+1})$ in cyclic or symmetric actions. We calculate the Mean Squared Error ($\text{MSE}_t$) of symmetric frame pairs, which is then mapped to a similarity score using a compound exponential function with an offset $a$ and decay coefficients $k_{\text{val}}$, $k_{\text{exp}}$. The final score is obtained using a weighted sum that emphasizes distant frames, where the weight $\hat{w}_t$ (first use) adopts a normalized inverse exponential decay algorithm (frame distance $d = |T/2 - t|$):

$$w_t = e^{-\gamma \cdot d}, \quad \hat{w}_t = \frac{w_t}{\sum_{k=1}^{\lfloor T/2 \rfloor} w_k} \quad (\gamma > 0) \tag{15}$$

where $\gamma > 0$ is the decay coefficient, and $d = |T/2 - t|$ is the distance between the $t$-th frame and the middle frame of the sequence (the closer to the start and end of the sequence, the smaller the distance, the larger the weight). $\hat{w}_t$ is the normalized weight (sum to 1). The final score formula is:

$$S_{\text{Memory}} = \sum_{t=1}^{\lfloor T/2 \rfloor} \hat{w}_t \cdot e^{-k_{\text{val}} \cdot [\max(0, \text{MSE}_t - a)]^{k_{\text{exp}}}} \tag{16}$$

The weight $\hat{w}_t$ increases as $t$ decreases (i.e., closer to the start and end of the sequence). This distant-enhanced mechanism is specifically designed to capture logical failures caused by memory decay in long-sequence generation.

**9. Trajectory Alignment ($S_{\text{Alignment}}$).** This metric evaluates the model's ability to maintain symmetric closed-loop camera trajectories in round-trip tasks. ViPE is used to extract the camera extrinsic parameters of each frame, resulting in a sequence $\{\mathbf{E}_t\}_{t=1}^{T}$, where $\mathbf{E}_t \in \mathbb{R}^{12}$ represents the reshaped extrinsic matrix. We calculate the deviation of motion features between the first half of the forward trajectory and the second half of the reverse trajectory at each symmetric point. Define the instantaneous displacement vector $\vec{\mathbf{v}}_t = \text{pos}(\mathbf{E}_{t+1}) - \text{pos}(\mathbf{E}_t)$, where $\text{pos}(\cdot)$ extracts the translation component from the extrinsic parameters. The trajectory score is defined by the mean of the mirror similarity (adopting upper-convex logarithmic transform Eq. 10 for score calibration) between the direction vectors of symmetric frame pairs (the similarity $\mathcal{S}(\cdot, \cdot)$ follows the cosine inner product definition in Eq. 3):

$$S_{\text{Alignment}} = \frac{1}{\lfloor T/2 \rfloor} \sum_{t=1}^{\lfloor T/2 \rfloor} \mathcal{L}\left(\mathcal{S}(\vec{\mathbf{v}}_t, -\vec{\mathbf{v}}_{T-t+1})\right) \tag{17}$$

where $\mathcal{L}(\cdot)$ is the upper-convex logarithmic transform (Eq. 10) to enhance the detail of low-middle score segments. This metric measures the model's ability to maintain spatial topology symmetry when handling the "go-and-return" logic.

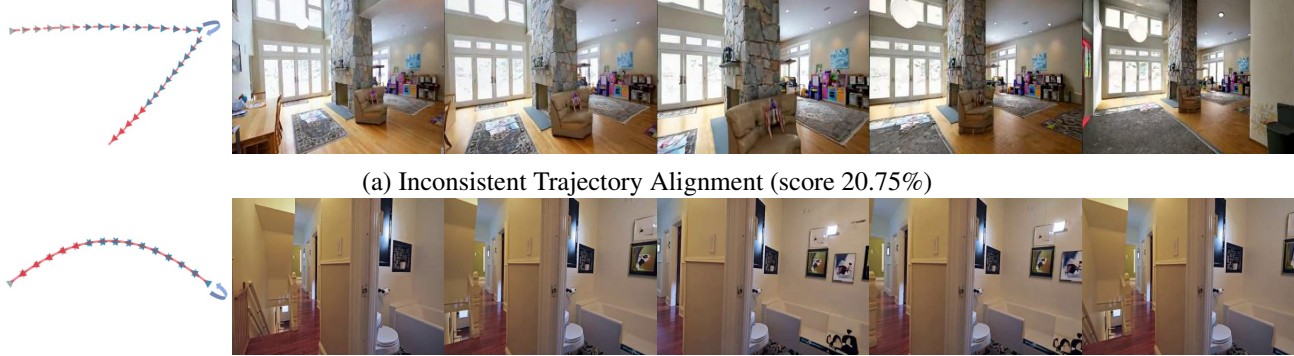

(a) Inconsistent Trajectory Alignment (score 20.75%)

(b) Consistent Trajectory Alignment (score 75.34%)

*Figure 13.* **Visualization of Subject Consistency of Trajectory Alignment.** Trajectory alignment measures the capabilities of the model following the memory route. (a) has a fold line while (b) has a great consistency throughout memory task.

# E. Expriment details

## E.1. Human Preference Validation Details

A fundamental requirement for any evaluation benchmark is that its automatic metrics faithfully reflect human subjective perception. To rigorously validate this property, we pose three statistical questions: **(Q1)** Do our metrics maintain strong rank-order agreement with human preference across the full model performance spectrum, including models with similar capability levels? **(Q2)** Are the human-perceived performance differences among evaluated models statistically significant and of practical magnitude? **(Q3)** Can our metrics reliably discriminate between models whose capabilities are genuinely close, rather than only separating models at opposite ends of the performance range? To answer these questions, we conduct a comprehensive human preference study with an expanded model pool and a systematically balanced task set, followed by a full statistical analysis encompassing Spearman rank correlation, Kruskal-Wallis global difference test, Dunn's post-hoc pairwise comparison, effect size estimation, and 95% confidence interval analysis. All scripts and raw data are released in the supplementary material.

**Experiment Design.** To obtain a comprehensive and unbiased assessment, we include 14 world models spanning the full performance spectrum of our benchmark across multiple camera control paradigms. Crucially, to probe metric sensitivity at the fine-grained end of the scale (Q3), six close-performing models—**ASTRA**, **CamI2V**, **CameraCtrl**, **Matrix-game 2.0**, **MotionCtrl**, and **WAN 2.2**—are explicitly included alongside stronger models, ensuring the evaluation is not artificially restricted to easily-separable pairs.

To avoid any systematic confound from uneven task difficulty that could bias human preference scores (Q2), we adopt a *stratified uniform sampling* strategy, drawing 4 representative tasks from each of the 4 predefined difficulty levels (D1–D4), yielding 16 standardised evaluation tasks shared identically across all models. Human annotators scored each generated video on a 5-point Likert scale (1 =extremely poor, 5 =excellent) along two core dimensions: *overall visual quality* and *camera control precision*—the two key evaluation dimensions of our benchmark. Each model received $N = 192$ individual ratings (12 annotators × 16 tasks), totalling $N_{\text{total}} = 2{,}688$ data points across all models.

**Statistical Analysis Framework.** We answer Q1–Q3 through a four-stage analysis pipeline:

- **Stage 1 — Spearman Rank Correlation (Q1):** Measures rank-order agreement between objective metric scores and mean human preference scores across all 14 models, directly testing whether our metrics preserve the human-perceived performance ordering (significance threshold $\alpha = 0.05$).

- **Stage 2 — Kruskal-Wallis H Test (Q2):** A non-parametric omnibus test examining whether the human preference score distributions differ significantly across all 14 models, establishing whether the performance gaps our benchmark captures are genuinely perceivable by humans.

- **Stage 3 — Effect Size ($\eta^2$) and 95% Confidence Intervals (Q2):** Quantify the practical magnitude of the observed differences and verify that model-level rankings are stable and non-overlapping in human perception.

- **Stage 4 — Dunn's Post-hoc Pairwise Comparison (Q3):** Bonferroni-corrected pairwise tests applied to the six close-performing models, directly testing whether our metrics retain discriminative power even among models with similar capability levels.

*Table 11.* Human preference scores, objective metric scores, 95% confidence intervals, and rank comparison across all 14 world models (sorted by human preference rank). **H-Rank**: rank by human preference score; **O-Rank**: rank by objective metric score; $|\Delta\text{Rank}|$: absolute rank difference between the two orderings.

| Model | Obj. Score | Human Score (Mean±SD) | 95% CI | | H-Rank | O-Rank | $|\Delta\text{Rank}|$ |
|---|---|---|---|---|---|---|---|
| | | | Lower | Upper | | | |
| HY-World 1.5 | 0.7873 | $3.865 \pm 0.882$ | 3.740 | 3.989 | 1 | 1 | 0 |
| HunyuanVideo-1.5 | 0.7188 | $3.641 \pm 0.832$ | 3.523 | 3.758 | 2 | 3 | 1 |
| videox-fun-Wan | 0.7474 | $3.484 \pm 0.971$ | 3.347 | 3.622 | 3 | 2 | 1 |
| NVIDIA Cosmos | 0.6275 | $2.813 \pm 0.756$ | 2.706 | 2.920 | 4 | 7 | 3 |
| YUME 1.5 | 0.6209 | $2.776 \pm 0.958$ | 2.641 | 2.912 | 5 | 8 | 3 |
| RealCam-I2V | 0.6865 | $2.490 \pm 1.023$ | 2.345 | 2.634 | 6 | 6 | 0 |
| CogVideoX-I2V | 0.6963 | $2.255 \pm 0.827$ | 2.138 | 2.372 | 7 | 5 | 2 |
| CamI2V | 0.5765 | $2.109 \pm 0.740$ | 2.005 | 2.214 | 8 | 10 | 2 |
| AC3D | 0.7149 | $1.953 \pm 0.967$ | 1.816 | 2.090 | 9 | 4 | 5 |
| Matrix-game 2.0 | 0.5663 | $1.776 \pm 0.797$ | 1.663 | 1.889 | 10 | 13 | 3 |
| CameraCtrl | 0.5762 | $1.604 \pm 0.639$ | 1.514 | 1.695 | 11 | 11 | 0 |
| MotionCtrl | 0.5486 | $1.302 \pm 0.514$ | 1.229 | 1.375 | 12 | 14 | 2 |
| WAN 2.2 | 0.5731 | $1.193 \pm 0.433$ | 1.131 | 1.254 | 13 | 12 | 1 |
| ASTRA | 0.5980 | $1.193 \pm 0.421$ | 1.133 | 1.252 | 13 | 9 | 4 |

**Mean $|\Delta\text{Rank}|$:** 1.93 averaged across all 14 models. Sample size per model: $N = 192$ (12 annotators × 16 tasks).

*Table 12.* Dunn's post-hoc pairwise comparison (Bonferroni-corrected $p$-values) among the six close-performing models. *n.s.*: $p \geq 0.05$ (not significant after correction); all other entries: $p < 0.001$.

| | ASTRA | CamI2V | CameraCtrl | Matrix-game 2.0 | MotionCtrl | WAN 2.2 |
|---|---|---|---|---|---|---|
| ASTRA | — | <0.001 | <0.001 | <0.001 | *n.s.* | *n.s.* |
| CamI2V | <0.001 | — | <0.001 | <0.001 | <0.001 | <0.001 |
| CameraCtrl | <0.001 | <0.001 | — | *n.s.* | <0.001 | <0.001 |
| Matrix-game 2.0 | <0.001 | <0.001 | *n.s.* | — | <0.001 | <0.001 |
| MotionCtrl | *n.s.* | <0.001 | <0.001 | <0.001 | — | *n.s.* |
| WAN 2.2 | *n.s.* | <0.001 | <0.001 | <0.001 | *n.s.* | — |

*Table 13.* Per-sub-metric objective benchmark scores for all 14 world models (sorted by human preference rank). **IQ**: Image Quality; **BC**: Brightness Consistency; **CTC**: Color Temperature Constraint; **SR**: Sharpness Retention; **MS**: Motion Smoothness[†]; **TA**: Trajectory Accuracy; **MSym**: Memory Symmetry; **TAl**: Trajectory Alignment. [†]Borrowed from VBench (**?**).

| Model | IQ | BC | CTC | SR | MS[†] | TA | MSym | TAl | Avg |
|---|---|---|---|---|---|---|---|---|---|
| HY-World 1.5 | 0.6675 | 0.8051 | 0.7819 | 0.6634 | 0.9921 | 0.7472 | 0.8481 | 0.6776 | 0.7729 |
| HunyuanVideo-1.5 | 0.7128 | 0.7027 | 0.7477 | 0.5545 | 0.9908 | 0.6844 | 0.6336 | 0.6449 | 0.7089 |
| videox-fun-Wan | 0.6410 | 0.5972 | 0.5473 | 0.5998 | 0.9858 | 0.7172 | 0.9009 | 0.6876 | 0.7096 |
| NVIDIA Cosmos | 0.6778 | 0.6952 | 0.7170 | 0.4363 | 0.9907 | 0.4955 | 0.3738 | 0.6419 | 0.6285 |
| YUME 1.5 | 0.6232 | 0.3810 | 0.4165 | 0.4023 | 0.9765 | 0.7113 | 0.5276 | 0.5988 | 0.5796 |
| RealCam-I2V | 0.6227 | 0.4130 | 0.5547 | 0.6269 | 0.9860 | 0.5630 | 0.7948 | 0.6668 | 0.6535 |
| CogVideoX-I2V | 0.6521 | 0.8988 | 0.8129 | 0.7951 | 0.9938 | 0.5950 | 0.6010 | 0.4084 | 0.7196 |
| CamI2V | 0.5284 | 0.4343 | 0.3568 | 0.4297 | 0.9861 | 0.6314 | 0.3631 | 0.6038 | 0.5417 |
| AC3D | 0.4573 | 0.7307 | 0.6524 | 0.5332 | 0.9919 | 0.5785 | 0.9068 | 0.6250 | 0.6845 |
| Matrix-game 2.0 | 0.4851 | 0.2963 | 0.2937 | 0.4149 | 0.9848 | 0.7008 | 0.3311 | 0.6362 | 0.5179 |
| CameraCtrl | 0.4473 | 0.3717 | 0.2511 | 0.4545 | 0.9796 | 0.6778 | 0.4279 | 0.6097 | 0.5274 |
| MotionCtrl | 0.4562 | 0.3980 | 0.2012 | 0.4294 | 0.9735 | 0.6730 | 0.3098 | 0.5932 | 0.5043 |
| WAN 2.2 | 0.5545 | 0.3886 | 0.3411 | 0.3428 | 0.9557 | 0.6514 | 0.4480 | 0.5703 | 0.5315 |
| ASTRA | 0.5335 | 0.5091 | 0.4338 | 0.5488 | 0.9799 | 0.6115 | 0.4323 | 0.5518 | 0.5751 |

Mean: 0.576 0.544 0.508 0.517 0.983 0.646 0.564 0.608    Std: 0.092 0.191 0.208 0.124 0.010 0.070 0.218 0.070

**Experimental Results and Analysis.**

*(i) Spearman Rank Correlation (Q1).* The Spearman rank correlation between our objective metric scores and the mean human preference scores across all 14 models is $r_s = 0.8053$ ($p = 0.0005 < \alpha = 0.05$), indicating strong and statistically significant rank-order agreement between our automated evaluation suite and human subjective judgment. Even with a model pool that deliberately includes similarly-performing entries, the mean absolute rank difference between metric-based and human-based orderings is only 1.93 positions (Table 11). **This answers Q1 affirmatively**: our metrics faithfully capture the relative quality ordering perceived by human annotators across the full performance spectrum.

*(ii) Kruskal-Wallis H Test and Effect Size (Q2).* The Kruskal-Wallis non-parametric omnibus test yields $H = 1496.8994$ ($p < 0.001$, $N_{\text{total}} = 2{,}688$), strongly rejecting the null hypothesis that human preference score distributions are identical across the 14 models. The Eta Squared effect size $\eta^2 = 0.5549$ ($k = 14$, $N = 2{,}688$) far exceeds the conventional large-effect threshold of $\eta^2 \geq 0.14$, meaning model identity accounts for approximately 55.5% of total variance in human preference scores. **This answers Q2**: the performance differences captured by our benchmark correspond to statistically real, human-perceivable distinctions of substantial practical magnitude—not merely numerical artefacts.

*(iii) 95% Confidence Intervals (Q2, supplementary).* As shown in Table 11, the 95% confidence intervals are non-overlapping across broad performance tiers—the top-tier group (HY-World 1.5, HunyuanVideo-1.5, videox-fun-Wan, mean scores $\geq 3.48$) is clearly separated from the mid-tier group (NVIDIA Cosmos, YUME 1.5, RealCam-I2V, mean scores 2.49–2.81) and the lower-tier group (mean scores $\leq 2.26$). This provides direct visual evidence that the benchmark rankings are stable in human perception. The limited CI overlap observed between a small number of immediately adjacent models within the same tier is statistically expected for models of similar capability, and is consistent with the Dunn test results below.

*(iv) Dunn's Post-hoc Test for Close-Performing Models (Q3).* Among the six close-performing models included precisely to probe fine-grained discrimination, the Bonferroni-corrected Dunn's test (Table 12) finds that **11 out of 15 pairwise comparisons are statistically significant at** $p < 0.001$. The four non-significant pairs—ASTRA vs. MotionCtrl, ASTRA vs. WAN 2.2, CameraCtrl vs. Matrix-game 2.0, and MotionCtrl vs. WAN 2.2—correspond precisely to model pairs whose mean human preference scores are virtually identical (pairwise mean differences $\leq 0.11$ on the 5-point scale), indicating genuine performance equivalence rather than metric insensitivity. **This answers Q3**: our metrics reliably discriminate between close-performing models wherever human annotators themselves perceive a difference; non-detections reflect true ties in human perception.

*(vi) Sub-metric Analysis and Rank Discrepancy Interpretation.* Table 13 provides the per-sub-metric breakdown of the 14 models' objective benchmark scores, offering fine-grained insight into the moderate rank discrepancies observed in Table 11 (mean $|\Delta\text{Rank}| = 1.93$). The largest discrepancy occurs for **AC3D** (H-Rank 9, O-Rank 4, $|\Delta\text{Rank}| = 5$): its high objective average is driven by exceptionally strong Memory Symmetry (0.9068) and Motion Smoothness (0.9919), yet its Image Quality score (0.4573) is the second-lowest among all models. This pattern suggests that human annotators place proportionally higher weight on perceptual image quality when forming overall preference judgments, compared to the uniform-weight aggregation used in the automatic metric. Similarly, **NVIDIA Cosmos** (H-Rank 4, O-Rank 7, $|\Delta\text{Rank}| = 3$) exhibits high visual quality (IQ = 0.6778, CTC = 0.7170) but comparatively lower Trajectory Accuracy (0.4955) and Memory Symmetry (0.3738), indicating that humans reward visually appealing output even when camera trajectory precision is moderate. These observations highlight complementary strengths of human evaluation and automated metrics, and motivate future work on perceptually-weighted metric aggregation.

**Summary.** Taken together, the three statistical questions posed at the outset are answered affirmatively. **Q1** (*rank-order agreement*): our metrics achieve strong Spearman

correlation with human preference ($r_s = 0.8053$, $p < 0.001$) with a mean rank difference of only 1.93 positions across 14 diverse models. **Q2** (*significance and magnitude*): human-perceived performance differences are highly significant ($H = 1496.90$, $p < 0.001$) and of large practical effect ($\eta^2 = 0.5549$), with non-overlapping confidence intervals confirming stable tier-level discrimination. **Q3** (*fine-grained discriminability*): our metrics successfully distinguish 11 out of 15 close-model pairs ($p < 0.001$), with the four non-significant cases attributable to genuine performance equivalence in human perception. Additionally, sub-metric analysis reveals that residual rank discrepancies stem from differential human weighting of perceptual image quality versus structural trajectory metrics, motivating perceptually-weighted aggregation as a direction for future work. "'

—

## E.2. Detailed comparison of world generation models

This subsection details the inference settings of all models, including their parameter configurations, model variants, and inference time.

*Table 14.* Detailed comparison of world generation models evaluated in our benchmark. We report the average generation time per instance measured on NVIDIA A800 GPUs. Version denotes the official release date of each model. [§] indicates whether a model supports explicit camera pose control as input.

| Method | Version | Ability | Resolution | Length (s) | FPS | Open Source | Speed[†] |
|---|---|---|---|---|---|---|---|
| NVIDIA Cosmos-predict2.5 | 25.09.25 | I2V | 1280×704 | 5.8 | 16 | ✓ | 11.4 min |
| HunyuanVideo-1.5 | 25.12.08 | I2V | 1248×720 | 5 | 24 | ✓ | 6 min |
| WAN 2.2 | 25.08.07 | I2V | 1280×704 | 5 | 24 | ✓ | 9 min |
| CogVideoX-5B-I2V | 25.06.30 | I2V | 720×480 | 4.9 | 10 | ✓ | 7.5 min |
| YUME 1.5 | 25.12.26 | I2V | 1280×704 | 5 | 16 | ✓ | 4.7 min |
| Matrix-game 2.0 | 25.08.12 | I2V | 640×352 | 3.4 | 24 | ✓ | 10 s |
| HY-World 1.5 | 25.12.17 | I2V | 832×480 | 3.2 | 24 | ✓ | 1 min |
| CameraCtrl | 24.04.02 | I2V | 256×256 | 7.6 | 10 | ✓ | 26 s |
| MotionCtrl | 23.12.06 | I2V | 256×256 | 7.6 | 10 | ✓ | 25 s |
| CamI2V | 25.07.12 | I2V | 512×320 | 7.6 | 10 | ✓ | 1 min |
| RealCam-I2V | 25.07.12 | I2V | 896×512 | 5 | 16 | ✓ | 2.5 min |
| videox-fun-Wan | 25.10.16 | I2V | 832×480 | 5 | 16 | ✓ | 2.9 min |
| AC3D | 25.04.01 | I2V | 720×480 | 6 | 8 | ✓ | 7.5 min |
| ASTRA | 25.12.09 | I2V | 832×480 | 4 | 20 | ✓ | 3.5 min |

