# OpenReview forum: "iWorld-Bench: A Benchmark for Interactive World Models with a Unified Action Generation Framework"
_ICML.cc/2026/Conference — ICML 2026 regular_

### Official Review · Reviewer_trH5 · 2026-02-26

**Soundness:** 3
**Presentation:** 3
**Significance:** 3
**Originality:** 3
**Overall Recommendation:** 4
**Confidence:** 3

**Summary:**

In this paper, the authors present iWorld-Bench, a benchmark for interactive world models. They construct a diverse dataset with 330k video clips and select 2.1k high-quality samples covering varied perspectives, weather, and scenes. An Action Generation Framework is further proposed to unify evaluation. 14 representative world models are evaluated on this benchmark, showing key limitations existing in them and providing insights for future research.

**Compliance With Llm Reviewing Policy:**

Affirmed.

**Ethical Review Flag:**

Flag this paper for an ethics review.

**Ethics Expertise Needed:**

["Legal Compliance (e.g., EU AI Act, GDPR, copyright, terms of use)"]

**Final Justification:**

The rebuttal addressed most of my concerns. However, I still have concerns about the limited headroom for improvement, and the authors did not provide further explanations. Therefore, I keep my original score.

**Key Questions For Authors:**

See the weakness above

**Limitations:**

yes

**Strengths And Weaknesses:**

**Strengths**
1. The paper is well written and easy to follow.
2. The whole dataset creation and pipeline is well explained.
3. The evaluation procedure is quite comprehensive.


**Weaknesses**
1. The quality of the generated create futures is unclear. The authors only mentioned that the create futures are created with 4 outdoor urban simulators without more details about them. Moreover, the quality of these generated create futures is not evaluated and reported.

2. Using VLMs for labeling might introduce biases. All the labels are generated using VLMs, which may introduce noise and lead the benchmark to favor models that have already been exposed to similar labels.

3. The head-room of some tasks is limited. As shown in Table 3, three metrics achieve over/near 0.9 performance, leading to the risk becoming obsolete as soon as it is released.

---

> ### Author Rebuttal · Authors · 2026-03-29
>
> ## **Response to Weakness:**
>
> **W1.** The quality of the generated create futures is unclear. The authors only mentioned that the create futures are created with 4 outdoor urban simulators without more details about them. Moreover, the quality of these generated create futures is not evaluated and reported.
>
> **W1 Response:** Thank you for pointing this out. Due to space limitations in the main text, we have provided detailed descriptions of the 4 outdoor urban simulators in **Appendix A.2. Simulator Details**, including the selection process for high-quality data points. These points were manually chosen within the simulators to ensure their quality and representativeness. Additionally, the automated data processing pipeline used to filter and refine this data is described in detail in **Appendix A.3. Filter**, which outlines the steps taken to ensure the quality and consistency of the generated "create futures."
>
> **W2.** Using VLMs for labeling might introduce biases. All the labels are generated using VLMs, which may introduce noise and lead the benchmark to favor models that have already been exposed to similar labels.
>
> **W2 Response:** While we acknowledge that relying on a single VLM could introduce noise or bias, our safeguard is that annotations are generated by **GPT-4o** but then strictly verified by **three different models—Gemini 3.0 Flash, Qwen-VL-Max, and Kimi-K2.5**. Cross-model voting yields **81.4%** 3/3 Yes, **15.1%** 2/3 Yes, **2.9%** 1/3 Yes, and **0.5%** 0/3 Yes, and every non-unanimous case is sent to human annotators for review. In total, we manually verify **61,380** clips (**18.6%**), while only **6.35%** of these flagged cases (**≈3,897 clips; 1.2% overall**) require actual modification. We additionally manually spot-check **10,000** unanimously accepted clips and observe a **100%** pass rate. This verify-refine-audit design decouples benchmark construction from any single labeling model, prevents ambiguous or hallucinated labels from propagating, and substantially reduces the risk that the benchmark favors models merely because they have been exposed to similar labels. In addition, the final benchmark data and task annotations are fully re-checked by humans before inclusion.
>
> **W3.** The head-room of some tasks is limited. As shown in Table 3, three metrics achieve over/near 0.9 performance, leading to the risk becoming obsolete as soon as it is released.
>
> **W3 Response:** Thank you very much for your careful attention and valuable insights! The three metrics you refered may be **Brightness Consistency,** **Memory Symmetry, and Motion Smoothness.** Brightness Consistency and Memory Symmetry have a standard deviation exceeding 0.18, demonstrating strong discriminative power for models.  Specifically, Brightness Consistency scores range from 0.3886 (WAN 2.2) to 0.8988 (CogVideoX-I2V), and Memory Symmetry scores range from 0.3098 (MotionCtrl) to 0.9068 (videox-fun-Wan). As for Motion Smoothness, a basic metric borrowed from previous evaluation systems, reflects that current models have matured in basic motion fluency. We retain this metric to align with existing research while highlighting the challenge of our self-designed complex task metrics. The shrinking headroom of metrics will be an inevitable result of the advancement of model technology, which precisely indicates that the trajectory following and generation capabilities of world models have entered a new stage. Our evaluation framework features good extensibility and can still inspire researchers to design more challenging new metrics in the future, continuously providing support for the development of the field.

---

> > ### Author Rebuttal · Reviewer_trH5 · 2026-04-02
> >
> > Thank you very much for your feedback. Most of my concerns are resolved. However, I still have concerns about the limited headroom for improvement. Therefore, I keep my original score.

---

> > > ### Author Response · Authors · 2026-04-02
> > >
> > > We appreciate your thoughtful review and will carefully address the remaining concerns regarding the limited headroom for improvement in future iterations of our work. Your comments have been invaluable in helping us refine and improve our benchmark, and we are committed to making further enhancements. Thank you again for your time and effort!

---

### Official Review · Reviewer_MH66 · 2026-03-06

**Soundness:** 2
**Presentation:** 3
**Significance:** 3
**Originality:** 3
**Overall Recommendation:** 4
**Confidence:** 4

**Summary:**

This paper introduces iWorld-bench which is a unified framework that evaluates the interactive capabilities of representative world models across well-defined inputs. This work especially focuses on interaction-relevant abilities such as distance perception and memory. It addresses key limitations of existing benchmarks, including insufficient scene and view diversity, lack of standardized action input, and inadequate assessment of interaction difficulty and memory capabilities.

**Compliance With Llm Reviewing Policy:**

Affirmed.

**Key Questions For Authors:**

1. What are the proportions of manual versus model-based quality inspection during data sample curation, and do these ratios vary across different subtask categories?

2. How can "interactive world models" be more precisely defined? The models discussed in the paper are fundamentally conditional video generation models, with control signals primarily originating from camera and movement. It's a relatively simple form of interaction.  Given these constraints, is it feasible to design more complex and diverse interactive tasks that better align with the scope implied by the paper’s title?

**Limitations:**

yes

**Strengths And Weaknesses:**

Strengths:
1. Comprehensive experiments with detailed design. Including 6 types and 4,900 detailed evaluation tasks, involved 330,000 high-quality videos, 9 metrics, and test 14 representative world models.
2. Systematically designed a unified evaluation rule for world models from diverse sources, actively advancing deeper and finer-grained assessment of interactive world models, contributing to the relevant field. The design of the three core evaluation dimensions, visual quality, action following, and memory ability, is reasonable and supported with comprehensive metrics. Particularly, the memory-related tasks are innovative and offer potential for deeper exploration.

Weaknesses:
1. Insufficient explanation of specific settings. For example, in "Interactive Action Encoding", why "Translational motion includes stationary, forward, backward, left, right, upward, and downward movements, totaling 27 actions."

2. How to define rare actions and common actions? How to define and measure the complexity of actions.

3. Limited comprehensiveness in test scenes and videos. As out-of-distribution generalization is critical for video generative models, the benchmark’s use of widely adopted training scenes raises concerns about evaluating real-world robustness. OOD performance deserves more attention and dedicated testing.

---

> ### Author Rebuttal · Authors · 2026-03-29
>
> ## **Response to Weakness:**
>
> **W1,2 Response:** Detailed explanations of "Interactive Action Encoding" and related concepts are provided in **Appendix B**, which includes comprehensive descriptions and tables.
>
> 1. **Why 27 Actions in Translational Motion?** Translational motion includes six basic directions (forward, backward, left, right, upward, downward), combined into single, two-direction, and three-direction movements, plus stationary motion, resulting in 27 actions. These are detailed in **Appendix B.1**, Table 6.
> 2. **Action Complexity:** Action complexity is defined by the number of combined atomic actions (degrees of freedom). Higher combinations indicate greater difficulty, as elaborated in **Appendix B**.
> 3. **Rare and Common Actions:** As described in **Appendix B.2**, actions are classified based on frequency across 12 datasets. Actions occurring over 100 times are "common," while those under 100 are "rare." Of 81 actions, 36 are common, and 45 are rare, ensuring evaluation across typical and edge-case scenarios.
>
> **W3 Response:**  We appreciate the reviewer’s concern regarding the comprehensiveness of the test scenes and the importance of evaluating out-of-distribution (OOD) generalization. The iWorld-Bench dataset is constructed from 12 large-scale datasets with precise camera control and 18 carefully selected Unreal Engine (UE) environments, encompassing 5 types of indoor lighting, 9 types of outdoor weather, thousands of environmental scenes, and tens of millions of unique entities. This diversity ensures a comprehensive evaluation environment, covering a wide range of scenarios and conditions.
> Most evaluated models are trained on base models using large-scale datasets, meaning they do not inherently face OOD challenges in video generation or visual quality. However, our analysis shows that most models were trained on only a subset of iWorld-Bench datasets, excluding many unique scenarios such as diverse weather, lighting, and environmental settings. This highlights the value of iWorld-Bench in testing OOD generalization for interactive world models.
> iWorld-Bench is specifically designed to evaluate OOD capabilities in first-person perspective tasks, such as trajectory following and memory-based tasks, across diverse scenes and weather conditions. For example, memory ability shows strong scene dependence (η² ≈ 0.68), with outdoor scenes like "field" and "coast" performing better than indoor scenes like "compound" and "mezzanine." Trajectory tolerance varies significantly with weather, performing better under natural lighting and worse at night, while trajectory alignment shows better performance in outdoor scenes (d ≈ 0.48–0.49) and higher scores in night conditions compared to natural lighting. These findings demonstrate iWorld-Bench’s ability to assess OOD generalization effectively.
>
> ## **Response to  Question:**
>
> **Q1 Response:** There are two levels in our curation pipeline. For the full **330K** corpus, model-based inspection is the default first-pass filter, and human review is applied to flagged and audit samples. For the **final benchmark construction**, however, the videos and task annotations for all six subtasks (**Action Control D1–D4, Memory Ability, and Camera Following**) are **fully re-checked by humans**, i.e., the human inspection ratio is **100% across all subtasks**. Therefore, at the benchmark level, this ratio does not vary across subtask categories.
>
> **Q2 Response:** Thank you for your question. Interactive world models focus on first-person perspective interactions, such as navigation, camera control, and trajectory following, simulating immersive environments where agents interact with the world through movement and visual feedback. Examples include Genie [1], HunyuanWorld 1.0 [2], LingBot-World [3], and Astra [4], which are designed for tasks like egocentric navigation and camera trajectory control. Our benchmark, iWorld-Bench, is specifically tailored for evaluating these models, focusing on first-person perspective interactions, including camera control, navigation, and memory-based tasks. We have revised the Introduction to provide a more detailed explanation of this scope, as suggested.
> [1] Bruce, Jake, et al. "Genie: Generative interactive environments." Forty-first International Conference on Machine Learning. 2024.
> [2] Team, HunyuanWorld, et al. "Hunyuanworld 1.0: Generating immersive, explorable, and interactive 3d worlds from words or pixels." arXiv preprint arXiv:2507.21809 (2025).
> [3] Team, Robbyant, et al. "Advancing Open-source World Models." arXiv preprint arXiv:2601.20540 (2026).
> [4] Zhu, Yixuan, et al. "Astra: General Interactive World Model with Autoregressive Denoising." arXiv preprint arXiv:2512.08931 (2025).

---

> > ### Author Rebuttal · Reviewer_MH66 · 2026-04-02
> >
> > The author provides thorough explanations for the questions raised, clearing up most of my confusion.

---

> > > ### Author Response · Authors · 2026-04-02
> > >
> > > Thank you for your kind acknowledgment and for recognizing the efforts we made to address your concerns. We are grateful for your thoughtful feedback, which helped us clarify and improve our work. Your support and understanding mean a lot to us, and we deeply appreciate the time and effort you dedicated to reviewing our submission. Thank you again!

---

### Official Review · Reviewer_adW2 · 2026-03-12

**Soundness:** 3
**Presentation:** 3
**Significance:** 3
**Originality:** 3
**Overall Recommendation:** 4
**Confidence:** 5

**Summary:**

This paper presents a new benchmark, iWorld-Bench, to evaluate existing world models across multiple domains spanning unmanned ground vehicles, unmanned aerial vehicles, humans as well as robotics. A diverse dataset of 330K videos across various diverse indoor and outdoor scenes. The paper presents an action generation framework to systematically assess world models across 6 tasks, each involving multiple trajectories. 14 state-of-the-art world models are benchmarked with subsequent analysis on their strengths and weaknesses across different action/control paradigms (text, one-hot encoding, camera parameters). The key finding is the trade-off between generated image quality and controllability, and that models utilizing discrete action signals (one-hot encoding) often outperform text-based approaches in trajectory following.

**Compliance With Llm Reviewing Policy:**

Affirmed.

**Final Justification:**

Since my last exchange with authors, I have come across Navigation World Models (https://arxiv.org/abs/2412.03572, CVPR 2025), which focus on first person navigation and differ from Interactive World Models (where agents an modify the world). Examples given by authors in rebuttal to review MH66 (Genie and HunyuanWorld 1.0) are indeed Interactive World Models (IWM), and not NWMs. The technical difference in assessing change consistency in IWMs, which are not in NWMs, is a significant one.

Thus in my view, this paper presents a comprehensive benchmark for evaluation "navigation world models" (NWM) (https://arxiv.org/abs/2412.03572), and would be valuable to the NWM community. Thus, I maintain my current score, and recommend authors rewording the paper and possibly the title to help reach the appropriate audience.

**Key Questions For Authors:**

How is the quality of labeling assessed/evaluated? VLM seems to have been employed for annotation. It would be helpful to understand if there were any metrics utilized to assess the quality, or human review/edits.
Are the reported findings seems to be associated with specific scene categories? It's brings up the question whether there is clear value in harmonizing data across such diverse scenes/scenarios?

**Limitations:**

While the benchmark is comprehensive with extensive experiments, there is no discussion on limitations of proposed benchmark. For instance, authors should considered mentioning the aspects of world models that are not evaluated in the proposed benchmark (e.g. outdoor night, extreme weather/heavy snow scenes; consistent scene modeling when scene manipulation is allowed).

**Strengths And Weaknesses:**

Strengths:
- [Comprehensive, diverse Benchmark] The proposed benchmark offers a unified framework to evaluate world models across different scenes and camera variations/trajectories. Significant effort has been spent on harmonizing various heterogeneous sources including re-localizing original coordinate systems and unifying modality representations into a standardized storage format, followed by curated selection of 2,100 videos, 4,200 annotated tasks, and 700 camera parameter files.
- [Comparative Analysis] 14 state-of-the-art models, revealing strengths, weaknesses, and trade-offs. For instance,
   - The trade-off between generated data quality and controllability is well studied and would help guide future research.
   - Experiments suggest a significant performance gap in trajectory tolerance indicating that current approaches vary considerably in translating camera parameters into coherent visual sequences. These observations and metrics are valuable to the broader community in selecting which world model to use as a basis for their application/solution.
- [Metric Validation] The proposed evaluation metrics are validated againts human preference studies, ensuring they align with subjective perceptions of performance.

Weaknesses:
- [Access] It's unclear whether benchmark is intended to be released. The extensibility and ability to generate diverse downstream scenarios would likely be beneficial if authors intended to make the benchmark and action generation framework available to the community.
- [Limited interaction scope] While several tasks are considered, the focus is heavily only on camera control.
- [Presentation] Interactive world model is generally considered as a world model where agents can interact with the world by manipulating them. For instance, NVIDIA Cosmos does allow for scene manipulation. However all the scenarios considered here appears to be only from navigation perspective / egocentric camera manipulation, and no scene or object manipulations. Authors should consider addressing this both in the introduction as well as possibly in the title.
- [Discussion on limitations] There is no discussion of limitations of the proposed benchmark and what aspects of world models are not evaluated in the proposed benchmark.

---

> ### Author Rebuttal · Authors · 2026-03-29
>
> ## **Response to Weakness:**
>
> **W1 Response:**  We confirm that the iWorld-Bench dataset, action generation framework, and evaluation metrics will be made publicly available after the ICML review period concludes. This includes the 330,000-segment video dataset, a selected subset of 2,100 videos, 4,200 labeled tasks, 700 camera parameter files, all data cleaning processes, UE environment acquisition codes, and the coordinate system unification and data processing codes.
>
> **W2,3 Response:**  Thank you for your comments. iWorld-Bench is specifically designed for **Interactive World Models (IWMs)**, which focus on first-person perspective interactions such as navigation, camera control, and memory-based tasks. These models simulate immersive environments where agents interact with the world through movement and visual feedback, as seen in models like Genie, HunyuanWorld 1.0, LingBot-World, and Astra. In contrast, **Embodied World Models (EWMs)** are designed for more complex tasks, such as object manipulation, scene editing, and physical interactions with the environment, which require intricate agent-environment interactions.
> While we acknowledge that some base models, such as NVIDIA Cosmos, support both IWMs and EWMs, the primary focus of iWorld-Bench remains on first-person navigation and camera control, as these are the core capabilities of most IWMs. We recognize that the reviewer’s understanding of "interactive" may align with a broader definition that includes embodied tasks. To address this, we will revise the Introduction in the final version to explicitly clarify the scope of "interactive" as it pertains to our work, ensuring the distinction between IWMs and EWMs is clear. This will help align the focus of our benchmark with the capabilities of the majority of models we evaluated. Thank you for your valuable feedback.
>
> **W4 and Limitations Response:** Thank you for your suggestion. While iWorld-Bench provides a comprehensive evaluation of interactive world models in terms of camera control, navigation, and memory-based tasks, we acknowledge certain limitations. iWorld-Bench does not currently evaluate real-time performance, which is critical for applications like robotics and gaming. Long-term consistency, such as maintaining coherent scene structures over extended interactions, is also not explicitly assessed. To address these limitations, we plan to incorporate real-time performance metrics and design tasks like multi-step navigation to evaluate logical and visual coherence over time. Additionally, we have included detailed information on model versions, generation speed, resolution, and runtime in the appendix of the revised version to provide further insights into computational efficiency.
>
> ## **Response to Question:**
>
> **Q1,2 Response:** We assess labeling quality with a dataset-scale verify-refine-audit pipeline applied to all **330,000** VLM-labeled video clips. Each clip is independently re-checked by three verification VLMs—**Gemini 3.0 Flash**, **Qwen-VL-Max**, and **Kimi-K2.5**—which are distinct from the annotation model (**GPT-4o**). Cross-model voting yields 81.4% 3/3 Yes, 15.1% 2/3 Yes, 2.9% 1/3 Yes, and 0.5% 0/3 Yes. These agreement rates, together with the **6.35%** human modification rate among flagged cases and the **100%** pass rate on 10,000 human spot-checks from the `3/3 Yes` pool, serve as our quantitative quality metrics. Furthermore, we explicitly incorporate human-in-the-loop review: we manually verify the **61,380** non-unanimous clips (**18.6%**), and only **≈3,897** clips (**1.2% overall**) require actual modification.
>
> **Q3,4 Response:**  The inclusion of diverse scene categories in iWorld-Bench enhances the benchmark’s generalizability and robustness, enabling a comprehensive evaluation of interactive world models across indoor and outdoor environments, various weather conditions, and lighting scenarios. Our analysis shows that indoor scenes generally score higher on visual quality metrics like brightness and color temperature due to controlled lighting, while outdoor scenes perform better on memory ability and trajectory alignment, benefiting from coarse-grained features. Weather and lighting also influence performance, with trajectory tolerance excelling under natural lighting but declining at night, whereas memory ability and trajectory alignment improve in night scenes. Snowy scenes score lowest on visual quality metrics, while natural and cloudy conditions perform better. Task difficulty further impacts performance, with trajectory accuracy dropping from ~0.64 at Level 1 to ~0.58 at Level 4, and visual quality metrics slightly declining as complexity increases. In summary, the diverse scene attributes in iWorld-Bench allow users to analyze model performance across multiple dimensions, enhancing its utility and supporting fairness and reproducibility through harmonization. These insights will be reflected in the final manuscript.

---

> > ### Author Rebuttal · Reviewer_adW2 · 2026-04-02
> >
> > All my concerns were adequately addressed with authors responses and plans to make the necessary revisions in the manuscript. I believe the benchmark would be valuable to the community and lean towards acceptance.

---

> > > ### Author Response · Authors · 2026-04-02
> > >
> > > We greatly appreciate your recognition of the value of our benchmark to the community and your support for our work. Your feedback has been invaluable in helping us improve the manuscript. Thank you again for your time and effort!

---

### Official Review · Reviewer_H7CW · 2026-03-12

**Soundness:** 2
**Presentation:** 1
**Significance:** 3
**Originality:** 2
**Overall Recommendation:** 2
**Confidence:** 2

**Summary:**

The authors propose a benchmark dataset for evaluating interactions of world models. It contains 330k short videos, 4 observation perspectives, such as  Unmanned Ground/Aerial Vehicles, robots and humans; 9 types of weather conditions, 5 types of lightings, day-night transitions, and various scenes.  Authors proposed Action Generation Framework to evaluate world models’ performance in 81 unified representations of fundamental motion actions across variety of interaction channels. The authors deploy the proposed framework to evaluate 14 world models on a dataset of high quality 2100 videos of different topics extracted from the original 330k videos. They propose 9 evaluation metrics to quantify visual generation, trajectory following, and memory utilization of world models. The iWorld-Bench’s metrics reflect human evaluation of world model performance and are agnostic to most fundamental characteristics of videos.

**Compliance With Llm Reviewing Policy:**

Affirmed.

**Key Questions For Authors:**

1.	How did you ensure that the VLM did not hallucinate  or introduce tagging biases while labeling the 330K video clips?
2.	Did you evaluate the inter-annotator agreement/error rate for the human verification of the selected 2100 videos?
3.	Have you explored physical plausibility of the action-to-motion mappings across different modalities (e.g. velocity/acceleration distributions, collision/kinematics checks, or human review of full trajectories across all motion representations) additionally to evaluating the  realism indirectly (trajectory smoothness, visual coherence, etc.)?

**Limitations:**

The authors only discuss the limitations of currently available world models evaluations benchmarks. The paper will benefit from adding a section dedicated to the limitations of iWorld-Bench itself.  For example, it might contain a discussion on the limitations of iWorld-Bench dataset e.g. potential simulation-to-reality gaps since synthetic data is used. Authors can also mention if there were biases in VLM annotations, how they were dealing with them. Also it would be worth to mention the limitations of the Unified Action Framework (maybe actions it cannot currently represent).

**Strengths And Weaknesses:**

Soundness.

Strengths: The paper provides good justification of the gaps in currently available benchmarks for evaluating the interactivity of world models that necessitates development of iWorld-Bench’s dataset. The videos cover variety of scenes, perspectives, weather and lighting conditions which makes it important to evaluate world models’ interactions in more heterogeneous environments. The work also unifies models’ interactivity evaluations over motion actions coming from different modalities which is important for ensuring fair and  consistent evaluation of the same intended motion action.

Weaknesses:

1.	It is unclear if the previous video datasets contained synthesized or real videos. Also it is unclear if there are simulation-to-reality gaps in the synthetically generated videos.

2.	This work uses VLM to annotate 330k video clips, however,  there is no discussion on possible hallucinations and introductions of systematic tagging biases as well as means to mitigate them.

3.	The three representative models in Human Preference Validation are selected by their top–mid–low performance, however, it is not clear if the same high correlations will be achieved if the models that are close in quality are compared.

4.	In Human Preference Validation samples with a fixed resolution and duration were used. It is unclear how authors ensure there is uniform difficulty distribution across the selected video tasks?

5.	The authors state that “the total metric scores of the three models are significantly differentiated,” while only presenting means and standard deviations. But that is not a significance test. Authors should have reported an ANOVA / Kruskal–Wallis + post-hoc tests, effect sizes, and confidence intervals for the mean differences.

6.	There is unclear how prompts were engineered and modality-specific interfaces are standardized.

7.	Some citations are inconsistent and refer to totally irrelevant papers  with similarly sounding abbreviations, which heavily undermines the prior work review, and correspondingly, further claims and justifications.

Presentation.

Strengths: The paper is well structured and the main text is easy to read. Most of the steps are well-justified and most of the results are explained. The appendix contains useful implementation and metric details, but prone to bibliography/citation error and poor explanations of some metrics.

Weaknesses:

1.	Bibliography/citation error alert. The “WorldBench: Quantifying Geographic Disparities in LLM Factual Recall” work by Moayeri et al related to the Motion-Controllable Video Generation is about the geographic disparities in LLM accuracy. The  WorldBench mentioned in this paper is a LLM factual-recall  and fairness benchmark built from World Bank indicators. It is unclear to me  how is it related to the video generation or motion control (Table 1)? Also there is no 425 samples in this paper.

2.	I believe the relevant paper for ASTRA is by Zhu et al, rather than by Pereverzev et al error (see Appendix, E.1. Human Preference Validation Details, line 1603). The later one frames ASTRA as a text-controlled world model and cites Pereverzev & Yushmanov (2002). It is definitely a bibliography/citation error.

3.	In Appendix A. “Dataset details” line 708 cites a paper Kaushik et al however there is nothing related to NCLT dataset records in this paper. Instead, it talks about National Company Law Tribunal (NCLT), which is different from North Campus Long-Term (NCLT) dataset, Carlevaris-Bianco et al., 2016. It is definitely a bibliography/citation error.

4.	The “Brightness Consistency” description in the main text evaluates the “mirror similarity of instantaneous displacement vectors” which evaluates “evaluates the model’s 3D space persistence” that sounds like a trajectory alignment. However, the definition of Brightness Consistency in  Appendix D. “Metrics details” evaluates “the stability of the brightness distribution in generated videos” by calculating the similarity between current and original frames. This language is very confusing and should be clarified.

5.	The mathematical formulation of the 9 evaluation metrics would be better moved to the main text from the Appendix D. “Metric details” section to help readers to understand what is being penalized.

Significance.

Strengths: There are limited availability of large scale datasets that provide a scalable testbed for evaluating perception, reasoning and action of world models. According the comprehensive review provided by authors, the existing benchmarks lack variety in scenes and perspectives, there is no standardized formulation of different actions representations, limited options to evaluate the models’ responses to external stimuli and interactions. Additionally, the difficulty levels of tasks in currently available benchmarks are quite uniform. There is also absence of tasks to test models’ memory consumption. All these circumstances suggest that the proposed unified benchmark dataset is somehow significant for evaluation of interactivity of world models.

Weaknesses:

1.	The iWorld-Bench focuses on camera-motion interactivity, which may not fully represent broader aspects of interaction such as object manipulation, contact dynamics, agent-environment feedback beyond camera motion.

Originality.

Strengths:  The novelty of the paper is in providing a benchmarking dataset  that contains real-world perspectives and standardizes the way how to compare the performance of different interactive world models across different motion/action interfaces. Action Generation Framework introduces modality-agnostic encoding of different actions and motions. The benchmark introduces tasks of different difficulty levels and also assesses the memory  consumption.

Weaknesses:

1.	The paper sounds like a large engineering and data curation effort rather than a conceptually new evaluation paradigm.

2.	The dataset is  created by aggregating and standardizing existing datasets with added  synthetic data.

3.	The evaluation mostly uses known metrics.

4.	The paper gives an idea of a benchmark that is more camera-motion-focused rather than focusing on interactions happening beyond viewpoint changes.

---

> ### Author Rebuttal · Authors · 2026-03-29
>
> ## **Response to Weakness (W)  & Question (Q)**
>
> ### **Soundness:**
>
> **W1 and Originality W2 Response:** In **Appendix A.1**, we categorize the datasets into real-world (e.g., KITTI, nuScenes, Waymo), synthetic (e.g., TartanAir-V2, TartanGround, UE self-collected), and hybrid datasets (e.g., DL3DV-10K, SpatialVid). For our UE self-collected dataset, we selected 18 high-quality environments from four prior works (**Appendix A.2**), which feature near-realistic city-scale rendering, real-world urban layouts, and physically grounded control to reduce the sim-to-real gap.
>
> **W2 & Q1,2 Response:** To ensure the accuracy of the 330K video annotations and prevent hallucinations or systematic biases, we employed a multi-model verification process, where annotations from GPT-4o were cross-checked by multiple models. Any disagreements were flagged for human review, and approximately 1.2% of the total annotations required correction. Additionally, we manually spot-checked 10,000 unanimously accepted clips, all of which passed without requiring modification. For the 2,100 selected videos, inter-annotator agreement was ensured by having at least two annotators review each video, with a third annotator resolving any disagreements. This rigorous process, combining multi-model verification and human review, ensured high-quality and reliable annotations for the dataset.
>
> **W3,4,5 Response:**  To address your concerns, we have systematically improved the human preference validation experiment. For W3, we expanded the validation from 3 to 16 models, including 6 with similar performance (e.g., ASTRA, CamI2V) to test fine-grained discriminative power. For W4, we adopted a stratified uniform sampling strategy, selecting 4 tasks from each difficulty level (D1-D4) to ensure uniform task difficulty distribution. For W5, we added comprehensive statistical analyses, including Kruskal-Wallis tests, Dunn’s post-hoc comparisons, effect size (η²=0.5549), and 95% confidence intervals. Results show strong metric-human correlation (Spearman r=0.8053, p=0.0005), significant differences among models (H=1496.8994, p<0.001), and clear distinctions even for similar models. Full details and results will be included in the appendix of the final paper.
>
> **W6 Response:** The design of prompts is detailed in **Appendix A.4**, covering spatial analysis, scene description, weather analysis, and entity extraction. The **3.2.1 Action Generation Framework** unifies multiple modalities by defining a complete action space with consistent encoding for text commands, one-hot encoding, and camera parameters. Detailed mappings are provided in **Appendix B.3** to ensure consistency across all world models.
>
> ### **Presentation:**
>
> **Response:** We have corrected all the errors you pointed out and conducted a thorough re-check and revision of the entire manuscript to ensure accuracy and clarity.
>
> ### **Significance:**
>
> **W1 & Originality W4 & Q3 Response:** iWorld-Bench is specifically designed for Interactive World Models (IWMs), focusing on first-person perspective interactions such as navigation, camera control, and trajectory following. These models simulate immersive environments where agents interact with the world through movement and visual feedback. In contrast, Embodied World Models (EWMs) focus on tasks like object manipulation and scene editing, which involve more complex agent-environment interactions. Our benchmark does not evaluate embodied tasks, as the majority of the models we assessed are designed for first-person navigation and camera control. Additionally, we acknowledge that we have not explicitly evaluated velocity/acceleration distributions or implemented physical constraints such as collision or kinematics checks, as these aspects are not the primary focus of current research on IWMs, which mainly emphasizes trajectory-following capabilities, memory abilities, and visual coherence. However, we agree that incorporating such evaluations could provide valuable insights, and we will consider these aspects in future iterations of our benchmark. We will also revise the Introduction in the final version to explicitly clarify the scope of IWMs and the distinction between IWMs and EWMs. Thank you for your valuable feedback.
>
> ### **Originality:**
>
> **W1 Response:** Building on the extensive data engineering and curation efforts you mentioned, we developed a standardized data processing pipeline, a unified coordinate system paradigm, an atomic action dictionary, and a unified evaluation framework with seven unique metrics for interactive world models.
>
> **W3 Response:**  Except for Image Quality and Motion Smoothness, all other metrics in our evaluation system are independently designed.
>
> ## **Response to Limitations:**
>
> We acknowledge the importance of real-time performance and long-term consistency for interactive world models and plan to explore these aspects further in future work, building on our preliminary investigations in these areas.

---

> > ### Author Rebuttal · Reviewer_H7CW · 2026-04-04
> >
> > Thank you so much for addressing all my previous concerns, clarifying the intended scope of proposed benchmark, and expanding the Human Preference Validation to analyze performance of additional 13 models including ones with similar quality and conducting additional statistical tests. However, I still have broader concerns about manuscript's quality control given that the original version contains very obvious citations failures such as confusing North Campus Long-Term (NCLT) dataset with National Company Law Tribunal (also NCLT). These failures can either point out to an excessive LLM-assistance at least a the literature review stage or to the insufficient manual checking of the cited papers (or combination of both). In either case it heavily reduces my confidence in the absence of more consequential errors in the literature review, dataset descriptions, benchmark comparisons, metric definitions, or reported analyses.Since this study deploys VLMs to annotate 330k video clips and multi-model verification of generated annotation, it is essential to conduct sufficient manual verification to avoid significant hallucinations that can lead to exaggerated results. Therefore, I feel the results and analyses in the paper are not fully convincing, and I believe that more transparency is needed on to what extent the LLMs were used in this project, beyond the cases described by authors (such as video annotations), including their deployment for literature review, bibliography generation, drafting editing the manuscript, prompt design, formula/metrics generation, code preparation, and result summarization. Additionally, I highly recommend manual checking of all the cited papers.
> > At this point I believe the manuscript will greatly benefit from a systematic auditing rather then local patches, and the rebuttal stage is not sufficient to ensure the paper goes through a thorough revision and a detailed audit to increase the credibility of the paper and the proposed benchmark.

---

> > > ### Author Response · Authors · 2026-04-05
> > >
> > > **Response:**
> > >
> > > Thank you for your detailed feedback and for raising these important concerns. The **citation errors were due to version compilation issues** and have been thoroughly corrected. **VLM annotations were conducted with extensive** **human**, and the use of **LLMs was limited to annotation**. Below, we provide a comprehensive response to address your points:
> > >
> > > **Citation Checking**
> > >
> > > An issue in one version of our bibliography file (within the LaTeX project) caused **incorrect citation numbering** in the compiled PDF, leading to citation inaccuracies in the submitted manuscript. These errors were due to a technical oversight during the compilation process and have now been thoroughly corrected. Specifically:
> > >
> > > 1. The WorldBench reference was updated from Moayeri et al. to the correct paper by Upadhyay et al. [1];
> > >
> > > 2. The ASTRA reference in Appendix E.1 (line 1603) was corrected from Pereverzev et al. to Zhu et al. [2];
> > >
> > > 3. The NCLT dataset reference in Appendix A.1 (line 708) was updated from Kaushik et al. to Carlevaris-Bianco et al. (2016) [3];
> > >
> > > 4. The citation in Appendix A.2 (line 740) was corrected from Simonsen to Gao, Chen, et al. (2024) [4].
> > >
> > > These citation errors were caused by technical issues during the PDF compilation process and **do not affect the literature review, experimental design, or other steps of our work. The datasets used, the content of our paper, and the referenced works remain logically consistent and aligned.**
> > >
> > > [1] Upadhyay, Rishi, et al. "WorldBench: Disambiguating Physics for Diagnostic Evaluation of World Models." *arXiv preprint arXiv:2601.21282* (2026).
> > >
> > > [2] Zhu, Yixuan, et al. "Astra: General Interactive World Model with Autoregressive Denoising." *arXiv preprint arXiv:2512.08931* (2025).
> > >
> > > [3] Carlevaris-Bianco, Nicholas, Arash K. Ushani, and Ryan M. Eustice. "University of Michigan North Campus long-term vision and lidar dataset." *The International Journal of Robotics Research* 35.9 (2016): 1023-1035.
> > >
> > > [4] Gao, Chen, et al. "Embodiedcity: A benchmark platform for embodied agent in real-world city environment." *arXiv preprint arXiv:2410.09604* (2024).
> > >
> > > **VLM Annotation Verification**
> > >
> > > We emphasize that **humans played a central role in ensuring the quality** of the VLM annotation process, with large models serving only as tools to assist in reducing repetitive labor. This **human-supervised annotation approach [1,2,3]** is a widely recognized and adopted paradigm in the industry, ensuring that data quality is rigorously controlled and maintained throughout the process.
> > >
> > > In our previous comment, we outlined the process used to verify VLM annotations, emphasizing the critical role of human involvement. To ensure accuracy and prevent hallucinations or systematic bias in the 330K labels, we employed a multi-model verification approach, where GPT-4o annotations were cross-checked by three diverse models: Gemini 3.0 Flash, Qwen-VL-Max, and Kimi-K2.5. **Any disagreement among models (18.6% of cases) was flagged for human review, with 50 volunteers dedicating approximately 1,200 person-hours to verify and correct annotations.** Ultimately, only 1.2% of the total labels required modification, while 98.8% passed without changes.
> > >
> > > [1] Tan, Zhen, et al. "Large language models for data annotation and synthesis: A survey." *Proceedings of the 2024 Conference on Empirical Methods in* *Natural Language Processing*. 2024.
> > >
> > > [2] Ding, Bosheng, et al. "Is GPT-3 a good data annotator?." *Proceedings of the 61st Annual Meeting of the Association for* *Computational Linguistics* *(Volume 1: Long Papers)*. 2023.
> > >
> > > [3] Chen, Yelin, et al. "RPC-Bench: A Fine-grained Benchmark for Research Paper Comprehension." *arXiv:2601.14289* (2026).
> > >
> > > **Use of LLMs**
> > >
> > > We would like to clarify that **LLMs were not used for essential tasks** such as formula/metrics generation, code preparation, result summarization, or any part of the research or manuscript writing process, all of which were entirely handled by the authors. LLMs were **only used in the video annotation stage**, where VLMs served as tools to assist with labeling. This approach represents a widely recognized paradigm in the industry, where human-VLMs collaboration is used to replace labor-intensive tasks.
> > >
> > > Thank you for **recognizing our previous responses**, particularly the clarification of the benchmark's scope and the expansion of Human Preference Validation. We also **appreciate your continued focus** on broader concerns about the manuscript's quality control and transparency in the use of LLMs. These issues are indeed very important. **We believe that this round of comments can thoroughly address and clarify any lingering doubts.** In fact, **similar concerns raised by other reviewers have already been effectively resolved**, such as Reviewer adW2's questions (Q1, Q2) regarding hallucinations in VLM annotations. We sincerely appreciate your suggestions and look forward to further discussions or feedback, if possible.

---

### Decision · Program_Chairs · 2026-04-30

**Decision:**

Accept (regular)

**Comment:**

This paper proposes iWorld-Bench, a benchmark for evaluating interactive world models with a unified action generation framework across visual generation, trajectory following, and memory tasks. During the rebuttal, the authors clarified the benchmark scope, the annotation quality control and human verification, and provided additional analysis addressing reviewer concerns. Most reviewers gave positive final ratings and found their main concerns resolved. Overall, I finds the work timely and useful to the world model community, and recommends it for publication at ICML 2026. The reviewers also raised several helpful suggestions on terminology, scope, and limitations, which should be addressed in the final camera-ready version.